# Global Convergence Analysis of Local SGD for Two-layer Neural Network without Overparameterization

Yajie Bao[1]     Amanda Shehu[2]     Mingrui Liu[2]*

[1]School of Mathematical Sciences, Shanghai Jiao Tong University, Shanghai, 200240
[2]Department of Computer Science, George Mason University, Fairfax, VA 22030
`baoyajie2019stat@sjtu.edu.cn`, `{ashehu,mingruil}@gmu.edu`

## Abstract

Local SGD, a cornerstone algorithm in federated learning, is widely used in training deep neural networks and shown to have strong empirical performance. A theoretical understanding of such performance on nonconvex loss landscapes is currently lacking. Analysis of the global convergence of SGD is challenging, as the noise depends on the model parameters. Indeed, many works narrow their focus to GD and rely on injecting noise to enable convergence to the local or global optimum. When expanding the focus to local SGD, existing analyses in the nonconvex case can only guarantee finding stationary points or assume the neural network is overparameterized so as to guarantee convergence to the global minimum through neural tangent kernel analysis. In this work, we provide the first global convergence analysis of the vanilla local SGD for two-layer neural networks *without overparameterization* and *without injecting noise*, when the input data is Gaussian. The main technical ingredients of our proof are *a self-correction mechanism* and *a new exact recursive characterization of the direction of global model parameters*. The self-correction mechanism guarantees the algorithm reaches a good region even if the initialization is in a bad region. A good (bad) region means updating the model by gradient descent will move closer to (away from) the optimal solution. The main difficulty in establishing a self-correction mechanism is to cope with the gradient dependency between two layers. To address this challenge, we divide the landscape of the objective into several regions to carefully control the interference of two layers during the correction process. As a result, we show that local SGD can correct the two layers and enter the good region in polynomial time. After that, we establish a new exact recursive characterization of the direction of global parameters, which is the key to showing convergence to the global minimum with linear speedup in the number of machines and reduced communication rounds. Experiments on synthetic data confirm theoretical results.

## 1 Introduction

Federated learning is a prevalent framework in distributed learning to significantly reduce the communication cost and effectively preserve the privacy of local clients [43, 27]. As the most popular algorithm in federated learning, local SGD has shown great empirical success in training deep neural networks (DNNs) [43, 39]. However, existing literature has not been able to fully explain or characterize the convergence of local SGD in training DNNs. Recently, extensive works are devoted to analyzing the convergence of local SGD and its variants in nonconvex optimization [39, 56, 20, 28,

---

*Corresponding Author.

37th Conference on Neural Information Processing Systems (NeurIPS 2023).

32, 40]. However, traditional nonconvex analysis only guarantees convergence to a stationary point, and convergence to the global minimum is in general NP-hard [21].

Despite the NP-hardness results for nonconvex optimization, an increasing body of research tries to address structured nonconvex optimization by first-order methods with noise injection. For instance, Ge et al. [17] considered strict-saddle functions and showed that SGD with isotropic noise can find local minima in polynomial time. This motivated several ensuing works [25, 33, 26, 55, 1] on designing different first-order algorithms to improve convergence to local minima by injecting noise. Noise-injecting schemes and their variants (such as, for instance, broadening from isotropic to anisotropic noise [64]) were shown to help convergence to global minima for many problems that satisfy one of two conditions: (i) local minima are global minima, as in matrix completion [18], dictionary learning [48], and certain deep linear networks [29]; or (ii) neural networks with distributional assumption, such as two-layer neural networks with the Gaussian input [63]. It is worth noting that there is also a rich history on noisy GD based on Langevin dynamics (LD) [30, 44, 52, 8, 41, 9]. For instance, recent work [6] proposes Exponential Family Langevin Dynamics (EFLD) to relax the Gaussian noise assumption and include noisy sign-SGD and variants of drop-out as special cases. When assuming the neural network is overparameterized, neural tangent kernel (NTK) analysis [23] guarantees convergence to the global minimum for local SGD [23, 22, 58, 10]. However, the NTK theory is far from sufficient, since neural networks outperform their NTK counterpart in practice [4] and in theory [38].

Despite existing global convergence analyses of first-order methods for solving nonconvex optimization problems such as neural networks, they either require explicitly injecting noise or assume overparameterization such that NTK analysis can apply. Practical federated learning algorithms such as local SGD do not inject any noise and do not belong to the NTK regime, but they can still converge to global minima. For example, McMahan et al. [43] shows that the local SGD algorithm can achieve around $99\%$ accuracy when training neural networks for an image classification task. This motivates us to study the following question in this paper:

**Is it possible to formally prove that the local SGD algorithm can find global minima for two-layer neural networks without injecting noise and without overparameterization?**

In this paper, we give a positive answer to this question under Gaussian input. We are inspired by a line of work on neural network learning theory with Gaussian input [7, 49, 15, 37, 61, 13] and, in particular address the distributed version of the setting in [13]. Suppose $N$ local machines share the following network:

$$f(\mathbf{Z}, \boldsymbol{w}, \boldsymbol{a}) = \sum_{j=1}^{k} a_j \sigma(\mathbf{Z}_i^\top \boldsymbol{w}),$$

where $\mathbf{Z} = (\mathbf{Z}_1, \cdots, \mathbf{Z}_k) \in \mathbb{R}^{d \times k}$ is the input matrix, $\boldsymbol{w} \in \mathbb{R}^d$ is the weight, $\boldsymbol{a} \in \mathbb{R}^k$ is the output weight, and $\sigma(x) = \max\{x, 0\}$ denotes the ReLU activation function. A good property of this network is positive homogeneity, i.e., $f(\mathbf{Z}, c\boldsymbol{w}, \boldsymbol{a}/c) = f(\mathbf{Z}, \boldsymbol{w}, \boldsymbol{a})$ holds for any $c > 0$. We assume each entry of the input $\mathbf{Z}$ is independently sampled from a standard Gaussian distribution. Then the response is generated by a noiseless teacher network: $y = f(\mathbf{Z}, \boldsymbol{w}^*, \boldsymbol{a}^*)$. Without loss of generality, we further assume $\|\boldsymbol{w}^*\| = 1$. We hope to learn a student network by collaboratively minimizing the following mean square loss among $N$ local machines:

$$L(\boldsymbol{w}, \boldsymbol{a}) = \frac{1}{2}\mathbb{E}\left[(y - f(\mathbf{Z}, \boldsymbol{w}, \boldsymbol{a}))^2\right] = \frac{1}{2}\mathbb{E}\left[(f(\mathbf{Z}, \boldsymbol{w}^*, \boldsymbol{a}^*) - f(\mathbf{Z}, \boldsymbol{w}, \boldsymbol{a}))^2\right]. \quad (1)$$

Obviously, $(\boldsymbol{w}^*, \boldsymbol{a}^*)$ is the global minimum of the objective (1) with zero loss. In particular, the loss function also has a spurious local minimum; hence, minimizing loss is a nonconvex optimization problem. Please refer to work in [13] for further details on the landscape of $L(\boldsymbol{w}, \boldsymbol{a})$.

Despite the special input distribution, to the best of our knowledge, there is currently no work demonstrating the global convergence of vanilla SGD or vanilla local SGD without overparameterization. Work in [13] proved that randomly initialized GD can converge to the global minimum or the local minimum with a constant probability. The initial region where GD can converge to the global minimum is also called the attraction basin, where the gradients of two layers both point in the correct directions to the ground truth. To obtain the global convergence with arbitrary initialization in the same initial region, Zhou et al. [63] proposed a new perturbed GD algorithm by carefully injecting noise to the weights in two layers. Although the convergence of vanilla SGD has not been explored

theoretically, the simulation results in [13, 63] show that vanilla SGD with random initialization can converge to the global minimum with probability 1 when the ratio $|\mathbf{1}^\top \boldsymbol{a}^*|/\|\boldsymbol{a}^*\|$ is large. This result motivates us to investigate the global convergence of local SGD without injecting additional noise to escape the local minimum.

In this paper, we analyze the global convergence of the vanilla local SGD for training a two-layer neural network with Gaussian input, whose initialization starts from the same initial region as in [13, 63]. Formally, our main contributions are summarized as follows:

1. We introduce a new self-correction mechanism of local SGD under a condition on $\boldsymbol{a}^*$ (see Assumption 1): the signals from two layers can be corrected in *polynomial* time even though the initial point comes from a bad region where the gradients point to the wrong direction of $(\boldsymbol{w}^*, \boldsymbol{a}^*)$. The condition also explains the simulation results in [13]. The self-correction process is very difficult to analyze due to the mutual influence effect of two layers. To address this challenge, we utilize a novel technique by carefully dividing the landscape of the objective (1) into several regions. In each region, the negative effect from one layer to another layer can be controlled to a negligible scale. We notice that Li and Yuan [37] also showed the self-correction phase of SGD for the two-layer network under Gaussian input. However, the network's structure in [37] is different from the network studied in this paper. In addition, Li and Yuan [37] also required bounding the noise of stochastic gradient and so cannot handle the vanilla SGD with Gaussian noise as in our case.

2. We show the global convergence of local SGD with linear speedup, which indicates that the iteration complexity is divided by the number of machines $N$. In addition, we also show that the communication complexity of local SGD is reduced compared with the naive parallel version of SGD which needs to communicate at every iteration. The analysis in the convergence stage is very different from the GD in Du et al. [13]. We establish a new recursive dynamic to characterize the direction of the *global weight* in the first layer. Moreover, the objective is not smooth, since the gradients incorporate the angle between the first layer's weight and the ground truth. Therefore, conventional analysis of local SGD for a general smooth objective [47, 54] cannot be applied in the convergence stage. Due to the inner structure of gradients under Gaussian input, we find that the discrepancy caused by the local updates can shrink as the angle decreases, which enables us to refine the bound of discrepancy to be dominated by the statistical bound of noise.

3. We conduct several simulations on the two-layer neural network to verify the theoretical results. The experiments demonstrate that local SGD indeed corrects the wrong signals from the initial point and exhibits speedup in the convergence stage, corroborating our theoretical results. The simulation results also verify that the condition imposed on $|\mathbf{1}^\top \boldsymbol{a}^*|/\|\boldsymbol{a}^*\|$ is almost necessary to show the convergence with almost arbitrary initialization.

## 2 Related Work

**Federated Optimization**   There is a wave of studies on federated optimization in different settings. In the convex optimization setting with homogeneous data, one-shot averaging was studied [65, 42, 60], where each machine solves a local optimization problem and the average happens only at the last iterate. Local SGD skips communication rounds, and the convergence analysis is shown for convex [47, 11, 34, 28, 54, 53, 32, 31] and nonconvex optimization problems [62, 24, 51, 39, 20, 56, 34, 28, 45, 59, 32]. There is a line of work which tried to compare minibatch SGD and local SGD in federated learning [54, 53]. However, these optimization algorithms only work for black-box functions and do not utilize the property of neural networks. As neural network loss landscapes are typically nonconvex, these federated optimization algorithm can only guarantee to find a stationary point instead of a global minimum.

**Optimization Theory for Neural Networks**   There is a line of work studying two layer neural networks with Gaussian input [49, 12, 37, 61, 7, 19, 5]. Li and Liang [36] studied two layer neural networks with cross-entropy loss and showed that SGD can find the global minimum when the neural network is overparameterized. Du et al. [16] proved that GD can find the global minimum for two layer overparameterized neural networks under $\ell_2$ loss. These results are later extended to deep neural networks by [14, 3, 66, 2] but are not directly applicable to analyzing local SGD in the distributed setting.

**Algorithm 1** Local SGD for training two-layer neural network with Gaussian input

---

Initialize $\boldsymbol{v}_0 \in \mathbb{S}^{d-1}$ and $\boldsymbol{a}_0 \in \mathbb{B}^k\left(\frac{|\mathbf{1}^\top \boldsymbol{a}^*|}{\sqrt{k}}\right)$.
**for** $r = 0, \ldots, R-1$ **do**
   **for** $i = 1, \ldots, N$ **do**
      Synchronization: $\boldsymbol{a}_{t_r}^i \leftarrow \boldsymbol{a}_{t_r}$ and $\boldsymbol{v}_{t_r}^i \leftarrow \boldsymbol{v}_{t_r}$.
      **for** $t = t_r, \ldots, t_{r+1} - 1$ **do**
         Sample $\mathbf{Z}_t^i$ from the standard Gaussian distribution and generate the response $y_t^i$.
         Update $\boldsymbol{a}_{t+1}^i = \boldsymbol{a}_t^i - \eta \nabla_{\boldsymbol{a}} \ell(\boldsymbol{v}_t^i, \boldsymbol{a}_t^i; \mathbf{Z}_t^i, y_t^i)$.
         Update $\boldsymbol{v}_{t+1}^i = \boldsymbol{v}_t^i - \eta \nabla_{\boldsymbol{v}} \ell(\boldsymbol{v}_t^i, \boldsymbol{a}_t^i; \mathbf{Z}_t^i, y_t^i)$.
      **end for**
   **end for**
   Update $\boldsymbol{a}_{t_{r+1}} = \frac{1}{N} \sum_{i=1}^N \boldsymbol{a}_{t_{r+1}}^i$ and $\boldsymbol{v}_{t_{r+1}} = \frac{1}{N} \sum_{i=1}^N \boldsymbol{v}_{t_{r+1}}^i$.
**end for**

---

**Federated Learning on Neural Networks**    There is a line of work which studied federated learning algorithms on overparameterized neural networks under the NTK regime [35, 22, 10, 58, 57]. In contrast, our analysis does not fall in the NTK regime: we directly study the dynamics of local SGD over neural networks without overparametrization.

## 3   Notations and Problem Setup

Denote $\|\cdot\|$ the Euclidean norm and $\langle\cdot,\cdot\rangle$ by the inner product. $\mathbb{S}^{d-1}$ denotes the $d$-dimensional unit sphere and $\mathbb{B}^k(\rho)$ denotes the $k$-dimensional ball with center zero and radius $\rho$. For two vectors $\boldsymbol{v}, \boldsymbol{w} \in \mathbb{R}^d$, denote $\angle(\boldsymbol{v}, \boldsymbol{w}) \in [0, \pi]$ the angle between $\boldsymbol{v}$ and $\boldsymbol{w}$. The uniform distribution is denoted by $\mathrm{Unif}(\cdot)$. Moreover, we use $\widetilde{O}$ to hide logarithmic factors.

We adopt the weight-normalization technique [46] to the first layer by re-parametrizing $\boldsymbol{w} = \boldsymbol{v}/\|\boldsymbol{v}\|$, which leads to the following prediction model:

$$f(\mathbf{Z}, \boldsymbol{v}, \boldsymbol{a}) = \sum_{j=1}^k a_j \frac{\sigma(\mathbf{Z}_i^\top \boldsymbol{v})}{\|\boldsymbol{v}\|}. \tag{2}$$

Given any sample $(\mathbf{Z}, y)$, we denote the empirical loss by

$$\ell(\boldsymbol{v}, \boldsymbol{a}; \mathbf{Z}, y) = \frac{1}{2}(y - f(\mathbf{Z}, \boldsymbol{v}, \boldsymbol{a}))^2 = \frac{1}{2}\left(f(\mathbf{Z}, \boldsymbol{w}^*, \boldsymbol{a}^*) - f(\mathbf{Z}, \boldsymbol{v}, \boldsymbol{a})\right)^2. \tag{3}$$

In the distributed environment, suppose there are $N$ local machines sharing the same teacher model $f(\mathbf{Z}, \boldsymbol{w}^*, \boldsymbol{a}^*)$. It means that given any local input $\mathbf{Z}^i$, the response is $y^i = f(\mathbf{Z}^i, \boldsymbol{w}^*, \boldsymbol{a}^*)$. Let $\mathcal{I} = \{t_0, ..., t_R\}$ be the set of synchronization time, where $t_0 = 0$, $t_R = T$ and $t_{r+1} - t_r = I$ for any $r$. The detailed procedure of local SGD is presented in Algorithm 1, where the initial point is from the same region in Du et al. [13] and Zhou et al. [63]: $\boldsymbol{v}_0 \in \mathbb{S}^{d-1}$ and $\boldsymbol{a}_0 \in \mathbb{B}^k(|\mathbf{1}^\top \boldsymbol{a}^*|/\sqrt{k})$. In each round, the $i$-th machine runs $I$ steps of SGD using the stochastic gradients $\nabla_{\boldsymbol{v}} \ell(\boldsymbol{v}_t^i, \boldsymbol{a}_t^i; \mathbf{Z}_t^i, y_t^i)$ and $\nabla_{\boldsymbol{a}} \ell(\boldsymbol{v}_t^i, \boldsymbol{a}_t^i; \mathbf{Z}_t^i, y_t^i)$ computed by the local input $(\mathbf{Z}_t^i, y_t^i)$. At the end of a round, the server aggregates local weights to obtain the global weight and then synchronizes the global weight to each machine.

## 4   Theoretical Analysis

We first introduce two auxiliary sequences $\boldsymbol{v}_t = \sum_{i=1}^N \boldsymbol{v}_t^i/N$ and $\boldsymbol{a}_t = \sum_{i=1}^N \boldsymbol{a}_t^i/N$, which often appears in the analysis of local SGD [47, 53, 28]. Notice that $(\boldsymbol{v}_t, \boldsymbol{a}_t) = (\boldsymbol{v}_t^i, \boldsymbol{a}_t^i)$ for $i \in [N]$ if $t$ is the synchronization time, i.e., $t \in \mathcal{I}$. Denote $\boldsymbol{w}_t^i = \boldsymbol{v}_t^i/\|\boldsymbol{v}_t^i\|$ the local normalized weight and $\mathbf{P}_t^i = (\mathbf{I} - \boldsymbol{w}_t^i(\boldsymbol{w}_t^i)^\top)/\|\boldsymbol{v}_t^i\|$ the respective projection matrix. Now let $L(\boldsymbol{w}, \boldsymbol{a}; \mathbf{Z})$ be the empirical version of the loss function $L(\boldsymbol{w}, \boldsymbol{a})$ defined in (1), that is $L(\boldsymbol{w}, \boldsymbol{a}; \mathbf{Z}) = \frac{1}{2}(f(\mathbf{Z}, \boldsymbol{w}^*, \boldsymbol{a}^*) - f(\mathbf{Z}, \boldsymbol{w}, \boldsymbol{a}))^2$. Since $y = f(\mathbf{Z}, \boldsymbol{w}^*, \boldsymbol{a}^*)$, we will hide $y$ and write $\ell(\boldsymbol{v}, \boldsymbol{a}; \mathbf{Z}) \equiv \ell(\boldsymbol{v}, \boldsymbol{a}; \mathbf{Z}, y)$ hereafter. Notice that $L(\boldsymbol{w}, \boldsymbol{a}; \mathbf{Z}) = \ell(\boldsymbol{v}, \boldsymbol{a}; \mathbf{Z})$ for $\boldsymbol{w} = \boldsymbol{v}/\|\boldsymbol{v}\|$. By recalling the normalized model (2), we have

$$\nabla_{\boldsymbol{v}} \ell(\boldsymbol{v}_t^i, \boldsymbol{a}_t^i; \mathbf{Z}_t^i) = \mathbf{P}_t^i \nabla_{\boldsymbol{w}} L(\boldsymbol{w}_t^i, \boldsymbol{a}_t^i; \mathbf{Z}_t^i), \quad \nabla_{\boldsymbol{a}} \ell(\boldsymbol{v}_t^i, \boldsymbol{a}_t^i; \mathbf{Z}_t^i) = \nabla_{\boldsymbol{a}} L(\boldsymbol{w}_t^i, \boldsymbol{a}_t^i; \mathbf{Z}_t^i).$$

According to Algorithm 1, the updates of auxiliary sequences can be written as

$$\boldsymbol{v}_{t+1} = \boldsymbol{v}_t - \eta \frac{1}{N} \sum_{i=1}^{N} \mathbf{P}_t^i \nabla_{\boldsymbol{w}} L(\boldsymbol{w}_t^i, \boldsymbol{a}_t^i; \mathbf{Z}_t^i), \quad \boldsymbol{a}_{t+1} = \boldsymbol{a}_t - \eta \frac{1}{N} \sum_{i=1}^{N} \nabla_{\boldsymbol{a}} L(\boldsymbol{w}_t^i, \boldsymbol{a}_t^i; \mathbf{Z}_t^i).$$

For ease of technical presentation, we denote the averaged noise terms by $\boldsymbol{\xi}_t = \frac{1}{N} \sum_{i=1}^{N} \mathbf{P}_t^i \boldsymbol{\xi}_t^i$ and $\boldsymbol{\epsilon}_t = \frac{1}{N} \sum_{i=1}^{N} \boldsymbol{\epsilon}_t^i$, where $\boldsymbol{\xi}_t^i = \nabla_{\boldsymbol{w}} L(\boldsymbol{w}_t^i, \boldsymbol{a}_t^i; \mathbf{Z}_t^i) - \nabla_{\boldsymbol{w}} L(\boldsymbol{w}_t^i, \boldsymbol{a}_t^i)$ and $\boldsymbol{\epsilon}_t^i = \nabla_{\boldsymbol{a}} L(\boldsymbol{w}_t^i, \boldsymbol{a}_t^i; \mathbf{Z}_t^i) - \nabla_{\boldsymbol{a}} L(\boldsymbol{w}_t^i, \boldsymbol{a}_t^i)$ are the local noises in stochastic gradients. In addition, for the iterates $\boldsymbol{v}_t^i$ and $\boldsymbol{v}_t$, we write $\phi_t^i = \angle(\boldsymbol{v}_t^i, \boldsymbol{w}^*)$ and $\phi_t = \angle(\boldsymbol{v}_t, \boldsymbol{w}^*)$, respectively.

## 4.1 Exact Dynamic of Each Layer

In this subsection, we will give the exact dynamic of each layer in the training process of local SGD, which is the starting point of our analysis. Denote $\mathbf{P}_t = \frac{1}{\|\boldsymbol{v}_t\|}(\mathbf{I} - \boldsymbol{w}_t \boldsymbol{w}_t^\top)$ by the global projection matrix, where $\boldsymbol{w}_t = \boldsymbol{v}_t / \|\boldsymbol{v}_t\|$. The proofs of this subsection are deferred to Appendix A.1.

**Lemma 1.** *Let* $\check{\boldsymbol{v}}_{t+1} = \boldsymbol{v}_t - \eta \left( \mathbf{P}_t \frac{1}{N} \sum_{i=1}^{N} \nabla_{\boldsymbol{w}} L(\boldsymbol{w}_t^i, \boldsymbol{a}_t^i) + \boldsymbol{\xi}_t \right)$ *and* $\check{\phi}_t = \angle(\check{\boldsymbol{v}}_t, \boldsymbol{w}^*)$. *The first layer in local SGD satisfies that*

$$\|\boldsymbol{v}_{t+1}\|^2 \sin^2 \phi_{t+1} = (1 - \eta \lambda_t \cos \phi_t)^2 \|\boldsymbol{v}_t\|^2 \sin^2 \phi_t - 2\eta M_{1,t} + \eta^2 M_{2,t} + H_t, \quad (4)$$

*where* $\lambda_t = \frac{1}{N} \sum_{i=1}^{N} \frac{\pi - \phi_t^i}{2\pi} \frac{(\boldsymbol{a}_t^i)^\top \boldsymbol{a}^*}{\|\boldsymbol{v}_t\|^2}$, $H_t = \|\boldsymbol{v}_{t+1}\|^2 \sin^2 \phi_{t+1} - \|\check{\boldsymbol{v}}_{t+1}\|^2 \sin^2 \check{\phi}_{t+1}$ *and*

$$M_{1,t} = \left( \boldsymbol{v}_t - \eta \mathbf{P}_t \frac{1}{N} \sum_{i=1}^{N} \nabla_{\boldsymbol{w}} L(\boldsymbol{w}_t^i, \boldsymbol{a}_t^i) \right)^\top \left( \mathbf{I} - \boldsymbol{w}^*(\boldsymbol{w}^*)^\top \right) \boldsymbol{\xi}_t, \quad M_{2,t} = \boldsymbol{\xi}_t^\top \left( \mathbf{I} - \boldsymbol{w}^*(\boldsymbol{w}^*)^\top \right) \boldsymbol{\xi}_t.$$

Lemma 1 is crucial to control the angle and show the linear speedup in further analysis. For each local machine, Lemma 5.5 in [13] provides the dynamic of $\sin^2 \phi_t^i$, which cannot characterize the dynamic of global quantity $\sin \phi_t$ due to the nonlinearity. Here we introduce a new *intermediate variable* $\check{\boldsymbol{v}}_{t+1}$ and find an equality (4) to show the recursive relation between $\|\boldsymbol{v}_{t+1}\|^2 \sin^2 \phi_{t+1}$ and $\|\boldsymbol{v}_t\|^2 \sin^2 \phi_t$ through the global quantities, such as $M_{1,t}$ and $M_{2,t}$. It is worthwhile noticing that $M_{1,t}$ is the averaged noise in local SGD, whose variance is divided by the number of clients $N$, namely *linear speedup term*. In fact, we can control the dynamic of $\sin \phi_t$ by upper bounding the *discrepancy term* $H_t$. Let us assume the last three terms in (4) are negligible and $(\boldsymbol{a}_t^i)^\top \boldsymbol{a}^* > 0$ for any $i \in [N]$:

(1) When $\phi_t > \pi/2$, $\|\boldsymbol{v}_t\|^2 \sin^2 \phi_t$ will continuously increase since $\lambda_t \cos \phi_t < 0$. It indicates that $\phi_t$ can decrease to $\pi/2$ if $\|\boldsymbol{v}_t\|$ is upper bounded by a constant, which also means the first layer can be corrected and avoided converging to the spurious local minima.

(2) When $\phi_t < \pi/2$, $\|\boldsymbol{v}_t\|^2 \sin^2 \phi_t$ will continuously decrease to zero. It indicates that $\phi_t$ can converge to zero if $\|\boldsymbol{v}_t\|$ is lower bounded by a constant. The initial region with $\phi_0 < \pi/2$ and $\boldsymbol{a}_0^\top \boldsymbol{a}^* > 0$ is also called the attraction basin in [13].

Through the remarks above, we can see that the *positive* signal of the second layer is crucial to both the self-correction of the first layer and global convergence. Next lemma presents an exact dynamic of the averaged weight in the second layer.

**Lemma 2.** *Let* $A_0 = |\mathbf{1}^\top \boldsymbol{a}^*|^2 - (\mathbf{1}^\top \boldsymbol{a}^*)(\mathbf{1}^\top \boldsymbol{a}_0)$ *and* $g(\phi) = (\pi - \phi) \cos \phi + \sin \phi$. *For local SGD algorithm started with the initial point* $(\boldsymbol{v}_0, \boldsymbol{a}_0)$, *the second layer satisfies that*

$$\boldsymbol{a}_t^\top \boldsymbol{a}^* = \left( 1 - \eta \frac{\pi - 1}{2\pi} \right)^t \boldsymbol{a}_0^\top \boldsymbol{a}^* + \frac{1 - \left( \frac{2\pi - \eta(k + \pi - 1)}{2\pi - \eta(\pi - 1)} \right)^t}{k} \left( 1 - \eta \frac{\pi - 1}{2\pi} \right)^t A_0$$

$$+ \frac{\eta \|\boldsymbol{a}^*\|^2}{2\pi} \sum_{s=0}^{t-1} \left( 1 - \eta \frac{\pi - 1}{2\pi} \right)^{t-1-s} \frac{1}{N} \sum_{i=1}^{N} B_s^i + S(\boldsymbol{\epsilon}_{0:t-1}), \quad (5)$$

*where* $S(\boldsymbol{\epsilon}_{0:t-1})$ *(defined in Appendix A) is the noise term involving* $\boldsymbol{\epsilon}_0, \cdots, \boldsymbol{\epsilon}_{t-1}$ *and*

$$B_s^i = g(\phi_s^i) - 1 + \frac{\eta}{2\pi} \frac{|\mathbf{1}^\top \boldsymbol{a}^*|^2}{\|\boldsymbol{a}^*\|^2} \sum_{l=0}^{s-1} \left( 1 - \eta \frac{\pi + k - 1}{2\pi} \right)^{s-1-l} (\pi - g(\phi_l^i)).$$

| $\alpha$ | 1/64 | 1/32 | 1/16 | 1/8 | 1/4 | 1/2 |
|---|---|---|---|---|---|---|
| Local SGD | 0.61 | 0.62 | 0.69 | 0.93 | 1.00 | 1.00 |
| Minibatch SGD | 0.60 | 0.62 | 0.69 | 0.91 | 1.00 | 1.00 |

Table 1: Success probabilities of converging to the global minima under different values of $\alpha = |\mathbf{1}^\top \boldsymbol{a}^*|^2/(k\|\boldsymbol{a}^*\|^2)$. The simulation setting is given in Section 5.

Notice that the first term $g(\phi_s^i) - 1$ in $B_s^i$ is negative whenever $\phi_s^i < \pi/2$, but the second term in $B_s^i$ is always positive for $\phi_s^i \in (0, \pi]$. In particular, $\pi - g(\phi_l^i)$ tends to be larger when $\phi_l^i$ gets closer to $\pi$ (i.e., $\boldsymbol{w}_t^i$ drifts away from $\boldsymbol{w}^*$). This insight provides a possibility that local SGD can correct the signal of the second layer by itself, instead of injecting additional noise like [63].

Through carefully inspecting the ingredients of dynamics, we have the following roadmap to show the global convergence of local SGD: (1) For arbitrary initialization $(\boldsymbol{v}_0, \boldsymbol{a}_0)$ except for a measure zero set, where the angle of the first layer between initialization and the global minimum is $\pi$, show that $\boldsymbol{a}_t^\top \boldsymbol{a}^*$ can turn to the positive signal in polynomial time; (2) Show that $\boldsymbol{a}_t^\top \boldsymbol{a}^*$ can be lower bounded by a positive constant value and $\phi_t$ will decrease below $\pi/2$. (3) After entering the attraction basin, show that $(\boldsymbol{w}_t, \boldsymbol{a}_t)$ will converge to the ground truth with a linear speedup guarantee.

### 4.2 Self-correction of Signals in Two Layers

In this subsection, we will show the iterates of local SGD can enter the attraction basin such that $\boldsymbol{a}_t^\top \boldsymbol{a}^* > 0$ and $\phi_t < \pi/2$ after $\widetilde{O}(\eta^{-1})$ steps. Before that, we introduce an assumption on $\boldsymbol{a}^*$.

**Assumption 1.** *Define* $\alpha = |\mathbf{1}^\top \boldsymbol{a}^*|^2/(k\|\boldsymbol{a}^*\|^2)$. *We assume* $k \geq 320(\pi - 1)^2$ *and the ground truth satisfies* $|\mathbf{1}^\top \boldsymbol{a}^*|^2 \leq \frac{k(\pi+k-1)}{720\pi \log(4+k^2)}$ *and*

$$\alpha > \frac{\pi + k - 1}{(\pi - 1)k}\left(1 - \frac{32(\pi - 1)}{k}\right)^{-1}. \tag{6}$$

The conditions on $k$ and $|\mathbf{1}^\top \boldsymbol{a}^*|^2$ are imposed for technical reasons. We believe that a constant lower bound (the right-hand side of (6)) for $\alpha$ is necessary to show the global convergence with almost arbitrary initialization, which is also verified by our simulation results in Table 1. We compute convergence probabilities of local SGD and minibatch SGD under different values of $\alpha$. The initial points are randomly selected by $\boldsymbol{v}_0 \sim \text{Unif}(\mathbb{S}^{d-1})$ and $\boldsymbol{a}_0 \sim \text{Unif}(\mathbb{B}^k(|\mathbf{1}^\top \boldsymbol{a}^*|/\sqrt{k}))$ in each trial. If the convergence probability reaches 1, it means that local SGD or minibatch SGD can converge to the global minima with arbitrary initialization except for a measure zero set. When $\alpha \leq 1/8$, both minibatch SGD and local SGD *cannot* converge to the global minima with probability 1.

**Theorem 1** (Self-correction of the second layer). *For any initial point* $(\boldsymbol{v}_0, \boldsymbol{a}_0)$ *with* $\phi_0 \in [0, \pi)$, *we denote* $\tau_a = \inf\{t \geq 0 : \boldsymbol{a}_t^\top \boldsymbol{a}^* \geq \gamma_a\}$ *the first time, where* $\gamma_a = \frac{16(\pi-1)|\mathbf{1}^\top \boldsymbol{a}^*|^2}{k(\pi+k-1)}$. *Under Assumption 1, if*

$$ck^2\sqrt{\log(Ndk/\delta)}\max\left\{\eta(\sqrt{Ik}+I), \eta(\sqrt{Id}+I), \sqrt{\frac{\eta k}{N}}\right\}\frac{\|\boldsymbol{a}^*\|^2}{|\mathbf{1}^\top \boldsymbol{a}^*|^2} < 1, \tag{7}$$

*for a sufficiently large constant* $c$, *then* $\tau_a \leq O(\eta^{-1} \log k)$ *with probability at least* $1 - \delta$.

This theorem completes the first step of the roadmap, the self-correction of the signal in the second layer, whose proof is given in Appendix D.1. If $|\mathbf{1}^\top \boldsymbol{a}^*|^2/\|\boldsymbol{a}^*\|^2 \leq 1/\text{poly}(p)$, Du et al. [13] proved GD can converge to the spurious local minima with the initial condition $\boldsymbol{a}_0^\top \boldsymbol{a}^* < 0$ and $\phi_0 > \pi/2$. Therefore, our results do not contradict theirs because we assume $|\mathbf{1}^\top \boldsymbol{a}^*|^2/\|\boldsymbol{a}^*\|^2$ is large. The simulations of [13, 63] show that the success probability of converging to the global optimum of SGD increases as the ratio $|\mathbf{1}^\top \boldsymbol{a}^*|/\|\boldsymbol{a}^*\|$ increasing. Theorem 1 can potentially explain this phenomenon since the condition (6) will be satisfied eventually when $|\mathbf{1}^\top \boldsymbol{a}^*|/\|\boldsymbol{a}^*\|$ keeps increasing.

**Proof sketch of Theorem 1.** The proof is very technical since the angle $\phi_t^i$ will affect the sign of $B_s^i$ in the dynamic (5) of $\boldsymbol{a}_t^\top \boldsymbol{a}^*$, while controlling $\phi_t^i$ also requires bounding the scale and controlling

the sign of $a_t^\top a^*$. To tackle this challenge: (1) We divide the initial region of $\phi_0$ into four regions: $(0, \tilde\phi^o)$, $[\tilde\phi^o, \pi/2)$, $[\pi/2, \tilde\phi^u)$ and $[\tilde\phi^u, \pi)$; (2) For each of the first three initial regions, we prove that either $a_t^\top a^* \geq \gamma_a$ in $\widetilde O(1/\eta)$ steps or enter the next region in $\widetilde O(1/\eta)$ steps; (3) In the last region $[\tilde\phi^u, \pi)$, we use the condition (6) to show $a_t^\top a^*$ will turn positive in $\widetilde O(1/\eta)$ steps. The dividing technique enables us to control the individual angle $\phi_t^i$ in four "nice" regions where $a_t^\top a^*$ can be corrected with negligible disturbance. Here $\tilde\phi^o = \arccos(1/5)$ is chosen for convenience to avoid dependence on the initial value $\sin\phi_0$ if $\phi_0 < \pi/2$, and $\tilde\phi^u$ is carefully chosen for technical reasons.

**Lemma 3.** *Denote* $\tilde\phi^u = \arccos\left(-\frac{144\pi}{\pi+k-1}\frac{|\mathbf{1}^\top a^*|^2}{k}\log(4+k^2)\right)$ *and*

$$\varpi = \min\left\{\sin\tilde\phi^u \mathbb{1}_{\phi_0\leq\tilde\phi^u} + \sin\phi_0 \mathbb{1}_{\phi_0>\tilde\phi^u},\ \sin\left[\pi - \left\{(20\|a^*\|^2)^{-1} \wedge 1\right\}\right]\right\},$$

*Under the same conditions of Theorem 1. If the learning rate satisfies*

$$c\|a^*\|^2\sqrt{\log(Ndk/\delta)}\max\left\{\eta(\sqrt{Id}+I), \sqrt{\frac{\eta}{N}}\right\} \leq \frac{\varpi^2}{16k}, \tag{8}$$

*for a large absolute constant $c > 0$, we have $\sin\phi_{\tau_a} \geq \frac{\varpi}{15\sqrt{k}}$ with probability at least $1 - \delta$.*

Recalling the dynamic (4), we know $\sin\phi_t$ tends to decrease to 0 when $\phi_t > \pi/2$ and $a_t^\top a^* < 0$. It means that the first layer moves in the direction of spurious local minima during the self-correction process of the second layer. Lemma 3 provides a lower bound for $\sin\phi_{\tau_a}$ to guarantee the first layer will not converge to the local minima. The proof of Lemma 3 is deferred to Appendix D.2.

**Theorem 2** (Self-correction of the first layer). *For the initial point $(v_0, a_0)$ with $a_0^\top a^* \geq \gamma_a$ (defined in Theorem 1) and $\sin\phi_0 > 0$, we denote $\tau_v = \inf\{t \geq 0 : \phi_v \leq \tilde\phi^l\}$, where $\cos\tilde\phi^l = \frac{144\pi}{\pi+k-1}\frac{|\mathbf{1}^\top a^*|^2}{k}\log(4+k^2)$. If the learning rate satisfies*

$$c\|a^*\|^2\sqrt{\log(Ndk/\delta)}\max\left\{\eta(\sqrt{Id}+I), \sqrt{\frac{\eta}{N}}\right\} \leq \sin^2\phi_0, \tag{9}$$

*for a large absolute constant $c > 0$, it holds that $\tau_v \leq O\left(\frac{1}{\eta}\frac{k^2}{\sin^3\phi_0}\right)$ and $a_{\tau_v}^\top a^* \geq \gamma_a/32$ with probability at least $1 - \delta$.*

This theorem finishes the second step of the roadmap to convergence. The detailed proof is deferred to Appendix D.3. Now we give the iteration complexity spent for self-correction for the worst initialization: $a_0^\top a^* < 0$ and $\phi_0 > \max\{\tilde\phi^u, \pi - (\|a^*\|^{-2}/20 \wedge 1)\}$. Since $\sin\tilde\phi^u = O(1)$, plugging the lower bound of $\sin\phi_{\tau_a}$ in Lemma 3, we can get

$$\tau_a + \tau_v \leq O\left(\frac{1}{\eta}\frac{k^{7/2}}{\sin^3\phi_0}\right). \tag{10}$$

**Proof sketch of Theorem 2.** We divide the initial region into three regions: $(\tilde\phi^f, \pi)$, $[\tilde\phi^f, \tilde\phi^l)$ and $[0, \tilde\phi^l]$, where $\tilde\phi^f$ is adapted to the choices for $\tilde\phi^l$ and $\tilde\phi^u$. For the first two regions, we show that $\phi_t$ will enter the next region in polynomial time, while the last region is our target. The difficult part to show the correction of $\phi_t$ falls in how to guarantee $a_t^\top a^*$ will be lower bounded by $O(\gamma_a)$ for any $t \geq 0$, which is divided and conquered by considering the intermediate quantities such as $\tilde\phi^l$, $\tilde\phi^f$ and $\tilde\phi^u$. To better understand the choices for these quantities and $\gamma_a$, we rearrange $B_s^i$ and have the following decomposition

$$B_s^i = \underbrace{\frac{\pi-1}{\pi+k-1}\frac{|\mathbf{1}^\top a^*|^2}{\|a^*\|^2} - 1}_{\zeta_a} - \underbrace{\frac{\pi-1}{\pi+k-1}\frac{|\mathbf{1}^\top a^*|^2}{\|a^*\|^2}\left(1 - \frac{\eta(\pi+k-1)}{2\pi}\right)^s}_{\varrho_a}$$

$$+ \underbrace{g(\phi_s^i) - \frac{\eta}{2\pi}\frac{|\mathbf{1}^\top a^*|^2}{\|a^*\|^2}\sum_{l=0}^{s-1}\left(1 - \eta\frac{\pi+k-1}{2\pi}\right)^{s-1-l}(1 - g(\phi_l^i))}_{\vartheta_a}.$$

Due to the condition (6) in Assumption 1, $\zeta_a$ is positive and used to correct the sign of $a_t^\top a^*$ and keep it positive. Since $\varrho_a$ is still not negligible when $s$ is small, we use $\gamma_a$ to compensate $-\varrho_a$ after enrolled summation. The uncertain part $\vartheta_a$ will be positive whenever $\tilde\phi^l \leq \phi_l^i \leq \tilde\phi^u$ for $l \leq s$.

## 4.3 Convergence with Linear Speedup

With the correction guarantees in the previous subsection, we are ready to proceed with the convergence analysis of local SGD.

**Theorem 3.** *Suppose the initial point $(\boldsymbol{v}_0, \boldsymbol{a}_0)$ satisfies $\boldsymbol{a}_0^\top \boldsymbol{a}^* \geq \gamma_a/32$ and $\phi_0 \leq \tilde{\phi}^l$. For any $\epsilon > 0$, we choose $\eta = \frac{1}{ck^2\|\boldsymbol{a}^*\|^2}\frac{N\epsilon}{d\log(dN/\epsilon\delta)}$ for some absolute constant $c > 0$. If $I \lesssim \frac{d}{\epsilon N}\min\left\{1, \frac{k^2 d^{1/2}}{N^{1/2}}, \frac{k^4 d}{N^2\epsilon}\right\}$ and $\epsilon < \min\{d^{-1}, dk^{-2}\}$, then $\ell(\boldsymbol{v}_T, \boldsymbol{a}_T) \lesssim \epsilon$ holds with probability at least $1 - \delta$ where $T = \widetilde{O}\left(\frac{dk^4}{N\epsilon}\right)$.*

We have the following implications about the result in Theorem 3:

(1) To the best of our knowledge, this is the first convergence result with linear speedup on the number of machines $N$ for two-layer neural networks. Besides, our convergence analysis does not rely on the overparameterization of the width of the second layer (i.e., $k$).

(2) The dependency on $\epsilon$ matches the best-known results of local SGD for strongly convex objective in Woodworth et al. [53]. In fact, the size of the first layer $d$ resembles the variance of stochastic gradients $\sigma^2$ in the traditional optimization literature. We can show $\langle \nabla_{\boldsymbol{a}} L(\boldsymbol{w}_t, \boldsymbol{a}_t), \boldsymbol{a}_t - \boldsymbol{a}^* \rangle \geq \frac{\pi-1}{2\pi}\|\boldsymbol{a}_t - \boldsymbol{a}^*\|^2 - O(\phi_t)\|\boldsymbol{a}^*\|^2$. Therefore, when the first layer converges has converged ($\phi_t \approx 0$), $\boldsymbol{a}_t$ can converge to $\boldsymbol{a}^*$ like the strongly convex regime.

(3) According to the condition on $I$, we can obtain the communication complexity in the convergence stage as

$$R_{\mathrm{conv}} = \frac{T}{I} = \widetilde{O}\left(\max\left\{k^4, k^2\sqrt{\frac{N}{d}}, \frac{N^2\epsilon}{d}\right\}\right). \tag{11}$$

When $N \lesssim \min\{dk^4, \epsilon^{-1}\}$, the communication complexity in this stage can be $R_{\mathrm{conv}} = \widetilde{O}(k^4)$, which is significantly reduced compared with the iteration complexity.

We show local SGD's convergence layer by layer. Next lemma ensures that the weight of the first layer can converge to the ground truth in polynomial time.

**Lemma 4** (Convergence of the first layer). *Under the settings in Theorem 3. Suppose the initial point satisfies $\phi_0 \leq \tilde{\phi}^l$ and $\boldsymbol{a}_0^\top \boldsymbol{a}^* \geq \gamma_a/32$. With probability at least $1 - \delta$, we can guarantee that $\sin^2\phi_t \lesssim \epsilon$ holds for any $\widetilde{O}\left(k^2\eta^{-1}\right) \leq t \leq \widetilde{O}(\eta^{-2})$.*

It is worthwhile noticing that $(\boldsymbol{v}_{\tau_v}, \boldsymbol{a}_{\tau_v})$ in Theorem 2 satisfies the conditions for initial point in Lemma 4. Therefore, local SGD enters the attraction basin after finishing the self-correction process. In fact, showing the complexity $\widetilde{O}(\epsilon^{-1})$ is not trivial based on the traditional analysis of local SGD [47, 54]. The issue comes from the inner-product noise term $M_{1,t}$ and the discrepancy term $H_t$ in Lemma 1, whose enrolled summations can only be bounded by $O(\sqrt{\epsilon/d})$ and $O\{(\sqrt{Id}+I)\epsilon\}$ respectively at the beginning of convergence stage. Thanks to the special structure of gradient, we find that the scales of $M_{1,t}$ and $H_t$ can shrink as $\sin\phi_t$ decreases. In light of this, we can continuously refine the bound by the following contraction: for $(K+1)T_v \leq t \leq \widetilde{O}(\eta^{-2})$ it holds that

$$\sin^2\phi_t \lesssim \max\left\{\left(\sqrt{\frac{\epsilon}{d}} + (\sqrt{Id}+I)\epsilon\right)^{1+\frac{K}{2}}, \epsilon\right\}.$$

By taking $K = O(\log(1/\epsilon))$, we can obtain the target convergence $\sin^2\phi_t \lesssim \epsilon$ with high probability. More details can be found in Appendix E.1.

**Lemma 5** (Convergence of the second layer). *Under the choice for $\eta$ and conditions for $\epsilon$ in Theorem 3. Suppose $\sin^2\phi_t \leq \epsilon$ holds for any $0 \leq t \leq \widetilde{O}(\eta^{-2})$. With probability at least $1 - \delta$, we can guarantee that $\|\boldsymbol{a}_t - \boldsymbol{a}^*\|^2 \lesssim \epsilon$ holds for any $\widetilde{O}\left(\eta^{-1}\right) \leq t \leq \widetilde{O}(\eta^{-2})$.*

The lemma stated above guarantees the convergence of the second layer after that of the first layer, whose proof is deferred to Appendix E.2. Equipped with Lemmas 4 and 5, we can prove Theorem 3 by leveraging the closed form of the objective $L(\boldsymbol{w}, \boldsymbol{a})$ (see Lemma 6).

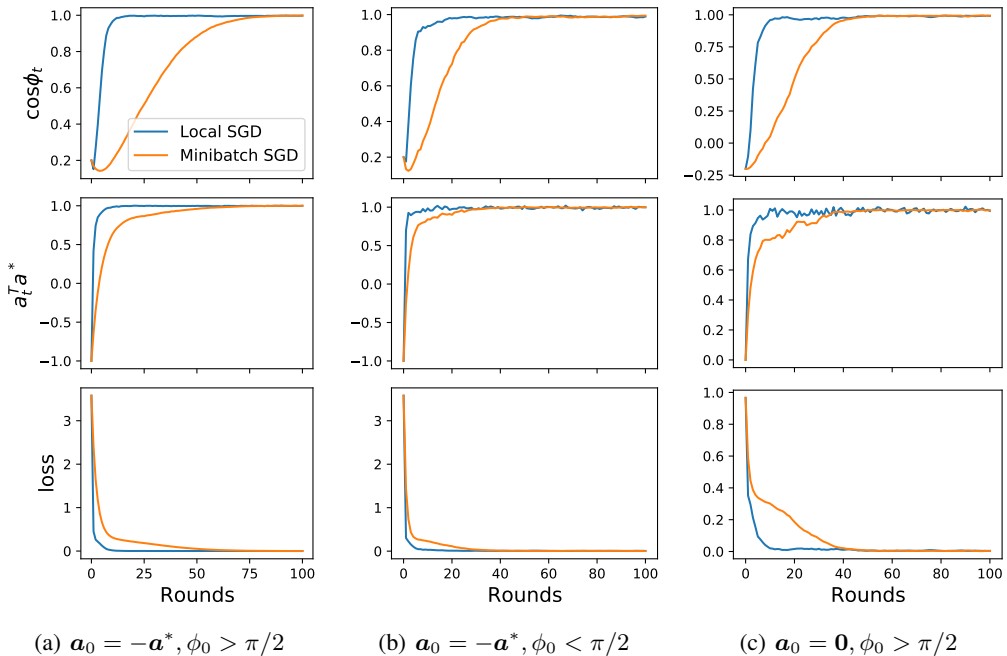

(a) $\boldsymbol{a}_0 = -\boldsymbol{a}^*, \phi_0 > \pi/2$    (b) $\boldsymbol{a}_0 = -\boldsymbol{a}^*, \phi_0 < \pi/2$    (c) $\boldsymbol{a}_0 = \boldsymbol{0}, \phi_0 > \pi/2$

Figure 1: Converged trajectories of local SGD and minibatch SGD with different bad initial points. The dimensions of the two layers are $k = 10$ and $d = 25$. The number of skipped communication is $I = 8$. The batch size is 4. The two algorithms' learning rates are $\eta = 0.1$.

The perturbed GD method in Zhou et al. [63] requires manually adjusting the scale of injected noises and the learning rate between the transition of two phases. In local SGD, we can use a *universal* learning rate in the self-correction stage and convergence stage. Considering an initial point that is closer to spurious local minima ($\phi_0 > \tilde{\phi}^u$) defined in Lemma 3, conditions (7), (8) and (9) on the learning rate can be satisfied if we choose $I = O\{d/(\epsilon N)\}$. Together with (10) and (11), if $N \lesssim \min\{dk^4, \epsilon^{-1}\}$, we can get the total communication complexity

$$R_{\text{total}} = \widetilde{O}\left(\frac{k^{11/2}}{\sin^3 \phi_0}\right).$$

Therefore, our theory shows that local SGD can correct the signals and converge to the global minima with almost arbitrary initialization except for the case of initializing at the local minima ($\phi_0 = \pi$).

## 5    Experiments

We now report some simulation results on synthetic data. We also compare the performance of two algorithms: local SGD and minibatch SGD. At each round, minibatch SGD updates the model weights by using the stochastic gradients with batch size $NI$ in total to update the model, where each local machine computes $I$ gradients and communicates with other machines. Minibatch SGD and local SGD have the same computation and communication structure [54].

In our first simulation, we set $\|\boldsymbol{w}^*\| = \|\boldsymbol{a}^*\| = 1$ and $|\mathbf{1}^\top \boldsymbol{a}^*| = \sqrt{k}$. The results starting from three bad initial regions (i.e., $\phi_0 > \pi/2$ or $\boldsymbol{a}_0^\top \boldsymbol{a}^* \leq 0$) are reported in Figure 1. As we can see, the signals of two layers can be both corrected to a good region (i.e., both $\cos(\phi_t)$ and $\boldsymbol{a}_t^\top \boldsymbol{a}^*$ go to 1 and the loss goes to 0 when $t$ increases) even in the worst case when $\boldsymbol{a}_0 = -\boldsymbol{a}^*$ and $\phi_0 > \pi/2$. An interesting phenomenon is that local SGD can correct the signals faster than minibatch SGD. The reason is that the statistical error of stochastic gradients is not dominating in the self-correction process, so the effect of large batch size in minibatch SGD (i.e., with batch size $NI$ for 1 iteration) is not as useful as smaller batch size in local SGD with more iterations (i.e., batch size 1 for $I$ iterations on each machine with $N$ machines in total). These results corroborate our theoretical analysis.

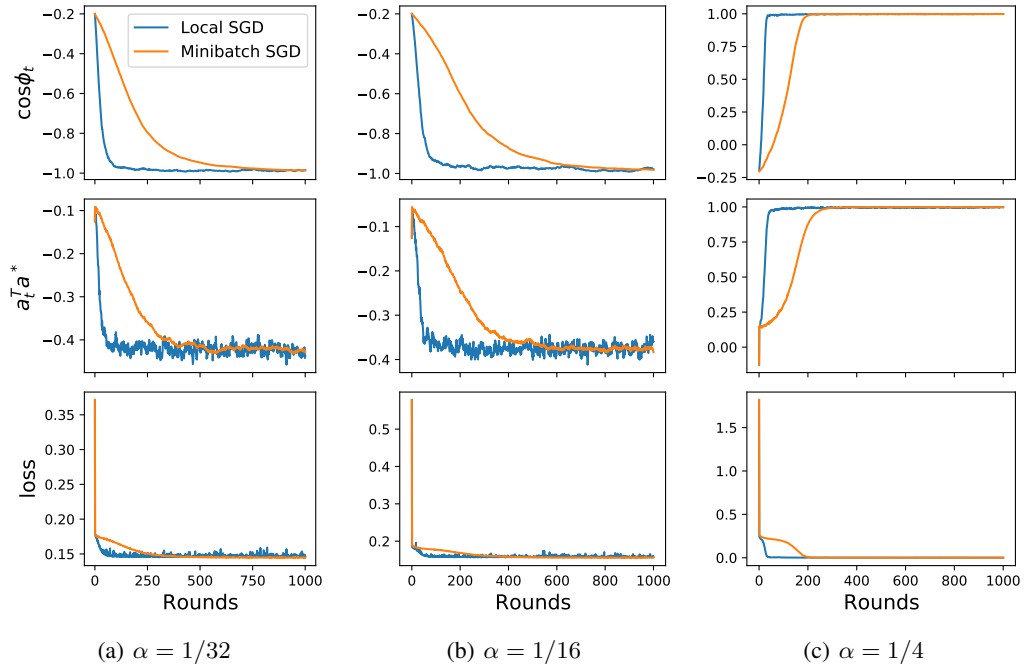

Figure 2: Trajectories of local SGD and minibatch SGD under different values of $\alpha$. The dimensions of the two layers are $k = 64$ and $d = 25$. The number of skipped communication is $I = 8$. The batch size is 16. The two algorithms' learning rates are $\eta = 0.1$.

In the second simulation, we plot the trajectories of local SGD and minibatch SGD under different values of $\alpha$ in Figure 2. Here we set $\boldsymbol{w}^* = \mathbf{1}_d/\sqrt{d}$ and $\boldsymbol{a}^* = (\mathbf{1}_{\alpha k}, 0, \ldots, 0)/\sqrt{\alpha k}$. The bad initial point is fixed as $\boldsymbol{v}_0 = (-1, 0, \ldots, 0)$ and $\boldsymbol{a}_0 = (-\sqrt{\alpha}, 0, \ldots, 0)$, where $\phi_0 > \pi/2$ and $\boldsymbol{a}_0^\top \boldsymbol{a}^* < 0$. When $\alpha = 1/32, 1/16$, we can see that minibatch SGD and local SGD converge to the local minima. When $\alpha = 1/4$, they can converge to the global minima with the same initial point. In this case, as we can see from Figure 2(c), the signals of two layers can be both corrected to a good region.

In the third simulation, we calculate the probabilities that local SGD and minibatch SGD converge to the global minimum under different values of $\alpha$. The averaged results are taken over 100 independent repeated simulations. In each trial, we generate initial points by $\boldsymbol{v}_0 \sim \text{Unif}(\mathbb{S}^{d-1})$ and $\boldsymbol{a}_0 \sim \text{Unif}(\mathbb{B}^k(|\mathbf{1}^\top \boldsymbol{a}^*|/\sqrt{k}))$. The results are given in Table 1.

## 6   Conclusion

We theoretically investigate the convergence of local SGD, a cornerstone algorithm in federated learning with strong empirical performance. We demonstrate convergence to the global minimum for two-layer neural networks *without overparameterization*, *without injecting noise*, and when the input data is Gaussian. A new self-correction mechanism guarantees the algorithm reaches a good region even if the initialization is in a bad region. The landscape of the objective is divided into several regions to carefully control the interference of the two layers during the correction process. A new exact recursive characterization of the direction of global parameter provides the key to show convergence to the global minimum with linear speedup in the number of machines and reduced communication rounds. Experiments on simulated data corroborate the theoretical results. To the best of our knowledge, this work is the first to theoretically demonstrate the global convergence of the vanilla local SGD for neural networks without overparameterization.

## Acknowledgments and Disclosure of Funding

We would like to thank the anonymous reviewers for their helpful comments. Mingrui Liu is supported by a grant from George Mason University. The work of Yajie Bao was done when he was virtually visiting Mingrui Liu's research group in the Department of Computer Science at George Mason University.

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

# Appendix

## Contents

## A  Preliminaries for Analysis

We write $\mathbf{0} = (0, \ldots, 0)^\top$, $\mathbf{1} = (1, \ldots, 1)^\top$ and denote $\mathbf{I}$ by the identity matrix. The following lemma gives the closed forms of the objective and gradients.

**Lemma 6** (Du et al. [13]). *If every entry of $\mathbf{Z}$ is i.i.d. sampled from a Gaussian distribution with mean 0 and variance 1, and $\|\boldsymbol{w}^*\| = 1$, then the population loss is*

$$L(\boldsymbol{w}, \boldsymbol{a}) = \frac{1}{2}\left[\frac{\pi - 1}{2\pi}\|\boldsymbol{a}\|^2 + \frac{\pi - 1}{2\pi}\|\boldsymbol{a}^*\|^2 - \frac{g(\phi) - 1}{\pi}\boldsymbol{a}^\top \boldsymbol{a}^* \right.$$

$$\left. + \frac{1}{2\pi}(\mathbf{1}^\top \boldsymbol{a}^*)^2 + \frac{1}{2\pi}(\mathbf{1}^\top \boldsymbol{a})^2 - \frac{1}{\pi}(\mathbf{1}^\top \boldsymbol{a})(\mathbf{1}^\top \boldsymbol{a})\right], \tag{A.1}$$

*where $g(\phi) = (\pi - \phi)\cos\phi + \sin\phi$. And the the expected gradient of $\boldsymbol{w}$ and $\boldsymbol{a}$ are*

$$\nabla_{\boldsymbol{w}} L(\boldsymbol{w}, \boldsymbol{a}) = -\frac{\pi - \phi}{2\pi}(\boldsymbol{a}^\top \boldsymbol{a}^*) \cdot \boldsymbol{w}^*, \tag{A.2}$$

$$\nabla_{\boldsymbol{a}} L(\boldsymbol{w}, \boldsymbol{a}) = \frac{1}{2\pi}\left[(\mathbf{1}\mathbf{1}^\top + (\pi - 1)\mathbf{I})\boldsymbol{a} - (\mathbf{1}\mathbf{1}^\top + (g(\phi) - 1)\mathbf{I})\boldsymbol{a}^*\right]. \tag{A.3}$$

Denote the noises of stochastic gradients in each local machine $i \in [N]$ by

$$\boldsymbol{\xi}_t^i = \nabla_{\boldsymbol{w}} L(\boldsymbol{w}_t^i, \boldsymbol{a}_t^i; \mathbf{Z}_t^i) - \nabla_{\boldsymbol{w}} L(\boldsymbol{w}_t^i, \boldsymbol{a}_t^i), \quad \boldsymbol{\epsilon}_t^i = \nabla_{\boldsymbol{a}} L(\boldsymbol{w}_t^i, \boldsymbol{a}_t^i; \mathbf{Z}_t^i) - \nabla_{\boldsymbol{a}} L(\boldsymbol{w}_t^i, \boldsymbol{a}_t^i).$$

Then we write the averaged noises as

$$\boldsymbol{\xi}_t = \frac{1}{N}\sum_{i=1}^N \mathbf{P}_t^i \boldsymbol{\xi}_t^i, \quad \boldsymbol{\epsilon}_t = \frac{1}{N}\sum_{i=1}^N \boldsymbol{\epsilon}_t^i.$$

Let $\mathbf{P}_t = \frac{1}{\|\boldsymbol{v}_t\|}(\mathbf{I} - \boldsymbol{w}_t \boldsymbol{w}_t^\top)$. We introduce an auxiliary sequence $\check{\boldsymbol{v}}_t$, which updates by

$$\check{\boldsymbol{v}}_{t+1} = \boldsymbol{v}_t - \eta\left(\mathbf{P}_t \frac{1}{N}\sum_{i=1}^N \nabla_{\boldsymbol{w}} L(\boldsymbol{w}_t^i; \boldsymbol{a}_t^i) + \boldsymbol{\xi}_t\right). \tag{A.4}$$

Then it holds that

$$\check{\boldsymbol{v}}_{t+1} = \boldsymbol{v}_{t+1} - \eta\frac{1}{N}\sum_{i=1}^N (\mathbf{P}_t^i - \mathbf{P}_t)\nabla_{\boldsymbol{w}} L(\boldsymbol{w}_t^i, \boldsymbol{a}_t^i)$$

$$=: \boldsymbol{v}_{t+1} - \eta \boldsymbol{h}_t. \tag{A.5}$$

We define the $\sigma$-filtration in the training process: $\mathcal{F}_0$ is the trivial filtration and

$$\mathcal{F}_t = \sigma\left(\{\boldsymbol{\xi}_s^i, \boldsymbol{\epsilon}_s^i : i \in [N]\}_{s=0}^{t-1}\right), \quad \text{for } t \geq 1.$$

We also write $\mathbb{E}_t[\cdot] = \mathbb{E}[\cdot \mid \mathcal{F}_t]$.

### A.1  Dynamic of the First Layer

**Lemma 1 restated.** The first layer in local SGD satisfies that

$$\|\boldsymbol{v}_{t+1}\|^2 \sin^2\phi_{t+1} = (1 - \eta\lambda_t \cos\phi_t)^2 \|\boldsymbol{v}_t\|^2 \sin^2\phi_t - 2\eta M_{1,t} + \eta^2 M_{2,t} + H_t, \tag{A.6}$$

where $\lambda_t = \frac{1}{N}\sum_{i=1}^N \frac{\pi - \phi_t^i}{2\pi}\frac{(\boldsymbol{a}_t^i)^\top \boldsymbol{a}^*}{\|\boldsymbol{v}_t\|^2}$, $H_t = \|\boldsymbol{v}_{t+1}\|^2 \sin^2\phi_{t+1} - \|\check{\boldsymbol{v}}_{t+1}\|^2 \sin^2\check{\phi}_{t+1}$ and

$$M_{1,t} = (\boldsymbol{v}_t - \eta\mathbf{P}_t\frac{1}{N}\sum_{i=1}^N \nabla_{\boldsymbol{w}} L(\boldsymbol{w}_t^i, \boldsymbol{a}_t^i))^\top \left(\mathbf{I} - \boldsymbol{w}^*(\boldsymbol{w}^*)^\top\right)\boldsymbol{\xi}_t, \quad M_{2,t} = \boldsymbol{\xi}_t^\top \left(\mathbf{I} - \boldsymbol{w}^*(\boldsymbol{w}^*)^\top\right)\boldsymbol{\xi}_t.$$

*Proof.* Let $\tilde{\boldsymbol{v}}_t = \boldsymbol{v}_t - \eta \mathbf{P}_t \frac{1}{N} \sum_{i=1}^N \nabla_{\boldsymbol{w}} L(\boldsymbol{w}_t^i, \boldsymbol{a}_t^i)$, then we have $\check{\boldsymbol{v}}_{t+1} = \tilde{\boldsymbol{v}}_t - \eta \boldsymbol{\xi}_t$. If follows that

$$
\begin{aligned}
\|\check{\boldsymbol{v}}_{t+1}\|^2 \sin^2 \check{\phi}_{t+1} &= \|\check{\boldsymbol{v}}_{t+1}\|^2 - \left(\check{\boldsymbol{v}}_{t+1}^\top \boldsymbol{w}^*\right)^2 \\
&= \|\tilde{\boldsymbol{v}}_t - \eta \boldsymbol{\xi}_t\|^2 - \left[(\tilde{\boldsymbol{v}}_t^\top \boldsymbol{w}^*)^2 - 2\eta(\boldsymbol{\xi}_t^\top \boldsymbol{w}^*)(\tilde{\boldsymbol{v}}_t^\top \boldsymbol{w}^*) + \eta^2(\boldsymbol{\xi}_t^\top \boldsymbol{w}^*)^2\right] \\
&= \|\tilde{\boldsymbol{v}}_t\|^2 - (\tilde{\boldsymbol{v}}_t^\top \boldsymbol{w}^*)^2 - 2\eta \tilde{\boldsymbol{v}}_t^\top (\mathbf{I} - \boldsymbol{w}^*(\boldsymbol{w}^*)^\top)\boldsymbol{\xi}_t + \eta^2 \boldsymbol{\xi}_t^\top (\mathbf{I} - \boldsymbol{w}^*(\boldsymbol{w}^*)^\top)\boldsymbol{\xi}_t \\
&= \|\tilde{\boldsymbol{v}}_t\|^2 - (\tilde{\boldsymbol{v}}_t^\top \boldsymbol{w}^*)^2 - 2\eta M_{1,t} + \eta^2 M_{2,t}.
\end{aligned} \tag{A.7}
$$

From the definition $\lambda_t = \frac{1}{N} \sum_{i=1}^N \frac{\pi - \phi_t^i}{2\pi} \frac{(\boldsymbol{a}_t^i)^\top \boldsymbol{a}^*}{\|\boldsymbol{v}_t\|^2}$ and (A.2), we can write

$$
\begin{aligned}
\tilde{\boldsymbol{v}}_t^\top \boldsymbol{w}^* &= \boldsymbol{v}_t^\top \boldsymbol{w}^* - \eta(\boldsymbol{w}^*)^\top \mathbf{P}_t \frac{1}{N} \sum_{i=1}^N \nabla_{\boldsymbol{w}} L(\boldsymbol{w}_t^i, \boldsymbol{a}_t^i) \\
&= \boldsymbol{v}_t^\top \boldsymbol{w}^* + \eta(\boldsymbol{w}^*)^\top \left(\mathbf{I} - \boldsymbol{w}_t \boldsymbol{w}_t^\top\right) \boldsymbol{w}^* \frac{1}{N} \sum_{i=1}^N \frac{\pi - \phi_t^i}{2\pi} \frac{(\boldsymbol{a}_t^i)^\top \boldsymbol{a}^*}{\|\boldsymbol{v}_t\|} \\
&= \|\boldsymbol{v}_t\| \cos \phi_t + \eta \|\boldsymbol{v}_t\| \left(1 - \cos^2 \phi_t\right) \frac{1}{N} \sum_{i=1}^N \frac{\pi - \phi_t^i}{2\pi} \frac{(\boldsymbol{a}_t^i)^\top \boldsymbol{a}^*}{\|\boldsymbol{v}_t\|^2} \\
&= \|\boldsymbol{v}_t\| \left(\cos \phi_t + \eta \lambda_t \sin^2 \phi_t\right),
\end{aligned} \tag{A.8}
$$

and

$$
\begin{aligned}
\|\tilde{\boldsymbol{v}}_t\|^2 &= \left\| \boldsymbol{v}_t - \eta \mathbf{P}_t \frac{1}{N} \sum_{i=1}^N \nabla_{\boldsymbol{w}} L(\boldsymbol{w}_t^i, \boldsymbol{a}_t^i) \right\|^2 \\
&\overset{(i)}{=} \|\boldsymbol{v}_t\|^2 + \left( \frac{1}{N} \sum_{i=1}^N \frac{\eta}{2\pi} \frac{\pi - \phi_t^i}{\|\boldsymbol{v}_t\|} (\boldsymbol{a}_t^i)^\top \boldsymbol{a}^* \right)^2 (\boldsymbol{w}^*)^\top \left(\mathbf{I} - \boldsymbol{w}_t \boldsymbol{w}_t^\top\right)^2 \boldsymbol{w}^* \\
&\overset{(ii)}{=} \|\boldsymbol{v}_t\|^2 + \left( \frac{1}{N} \sum_{i=1}^N \frac{\eta}{2\pi} \frac{\pi - \phi_t^i}{\|\boldsymbol{v}_t\|} (\boldsymbol{a}_t^i)^\top \boldsymbol{a}^* \right)^2 (1 - \cos^2 \phi_t) \\
&\overset{(iii)}{=} \|\boldsymbol{v}_t\|^2 \left(1 + \eta^2 \lambda_t^2 \sin^2 \phi_t\right),
\end{aligned} \tag{A.9}
$$

where $(i)$ holds since $\mathbf{P}_t \boldsymbol{v}_t = \mathbf{0}$, $(ii)$ is true due to $\mathbf{I} - \boldsymbol{w}_t \boldsymbol{w}_t^\top$ is an idempotent matrix, and $(iii)$ comes from $\|\boldsymbol{w}^*\| = 1$. Combining (A.8) and (A.9), we can get

$$
\begin{aligned}
\|\tilde{\boldsymbol{v}}_t\|^2 - (\tilde{\boldsymbol{v}}_t^\top \boldsymbol{w}^*)^2 &= \left[ 1 + \eta^2 \lambda_t^2 \sin^2 \phi_t - \left(\cos \phi_t + \eta \lambda_t \sin^2 \phi_t\right)^2 \right] \|\boldsymbol{v}_t\|^2 \\
&= \left[\sin^2 \phi_t - 2\eta \lambda_t \cos \phi_t \sin^2 \phi_t + \eta^2 \lambda_t^2 (\sin^2 \phi_t - \sin^4 \phi_t)\right] \|\boldsymbol{v}_t\|^2 \\
&= \left(\sin^2 \phi_t - 2\eta \lambda_t \cos \phi_t \sin^2 \phi_t + \eta^2 \lambda_t^2 \sin^2 \phi_t \cos^2 \phi_t\right) \|\boldsymbol{v}_t\|^2 \\
&= \left(1 - 2\eta \lambda_t \cos \phi_t + \eta^2 \lambda_t^2 \cos^2 \phi_t\right) \|\boldsymbol{v}_t\|^2 \sin^2 \phi_t \\
&= (1 - \eta \lambda_t \cos \phi_t)^2 \|\boldsymbol{v}_t\|^2 \sin^2 \phi_t.
\end{aligned} \tag{A.10}
$$

Plugging (A.10) into (A.7) gives

$$
\|\check{\boldsymbol{v}}_{t+1}\|^2 \sin^2 \check{\phi}_{t+1} = (1 - \eta \lambda_t \cos \phi_t)^2 \|\boldsymbol{v}_t\|^2 \sin^2 \phi_t - 2\eta M_{1,t} + \eta^2 M_{2,t},
$$

where $\mathbb{E}_t[M_{1,t}] = 0$ since $\mathbf{P}_t, \tilde{\boldsymbol{v}}_t \in \mathcal{F}_t$ and $\mathbb{E}_t[\boldsymbol{\xi}_t] = \mathbf{0}$, and $M_{2,t} \geq 0$ because $\mathbf{P}_t$ and $\mathbf{I} - \boldsymbol{w}^*(\boldsymbol{w}^*)^\top$ are both semi positive definite. The result follows from the definition of $H_t$. $\qquad\square$

## A.2 Dynamic of the Second Layer

**Lemma 2 restated.** Let $A_0 = (\mathbf{1}^\top \boldsymbol{a}^*)^2 - (\mathbf{1}^\top \boldsymbol{a}^*)(\mathbf{1}^\top \boldsymbol{a}_0)$. For local SGD algorithm started with the initial point $(\boldsymbol{v}_0, \boldsymbol{a}_0)$, the second layer satisfies that

$$
\boldsymbol{a}_t^\top \boldsymbol{a}^* = \left(1 - \eta \frac{\pi - 1}{2\pi}\right)^t \boldsymbol{a}_0^\top \boldsymbol{a}^* + \frac{1 - \left(\frac{2\pi - \eta(k + \pi - 1)}{2\pi - \eta(\pi - 1)}\right)^t}{k} \left(1 - \eta \frac{\pi - 1}{2\pi}\right)^t A_0
$$

$$+ \frac{\eta \|\boldsymbol{a}^*\|^2}{2\pi} \sum_{s=0}^{t-1} \left(1 - \eta \frac{\pi - 1}{2\pi}\right)^{t-1-s} \frac{1}{N} \sum_{i=1}^{N} B_s^i + S(\boldsymbol{\epsilon}_{0:t-1}), \qquad (A.11)$$

where

$$S(\boldsymbol{\epsilon}_{0:t-1}) = \frac{\eta^2}{2\pi} \sum_{s=0}^{t-1} \left(1 - \eta \frac{\pi - 1}{2\pi}\right)^{t-1-s} \sum_{l=0}^{s-1} \left(1 - \eta \frac{\pi + k - 1}{2\pi}\right)^{s-1-l} (\mathbf{1}^\top \boldsymbol{a}^*)(\mathbf{1}^\top \boldsymbol{\epsilon}_l)$$

$$- \eta \sum_{s=0}^{t-1} \left(1 - \eta \frac{\pi - 1}{2\pi}\right)^{t-1-s} \boldsymbol{\epsilon}_s^\top \boldsymbol{a}^*.$$

and

$$B_s^i = (g(\phi_s^i) - 1) + \frac{\eta}{2\pi} \frac{(\mathbf{1}^\top \boldsymbol{a}^*)^2}{\|\boldsymbol{a}^*\|^2} \sum_{l=0}^{s-1} \left(1 - \eta \frac{\pi + k - 1}{2\pi}\right)^{s-1-l} (\pi - g(\phi_l^i)).$$

*Proof.* From (A.3), we have

$$\frac{1}{N} \sum_{i=1}^{N} \nabla L_{\boldsymbol{a}}(\boldsymbol{w}_t^i, \boldsymbol{a}_t^i) = \frac{1}{2\pi} \left(\mathbf{1}\mathbf{1}^\top + (\pi - 1)\mathbf{I}\right) \left(\frac{1}{N} \sum_{i=1}^{N} \boldsymbol{a}_t^i\right) - \frac{1}{2\pi} \left[\mathbf{1}\mathbf{1}^\top + \left(\frac{1}{N} \sum_{i=1}^{N} g(\phi_t^i) - 1\right) \mathbf{I}\right] \boldsymbol{a}^*$$

$$= \frac{1}{2\pi} \left(\mathbf{1}\mathbf{1}^\top + (\pi - 1)\mathbf{I}\right) \boldsymbol{a}_t - \frac{1}{2\pi} \left[\mathbf{1}\mathbf{1}^\top + \left(\frac{1}{N} \sum_{i=1}^{N} g(\phi_t^i) - 1\right) \mathbf{I}\right] \boldsymbol{a}^*.$$

Let $A_t = (\mathbf{1}^\top \boldsymbol{a}^*)^2 - (\mathbf{1}^\top \boldsymbol{a}^*)(\mathbf{1}^\top \boldsymbol{a}_t)$. By the definition of $\boldsymbol{a}_{t+1}$, we know

$$\boldsymbol{a}_{t+1}^\top \boldsymbol{a}^* = \frac{1}{N} \sum_{i=1}^{N} (\boldsymbol{a}_{t+1}^i)^\top \boldsymbol{a}^* = \boldsymbol{a}_t^\top \boldsymbol{a}^* - \eta \frac{1}{N} \sum_{i=1}^{N} \nabla_{\boldsymbol{a}} L(\boldsymbol{w}_t^i, \boldsymbol{a}_t^i)^\top \boldsymbol{a}^* - \eta \boldsymbol{\epsilon}_t^\top \boldsymbol{a}^*$$

$$= \left(1 - \eta \frac{\pi - 1}{2\pi}\right) \boldsymbol{a}_t^\top \boldsymbol{a}^* + \frac{\eta}{2\pi} \left(\frac{1}{N} \sum_{i=1}^{N} g(\phi_t) - 1\right) \|\boldsymbol{a}^*\|^2 + \frac{\eta}{2\pi} A_t - \eta \boldsymbol{\epsilon}_t^\top \boldsymbol{a}^*, \quad (A.12)$$

For $t \geq 0$, we also have

$$A_{t+1} = (\mathbf{1}^\top \boldsymbol{a}^*)^2 - \left(1 - \eta \frac{\pi + k - 1}{2\pi}\right) (\mathbf{1}^\top \boldsymbol{a}^*)(\mathbf{1}^\top \boldsymbol{a}_t)$$

$$- \eta \frac{k + \frac{1}{N} \sum_{i=1}^{N} g(\phi_t^i) - 1}{2\pi} (\mathbf{1}^\top \boldsymbol{a}^*)^2 + \eta (\mathbf{1}^\top \boldsymbol{a}^*)(\mathbf{1}^\top \boldsymbol{\epsilon}_t)$$

$$= \left(1 - \eta \frac{\pi + k - 1}{2\pi}\right) A_t + \eta \frac{\pi - \frac{1}{N} \sum_{i=1}^{N} g(\phi_t^i)}{2\pi} (\mathbf{1}^\top \boldsymbol{a}^*)^2 + \eta (\mathbf{1}^\top \boldsymbol{a}^*)(\mathbf{1}^\top \boldsymbol{\epsilon}_t)$$

$$= \left(1 - \eta \frac{\pi + k - 1}{2\pi}\right)^{t+1} A_0 + \frac{\eta (\mathbf{1}^\top \boldsymbol{a}^*)^2}{2\pi} \sum_{s=0}^{t} \left(1 - \eta \frac{\pi + k - 1}{2\pi}\right)^{t-s} \left(\pi - \frac{1}{N} \sum_{i=1}^{N} g(\phi_s^i)\right)$$

$$+ \eta \sum_{s=0}^{t} \left(1 - \eta \frac{\pi + k - 1}{2\pi}\right)^{t-s} (\mathbf{1}^\top \boldsymbol{a}^*)(\mathbf{1}^\top \boldsymbol{\epsilon}_s). \qquad (A.13)$$

Plugging (A.13) into (A.12), then enrolling (A.12) from timestamp 0 to $t$ gives that

$$\boldsymbol{a}_{t+1}^\top \boldsymbol{a}^* = \left(1 - \eta \frac{\pi - 1}{2\pi}\right)^{t+1} \boldsymbol{a}_0^\top \boldsymbol{a}^* + \frac{\eta \|\boldsymbol{a}^*\|^2}{2\pi} \sum_{s=0}^{t} \left(1 - \eta \frac{\pi - 1}{2\pi}\right)^{t-s} \left(\frac{1}{N} \sum_{i=1}^{N} g(\phi_s^i) - 1\right)$$

$$+ \left(\frac{\eta |\mathbf{1}^\top \boldsymbol{a}^*|}{2\pi}\right)^2 \sum_{s=0}^{t} \left(1 - \eta \frac{\pi - 1}{2\pi}\right)^{t-s} \sum_{l=0}^{s-1} \left(1 - \eta \frac{\pi + k - 1}{2\pi}\right)^{s-1-l} \left(\pi - \frac{1}{N} \sum_{i=1}^{N} g(\phi_l^i)\right)$$

$$+ \frac{\eta^2}{2\pi} \sum_{s=0}^{t} \left(1 - \eta \frac{\pi - 1}{2\pi}\right)^{t-s} \sum_{l=0}^{s-1} \left(1 - \eta \frac{\pi + k - 1}{2\pi}\right)^{s-1-l} (\mathbf{1}^\top \boldsymbol{a}^*)(\mathbf{1}^\top \boldsymbol{\epsilon}_l)$$

$$- \eta \sum_{s=0}^{t} \left( 1 - \eta \frac{\pi - 1}{2\pi} \right)^{t-s} \boldsymbol{\epsilon}_s^\top \boldsymbol{a}^* + \frac{1 - \left( \frac{2\pi - \eta(k+\pi-1)}{2\pi - \eta(\pi-1)} \right)^t}{k} \left( 1 - \eta \frac{\pi - 1}{2\pi} \right)^{t+1} A_0$$

$$= \left( 1 - \eta \frac{\pi - 1}{2\pi} \right)^t \boldsymbol{a}_0^\top \boldsymbol{a}^* + \frac{1 - \left( \frac{2\pi - \eta(k+\pi-1)}{2\pi - \eta(\pi-1)} \right)^t}{k} \left( 1 - \eta \frac{\pi - 1}{2\pi} \right)^t A_0$$

$$+ \frac{\eta \|\boldsymbol{a}^*\|^2}{2\pi} \sum_{s=0}^{t-1} \left( 1 - \eta \frac{\pi - 1}{2\pi} \right)^{t-1-s} \frac{1}{N} \sum_{i=1}^{N} B_s^i + S(\boldsymbol{\epsilon}_{0:t-1}).$$

In fact, we also used the following summation result in the first equality

$$\frac{\eta}{2\pi} \sum_{s=0}^{t} \left( 1 - \eta \frac{\pi - 1}{2\pi} \right)^{t-s} \left( 1 - \eta \frac{\pi + k - 1}{2\pi} \right)^s$$

$$= \frac{\eta}{2\pi} \left( 1 - \eta \frac{\pi - 1}{2\pi} \right)^t \sum_{s=0}^{t} \left( \frac{1 - \eta \frac{\pi + k - 1}{2\pi}}{1 - \eta \frac{\pi - 1}{2\pi}} \right)^s$$

$$= \frac{\eta}{2\pi} \left( 1 - \eta \frac{\pi - 1}{2\pi} \right)^t \frac{2\pi - \eta(\pi - 1)}{\eta k} \left[ 1 - \left( \frac{2\pi - \eta(k + \pi - 1)}{2\pi - \eta(\pi - 1)} \right)^{t+1} \right]$$

$$= \frac{1 - \left( \frac{2\pi - \eta(k+\pi-1)}{2\pi - \eta(\pi-1)} \right)^{t+1}}{k} \left( 1 - \eta \frac{\pi - 1}{2\pi} \right)^{t+1}.$$

$\square$

## A.3   Concentration Inequalities for the Noises

### A.3.1   Sub-exponential Martingale Difference

**Definition A.1** (Sub-exponential and sub-gaussian random variable)**.** *The random variable* $\mathbf{x} \in \mathbb{R}$ *is called sub-exponential with norm* $\|\mathbf{x}\|_{\psi_1}$ *if*

$$\|\mathbf{x}\|_{\psi_1} = \inf \left\{ s \geq 0 : \mathbb{E} \left[ e^{|\mathbf{x}|/s} \right] \leq 2 \right\}.$$

*In addition, the random variable* $\mathbf{x} \in \mathbb{R}$ *is called sub-gaussian with norm* $\|\mathbf{x}\|_{\psi_2}$ *if*

$$\|\mathbf{x}\|_{\psi_2} = \inf \left\{ s \geq 0 : \mathbb{E} \left[ e^{\mathbf{x}^2/s} \right] \leq 2 \right\}.$$

**Lemma 7** (Proposition 2.6.1 in [50])**.** *Let* $\mathbf{x}_1, ..., \mathbf{x}_n$ *be independent mean-zero sub-gaussian random variables. Then* $\sum_{i=1}^{N} \mathbf{x}_i$ *is also a sub-gaussian random variable with*

$$\left\| \sum_{i=1}^{N} \mathbf{x}_i \right\|_{\psi_2}^2 \lesssim \sum_{i=1}^{N} \|\mathbf{x}_i\|_{\psi_2}^2.$$

**Lemma 8** (Lemma 2.6.8 in [50])**.** *If* $\mathbf{x}$ *is a sub-gaussian random variable, then* $\mathbf{x} - \mathbb{E}[\mathbf{x}]$ *is sub-gaussian too and*

$$\|\mathbf{x} - \mathbb{E}[\mathbf{x}]\|_{\psi_2} \lesssim \|\mathbf{x}\|_{\psi_2}.$$

**Lemma 9** (Lemma 2.7.7 in [50])**.** *Let* $\mathbf{x}$ *and* $\mathbf{y}$ *be sub-gaussian random variables. Then* $\mathbf{xy}$ *is sub-exponential and*

$$\|\mathbf{xy}\|_{\psi_1} \leq \|\mathbf{x}\|_{\psi_2} \|\mathbf{y}\|_{\psi_2}.$$

**Lemma 10.** *Suppose* $\{\mathbf{x}_t\}_{t \geq 1} \subseteq \mathbb{R}$ *are sub-exponential martingale difference sequence with norm* $\{\|\mathbf{x}_t\|_{\psi_1}\}_{t \geq 1}$ *adapted to the filtration* $\{\mathcal{F}_t\}_{t \geq 1}$, *that is* $\mathbb{E}[\mathbf{x}_t | \mathcal{F}_t] = 0$ *and* $\|\mathbf{x}_t\|_{\psi_1} = \inf\{s \geq 0 : \mathbb{E}[e^{|\mathbf{x}_t|/s} | \mathcal{F}_t] \leq 2\}$. *Then we have*

$$\mathbb{P} \left( \left| \sum_{s=1}^{t} \mathbf{x}_s \right| \geq \frac{1}{c} \sqrt{\log(1/\delta) \sum_{s=1}^{t} \|\mathbf{x}_s\|_{\psi_1}^2} + c \log(1/\delta) \max_{1 \leq s \leq t} \|\mathbf{x}_s\|_{\psi_1} \right) \leq 2\delta,$$

*where* $c > 0$ *is an absolute constant.*

*Proof.* Invoking Proposition 2.7.1 in [50], we know

$$\mathbb{E}\left[\exp(\lambda \mathbf{x}_t) \mid \mathcal{F}_t\right] \leq \exp\left(c^{-2}\lambda^2 \|\mathbf{x}_t\|_{\psi_1}^2\right), \quad \text{if } |\lambda| \leq \frac{c}{\|\mathbf{x}_t\|_{\psi_1}}. \tag{A.14}$$

Define

$$M_t = \exp\left\{\lambda \sum_{s=1}^{t} \mathbf{x}_s - c^{-2}\lambda^2 \sum_{s=1}^{t} \|\mathbf{x}_s\|_{\psi_1}^2\right\}.$$

It follows from (A.14) that for any $t \geq 1$,

$$\mathbb{E}\left[M_t \mid \mathcal{F}_t\right] = M_{t-1}\mathbb{E}\left[\exp\left\{\lambda \mathbf{x}_t - c^{-2}\lambda^2 \|\mathbf{x}_t\|_{\psi_1}^2\right\} \mid \mathcal{F}_t\right] \leq M_{t-1}.$$

It further implies that

$$\mathbb{E}\left[M_t\right] = \mathbb{E}\left[\mathbb{E}\left[M_t \mid \mathcal{F}_t\right]\right] \leq \mathbb{E}\left[M_{t-1}\right] \leq \cdots \leq \mathbb{E}\left[M_1\right] \leq 1.$$

Using Markov's inequality, for $0 < \lambda \leq \frac{c}{\max_{1\leq s\leq t}\|\mathbf{x}_t\|_{\psi_1}}$, we have

$$\mathbb{P}\left(\sum_{s=1}^{t}\mathbf{x}_s \geq c^{-2}\lambda \sum_{s=1}^{t}\|\mathbf{x}_s\|_{\psi_1}^2 + \frac{1}{\lambda}\log(1/\delta)\right) = \mathbb{P}\left(M_t \geq \frac{1}{\delta}\right) \leq \delta. \tag{A.15}$$

Now taking

$$\lambda = \min\left\{\sqrt{\frac{c^2\log(1/\delta)}{\sum_{s=1}^{t}\|\mathbf{x}_s\|_{\psi_1}^2}}, \frac{c}{\max_{1\leq s\leq t}\|\mathbf{x}_s\|_{\psi_1}}\right\},$$

and plugging it into (A.15) gives

$$\mathbb{P}\left(\sum_{s=1}^{t}\mathbf{x}_s \geq \frac{1}{c}\sqrt{\log(1/\delta)\sum_{s=1}^{t}\|\mathbf{x}_s\|_{\psi_1}^2} + c\log(1/\delta)\max_{1\leq s\leq t}\|\mathbf{x}_s\|_{\psi_1}\right) \leq \delta.$$

Similarly, we can also prove

$$\mathbb{P}\left(\sum_{s=1}^{t}\mathbf{x}_s \leq -\frac{1}{c}\sqrt{\log(1/\delta)\sum_{s=1}^{t}\|\mathbf{x}_s\|_{\psi_1}^2} - c\log(1/\delta)\max_{1\leq s\leq t}\|\mathbf{x}_s\|_{\psi_1}\right) \leq \delta.$$

Then the proof is finished. $\qquad\square$

### A.3.2 Scales of Sub-exponential Norm for Noises in Stochastic Gradients

For any $(\boldsymbol{v}, \boldsymbol{a})$ with $\boldsymbol{w} = \boldsymbol{v}/\|\boldsymbol{v}\|$, we denote

$$\boldsymbol{\xi}_{(1)} = \left(\sum_{i=1}^{k}a_i^*\sigma(\mathbf{Z}_i^\top \boldsymbol{w})\right)\left(\sum_{i=1}^{k}a_i\mathbf{Z}_i\mathbb{1}\left\{\mathbf{Z}_i^\top \boldsymbol{w} \geq 0\right\}\right)$$

$$- \mathbb{E}\left[\left(\sum_{i=1}^{k}a_i^*\sigma(\mathbf{Z}_i^\top \boldsymbol{v})\right)\left(\sum_{i=1}^{k}a_i\mathbf{Z}_i\mathbb{1}\left\{\mathbf{Z}_i^\top \boldsymbol{w} \geq 0\right\}\right)\right],$$

$$\boldsymbol{\xi}_{(2)} = \left(\sum_{i=1}^{k}a_i^*\sigma(\mathbf{Z}_i^\top \boldsymbol{w}^*)\right)\left(\sum_{i=1}^{k}a_i\mathbf{Z}_i\mathbb{1}\left\{\mathbf{Z}_i^\top \boldsymbol{w} \geq 0\right\}\right)$$

$$- \mathbb{E}\left[\left(\sum_{i=1}^{k}a_i^*\sigma(\mathbf{Z}_i^\top \boldsymbol{w}^*)\right)\left(\sum_{i=1}^{k}a_i\mathbf{Z}_i\mathbb{1}\left\{\mathbf{Z}_i^\top \boldsymbol{w} \geq 0\right\}\right)\right],$$

$$\boldsymbol{\epsilon}_{(1)} = \sigma(\mathbf{Z}\boldsymbol{w})(\boldsymbol{a})^\top\sigma(\mathbf{Z}\boldsymbol{w}) - \mathbb{E}\left[\sigma(\mathbf{Z}\boldsymbol{w})(\boldsymbol{a})^\top\sigma(\mathbf{Z}\boldsymbol{w})\right],$$

$$\boldsymbol{\epsilon}_{(2)} = \sigma(\mathbf{Z}\boldsymbol{w})(\boldsymbol{a}^*)^\top\sigma(\mathbf{Z}\boldsymbol{w}^*) - \mathbb{E}\left[\sigma(\mathbf{Z}\boldsymbol{w})(\boldsymbol{a}^*)^\top\sigma(\mathbf{Z}\boldsymbol{w}^*)\right].$$

**Lemma 11.** *Let* $\mathbf{P} = \frac{1}{\|\boldsymbol{v}\|}\left(\mathbf{I} - \boldsymbol{w}\boldsymbol{w}^\top\right)$. *The following noise terms satisfy: for* $\ell = 1, 2$

(1) *For any fixed vector $\boldsymbol{u} \in \mathbb{R}^k$, $\boldsymbol{u}^\top \boldsymbol{\epsilon}_{(\ell)}$ is sub-exponential with $\max_{\ell=1,2} \|\boldsymbol{u}^\top \boldsymbol{\epsilon}_{(\ell)}\|_{\psi_1} \leq c\|\boldsymbol{u}\| \max\{\|\boldsymbol{a}\|, \|\boldsymbol{a}^*\|\}$.*

(2) *For any fixed vector $\boldsymbol{u} \in \mathbb{R}^d$, $\boldsymbol{u}^\top \mathbf{P} \boldsymbol{\xi}_{(\ell)}$ is sub-exponential with $\max_{\ell=1,2} \|\boldsymbol{u}^\top \mathbf{P} \boldsymbol{\xi}_{(\ell)}\|_{\psi_1} \leq c\|\boldsymbol{a}\|\|\boldsymbol{a}^*\|/\|\boldsymbol{v}\|$.*

*Proof.* Recall the definition of $\boldsymbol{\epsilon}_{(1)}$, then we have

$$(\boldsymbol{\epsilon}_{(1)})^\top \boldsymbol{u} = \boldsymbol{u}^\top \sigma(\mathbf{Z}\boldsymbol{w})(\boldsymbol{a})^\top \sigma(\mathbf{Z}\boldsymbol{w}) - \mathbb{E}\left[\boldsymbol{u}^\top \sigma(\mathbf{Z}\boldsymbol{w})(\boldsymbol{a})^\top \sigma(\mathbf{Z}\boldsymbol{w})\right].$$

Applying Lemma 7 yields

$$
\begin{aligned}
\left\|\boldsymbol{u}^\top \sigma(\mathbf{Z}\boldsymbol{w}) - \mathbb{E}\left[\boldsymbol{u}^\top \sigma(\mathbf{Z}\boldsymbol{w})\right]\right\|_{\psi_2}^2 &= \left\|\sum_{j=1}^k u_j \left(\sigma(\mathbf{Z}_j^\top \boldsymbol{w}) - \mathbb{E}\left[\sigma(\mathbf{Z}_j^\top \boldsymbol{w})\right]\right)\right\|_{\psi_2}^2 \\
&\overset{(i)}{\lesssim} \sum_{j=1}^k \left\|u_j \left(\sigma(\mathbf{Z}_j^\top \boldsymbol{w}) - \mathbb{E}\left[\sigma(\mathbf{Z}_j^\top \boldsymbol{w})\right]\right)\right\|_{\psi_2}^2 \\
&\overset{(ii)}{\lesssim} \sum_{j=1}^k \left\|u_j \sigma(\mathbf{Z}_j^\top \boldsymbol{w})\right\|_{\psi_2}^2 \\
&\overset{(iii)}{\lesssim} \|\boldsymbol{u}\|^2 \|\boldsymbol{w}\|^2 \\
&\overset{(iv)}{=} \|\boldsymbol{u}\|^2,
\end{aligned}
\tag{A.16}
$$

where $(i)$ and $(ii)$ comes from Lemma 7 and 8 respectively, $(iii)$ holds since $\sigma(\mathbf{Z}_j^\top \boldsymbol{w})$ is a half-normal random variable with variance $\|\boldsymbol{w}\|^2/2$, and $(iv)$ holds due to $\|\boldsymbol{w}\| = 1$. Combining (A.16) and Lemma 8, we can verify that

$$\left\|\boldsymbol{u}^\top \sigma(\mathbf{Z}\boldsymbol{w})\right\|_{\psi_2} \lesssim \|\boldsymbol{u}\|, \quad \left\|\boldsymbol{a}^\top \sigma(\mathbf{Z}\boldsymbol{w})\right\|_{\psi_2} \lesssim \|\boldsymbol{a}\|, \quad \left\|(\boldsymbol{a}^*)^\top \sigma(\mathbf{Z}\boldsymbol{w})\right\|_{\psi_2} \lesssim \|\boldsymbol{a}^*\|.$$

Applying Lemma 9 gives

$$\left\|\boldsymbol{u}^\top \sigma(\mathbf{Z}\boldsymbol{w})(\boldsymbol{a})^\top \sigma(\mathbf{Z}\boldsymbol{w})\right\|_{\psi_1} \lesssim \left\|\boldsymbol{u}^\top \sigma(\mathbf{Z}\boldsymbol{w})\right\|_{\psi_2} \left\|\boldsymbol{a}^\top \sigma(\mathbf{Z}\boldsymbol{w})\right\|_{\psi_2} \lesssim \|\boldsymbol{u}\|\|\boldsymbol{a}\|$$

$$\left\|\boldsymbol{u}^\top \sigma(\mathbf{Z}\boldsymbol{w})(\boldsymbol{a}^*)^\top \sigma(\mathbf{Z}\boldsymbol{w})\right\|_{\psi_1} \lesssim \left\|\boldsymbol{u}^\top \sigma(\mathbf{Z}\boldsymbol{w})\right\|_{\psi_2} \left\|(\boldsymbol{a}^*)^\top \sigma(\mathbf{Z}\boldsymbol{w})\right\|_{\psi_2} \lesssim \|\boldsymbol{u}\|\|\boldsymbol{a}^*\|.$$

Invoking Lemma 8, we prove conclusion (1) immediately.

Recall the definitions $\mathbf{P} = \frac{1}{\|\boldsymbol{v}\|}(\mathbf{I} - \boldsymbol{w}\boldsymbol{w}^\top)$. Notice that, for any fixed vector $\boldsymbol{u} \in \mathbb{R}^d$,

$$
\begin{aligned}
\boldsymbol{u}^\top \mathbf{P} \boldsymbol{\xi}_{(1)} &= \left(\sum_{i=1}^k a_i^* \sigma(\mathbf{Z}_i^\top \boldsymbol{w})\right)\left(\sum_{i=1}^k a_i \boldsymbol{u}^\top \mathbf{P} \mathbf{Z}_i \mathbb{1}\left\{\mathbf{Z}_i^\top \boldsymbol{w} \geq 0\right\}\right) \\
&\quad - \mathbb{E}\left[\left(\sum_{i=1}^k a_i^* \sigma(\mathbf{Z}_i^\top \boldsymbol{w})\right)\left(\sum_{i=1}^k a_i \boldsymbol{u}^\top \mathbf{P} \mathbf{Z}_i \mathbb{1}\left\{\mathbf{Z}_i^\top \boldsymbol{w} \geq 0\right\}\right)\right].
\end{aligned}
$$

It follows that

$$
\begin{aligned}
\left\|\boldsymbol{u}^\top \mathbf{P} \boldsymbol{\xi}_{(1)}\right\|_{\psi_1}^2 &\lesssim \left\|\left(\sum_{i=1}^k a_i^* \sigma(\mathbf{Z}_i^\top \boldsymbol{w})\right)\left(\sum_{i=1}^k a_i \boldsymbol{u}^\top \mathbf{P} \mathbf{Z}_i \mathbb{1}\left\{\mathbf{Z}_i^\top \boldsymbol{w} \geq 0\right\}\right)\right\|_{\psi_1}^2 \\
&\lesssim \left\|\sum_{i=1}^k a_i^* \sigma(\mathbf{Z}_i^\top \boldsymbol{w})\right\|_{\psi_2}^2 \left\|\sum_{i=1}^k a_i \boldsymbol{u}^\top \mathbf{P} \mathbf{Z}_i \mathbb{1}\left\{\mathbf{Z}_i^\top \boldsymbol{w} \geq 0\right\}\right\|_{\psi_2}^2 \\
&\lesssim \|\boldsymbol{a}^*\|^2 \|\boldsymbol{w}\|^2 \|\boldsymbol{a}\|^2 \|\mathbf{P}\boldsymbol{u}\|^2 \\
&\leq \|\boldsymbol{a}^*\|^2 \|\boldsymbol{a}\|^2 / \|\boldsymbol{v}\|^2,
\end{aligned}
$$

where we used $\|\boldsymbol{w}\| = 1$ and $\|\mathbf{P}\| \leq 1/\|\boldsymbol{v}\|$. The counterpart for $\boldsymbol{u}^\top \mathbf{P} \boldsymbol{\xi}_{(2)}$ follows from similar arguments. $\qquad\square$

### A.3.3 Corollaries of Lemma 11.

Let $\{\alpha_l\}_{l \geq 0}$ be a sequence of real numbers. Recall that $\boldsymbol{\xi}_t = \frac{1}{N} \sum_{i=1}^{N} \mathbf{P}_t^i \boldsymbol{\xi}_t^i$ and $\boldsymbol{\epsilon}_t = \sum_{i=1}^{N} \boldsymbol{\epsilon}_t^i / N$, where $\{\boldsymbol{\xi}_t^i\}_{i \in [N]}$ and $\{\boldsymbol{\epsilon}_t^i\}_{i \in [N]}$ are independent random variables given $\mathcal{F}_t$. Applying Lemma 10 and Lemma 11(1), we can directly get the following results

$$\left| \sum_{l=0}^{s-1} \alpha_l \boldsymbol{a}_l^\top \boldsymbol{\epsilon}_l \right| \leq \frac{c}{N} \sqrt{\log(2/\delta) \sum_{l=0}^{s-1} \alpha_l^2 \sum_{i=1}^{N} \|\boldsymbol{a}_l\|^2 (\|\boldsymbol{a}^*\| \vee \|\boldsymbol{a}_l^i\|)^2}$$
$$+ \frac{c\log(2/\delta)}{N} \max_{l \leq s-1, i \in [N]} \left\{ \alpha_l \|\boldsymbol{a}_l\| \left( \|\boldsymbol{a}^*\| \vee \|\boldsymbol{a}_l^i\| \right) \right\}, \tag{A.17}$$

and

$$\left| \sum_{l=0}^{s-1} \alpha_l \mathbf{1}^\top \boldsymbol{\epsilon}_l \right| \leq \frac{c}{N} \sqrt{k \log(2/\delta) \sum_{l=0}^{s-1} \alpha_l^2 (\|\boldsymbol{a}^*\| \vee \|\boldsymbol{a}_l\|)^2}$$
$$+ \frac{c\sqrt{k}\log(2/\delta)}{N} \max_{l \leq s-1, i \in [N]} \left\{ \alpha_l \left( \|\boldsymbol{a}^*\| \vee \|\boldsymbol{a}_l^i\| \right) \right\}. \tag{A.18}$$

Specially, choosing $\boldsymbol{u} = \boldsymbol{e}_j$ for $j \in [k]$, we also have

$$\|\boldsymbol{\epsilon}_s\|^2 \leq k \max_{j \in [k]} |\boldsymbol{e}_j^\top \boldsymbol{\epsilon}_s|^2 \leq \frac{ck\log(k/\delta)}{N} \max_{i \in [N]} \left\{ \left( \|\boldsymbol{a}^*\| \vee \|\boldsymbol{a}_s^i\| \right)^2 \right\}, \tag{A.19}$$

hold with probability at least $1 - \delta$.

Similarly, using Lemma 11(2), we also have

$$\left| \sum_{l=0}^{s-1} \boldsymbol{\xi}_l^\top \boldsymbol{w}^* \right| \leq \frac{c\|\boldsymbol{a}^*\|}{N\|\boldsymbol{v}\|} \sqrt{\log(2/\delta) \sum_{l=0}^{s-1} \sum_{i=1}^{N} \|\boldsymbol{a}_s^i\|^2} + \frac{c\log(2/\delta)}{N\|\boldsymbol{v}\|} \|\boldsymbol{a}^*\| \max_{i \in [N]} \|\boldsymbol{a}_s^i\|, \tag{A.20}$$

and

$$\|\boldsymbol{\xi}_s\|^2 \leq \frac{cd\log(d/\delta)}{N\|\boldsymbol{v}\|} \|\boldsymbol{a}^*\|^2 \max_{i \in [N]} \|\boldsymbol{a}_s^i\|^2, \tag{A.21}$$

hold with probability at least $1 - \delta$.

## B  Bound the Weight's Norm during Training Process

### B.1  The Scale of the Second Layer's Weight

**Lemma 12** (The scale of the second layer's weight). *Suppose the learning rate satisfies that*

$$c \max \left\{ \eta(k \vee I) \log(k/\eta\delta), \sqrt{\frac{\eta k \log(k/\eta\delta)}{N}} \right\} < 1, \tag{B.1}$$

*for a large absolute constant $c$. Then for any $0 \leq t \leq (\eta\|\boldsymbol{a}^*\|^2)^{-2}$, we are guaranteed that*

$$\max_{i \in [N]} \|\boldsymbol{a}_t^i\| \leq K_a := 5\|\boldsymbol{a}^*\|, \quad A_t \geq -cK_a \sqrt{\frac{\eta \log(1/\eta\delta)}{N}},$$

*with probability at least $1 - \delta$.*

*Proof.* Recall the definition $\boldsymbol{a}_t = \frac{1}{N} \sum_{i=1}^{N} \boldsymbol{a}_t^i$, $\boldsymbol{\epsilon}_t = \frac{1}{N} \sum_{i=1}^{N} \boldsymbol{\epsilon}_t^i$ and

$$\nabla_{\boldsymbol{a}} L(\boldsymbol{w}_t^i, \boldsymbol{a}_t^i) = \frac{1}{2\pi} \left( \mathbf{1}\mathbf{1}^\top + (\pi - 1)\mathbf{I} \right) \boldsymbol{a}_t^i - \frac{1}{2\pi} (\mathbf{1}\mathbf{1}^\top + (g(\phi_t^i) - 1)\mathbf{I})\boldsymbol{a}^*.$$

Then notice that

$$\left\| \frac{1}{N} \sum_{i=1}^{N} \nabla_{\boldsymbol{a}} L(\boldsymbol{w}_t^i, \boldsymbol{a}_t^i; \boldsymbol{\xi}_t^i) \right\|^2 \le 2 \left\| \frac{1}{N} \sum_{i=1}^{N} \nabla_{\boldsymbol{a}} L(\boldsymbol{w}_t^i, \boldsymbol{a}_t^i) \right\|^2 + 2 \left\| \frac{1}{N} \sum_{i=1}^{N} \boldsymbol{\epsilon}_t^i \right\|^2$$

$$= 2 \left\| \frac{1}{2\pi} \left\{ (\mathbf{1}\mathbf{1}^\top + (\pi - 1)\mathbf{I})\boldsymbol{a}_t - \frac{1}{N} \sum_{i=1}^{N} (\mathbf{1}\mathbf{1}^\top + (g(\phi_t^i) - 1)\,\mathbf{I})\boldsymbol{a}^* \right\} \right\|^2 + 2 \left\| \boldsymbol{\epsilon}_t \right\|^2$$

$$\le \frac{(k + \pi - 1)^2}{\pi^2} \left( \|\boldsymbol{a}_t\|^2 + \|\boldsymbol{a}^*\|^2 \right) + 2 \left\| \boldsymbol{\epsilon}_t \right\|^2,$$

where the last inequality holds since $\|\mathbf{1}\mathbf{1}^\top + (\pi - 1)\mathbf{I}\| \le \pi + k - 1$ and $g(\phi_t^i) \le \pi$. Denote $A_t = (\mathbf{1}^\top \boldsymbol{a}_t)^2 - (\mathbf{1}^\top \boldsymbol{a}_t)(\mathbf{1}^\top \boldsymbol{a}^*)$. Together with the definition of $\boldsymbol{a}_{t+1}$, we have

$$\|\boldsymbol{a}_{t+1}\|^2 = \left\| \boldsymbol{a}_t - \eta \frac{1}{N} \sum_{i=1}^{N} \nabla_{\boldsymbol{a}} L(\boldsymbol{w}_t^i, \boldsymbol{a}_t^i; \boldsymbol{\xi}_t^i) \right\|^2$$

$$= \|\boldsymbol{a}_t\|^2 - 2\eta \frac{1}{N} \sum_{i \in [N]} \left\langle \nabla_{\boldsymbol{a}} L(\boldsymbol{w}_t^i, \boldsymbol{a}_t^i), \boldsymbol{a}_t \right\rangle - 2\eta \boldsymbol{a}_t^\top \boldsymbol{\epsilon}_t + \eta^2 \left\| \frac{1}{N} \sum_{i=1}^{N} \nabla_{\boldsymbol{a}} L(\boldsymbol{w}_t^i, \boldsymbol{a}_t^i; \boldsymbol{\xi}_t^i) \right\|^2$$

$$= \|\boldsymbol{a}_t\|^2 - \frac{\eta}{\pi} \boldsymbol{a}_t^\top \left( \mathbf{1}\mathbf{1}^\top + (\pi - 1)\mathbf{I} \right) \boldsymbol{a}_t + \frac{\eta}{\pi} \frac{1}{N} \sum_{i=1}^{N} \boldsymbol{a}_t^\top \left( \mathbf{1}\mathbf{1}^\top + (g(\phi_t^i) - 1)\,\mathbf{I} \right) \boldsymbol{a}^*$$

$$\quad - 2\eta \boldsymbol{a}_t^\top \boldsymbol{\epsilon}_t + \eta^2 \left\| \frac{1}{N} \sum_{i=1}^{N} \nabla_{\boldsymbol{a}} L(\boldsymbol{w}_t^i, \boldsymbol{a}_t^i; \boldsymbol{\xi}_t^i) \right\|^2$$

$$\le \left( 1 - \frac{\eta(\pi - 1)}{2\pi} \right) \|\boldsymbol{a}_t\|^2 - \frac{\eta}{\pi} A_t + \frac{\eta(\pi - 1)}{2\pi} \|\boldsymbol{a}_t\| \|\boldsymbol{a}^*\| - 2\eta \boldsymbol{a}_t^\top \boldsymbol{\epsilon}_t + \eta^2 \left\| \frac{1}{N} \sum_{i=1}^{N} \nabla_{\boldsymbol{a}} L(\boldsymbol{w}_t^i, \boldsymbol{a}_t^i; \boldsymbol{\xi}_t^i) \right\|^2$$

$$\le \left( 1 - \frac{\eta(\pi - 1)}{2\pi} \right)^{t+1} \|\boldsymbol{a}_0\|^2 + \frac{\eta(\pi - 1)}{2\pi} \sum_{s=0}^{t} \left( 1 - \frac{\eta(\pi - 1)}{2\pi} \right)^{t-s} \|\boldsymbol{a}_s\| \|\boldsymbol{a}^*\|$$

$$\quad - \frac{\eta}{\pi} \sum_{s=0}^{t} \left( 1 - \frac{\eta(\pi - 1)}{2\pi} \right)^{t-s} A_s + 2\eta \left| \sum_{s=0}^{t} \left( 1 - \frac{\eta(\pi - 1)}{2\pi} \right)^{t-s} \boldsymbol{a}_s^\top \boldsymbol{\epsilon}_s \right|$$

$$\quad + 2\eta^2 \sum_{s=0}^{t} \left( 1 - \frac{\eta(\pi - 1)}{2\pi} \right)^{t-s} \left( \frac{(k + \pi - 1)^2}{\pi^2} \left( \|\boldsymbol{a}_s\|^2 + \|\boldsymbol{a}^*\|^2 \right) + \|\boldsymbol{\epsilon}_s\|^2 \right).$$

$$\tag{B.2}$$

Denote the event

$$\mathcal{E}_t = \left\{ \forall s \le t : \max_{i \in [N]} \|\boldsymbol{a}_s^i\| \le 5 \|\boldsymbol{a}^*\| \right\}.$$

Let $\alpha_l = \left( 1 - \eta \frac{\pi - 1}{2\pi} \right)^l$, then it holds that $\sum_{l=0}^{s-1} \alpha_l^2 \le \sum_{l=0}^{s-1} \alpha_l \le \frac{2\pi}{\eta(\pi - 1)}$. Using concentration (A.17), we have

$$\mathbb{1}_{\mathcal{E}_t} \eta \left| \sum_{s=0}^{t} \alpha_s \boldsymbol{a}_s^\top \boldsymbol{\epsilon}_s \right| \le \mathbb{1}_{\mathcal{E}_t} \eta c \sqrt{ \frac{\log(1/\delta)}{N} \sum_{s=0}^{t} \alpha_s \|\boldsymbol{a}_s\|^2 (\|\boldsymbol{a}^*\| \vee \max_i \|\boldsymbol{a}_s^i\|)^2 }$$

$$\quad + \mathbb{1}_{\mathcal{E}_t} c \frac{\log(1/\delta)}{N} \max_{s \le t} \alpha_s \|\boldsymbol{a}_s\| (\|\boldsymbol{a}^*\| \vee \max_i \|\boldsymbol{a}_s^i\|)$$

$$\le \mathbb{1}_{\mathcal{E}_t} \cdot 10c \|\boldsymbol{a}^*\|^2 \sqrt{ \frac{\eta \log(1/\delta)}{N} }, \tag{B.3}$$

holds with probability at least $1 - \delta$. Applying the concentration inequality (A.19) gives

$$\mathbb{1}_{\mathcal{E}_t} \eta^2 \sum_{s=0}^{t} \alpha_s \|\boldsymbol{\epsilon}_t\|^2 \le \mathbb{1}_{\mathcal{E}_t} \eta^2 \sum_{s=0}^{t} \alpha_s \frac{ck \log(tk/\delta)}{N} (\|\boldsymbol{a}^*\| \vee \max_i \|\boldsymbol{a}_s^i\|)^2$$

$$\leq \mathbb{1}_{\mathcal{E}_t} \cdot \frac{50ck\eta \log(tk/\delta)}{N}\|\boldsymbol{a}^*\|^2, \tag{B.4}$$

holds with probability at least $1 - \delta$. In addition, from (A.13), we know

$$\mathbb{1}_{\mathcal{E}_t} A_s = \mathbb{1}_{\mathcal{E}_t} \left\{ \left(1 - \eta\frac{\pi+k-1}{2\pi}\right)^s A_0 + \frac{\eta(\mathbf{1}^\top \boldsymbol{a}^*)^2}{2\pi}\sum_{l=0}^{s-1}\left(1 - \eta\frac{\pi+k-1}{2\pi}\right)^{s-1-l}\left(\pi - \frac{1}{N}\sum_{i=1}^N g(\phi_l^i)\right) \right.$$

$$\left. + \eta\sum_{l=0}^{s-1}\left(1 - \eta\frac{\pi+k-1}{2\pi}\right)^{s-1-l}(\mathbf{1}^\top \boldsymbol{a}^*)(\mathbf{1}^\top \boldsymbol{\epsilon}_l) \right\}$$

$$\geq \mathbb{1}_{\mathcal{E}_t}\left\{ -\eta\left|\sum_{l=0}^{s-1}\left(1 - \eta\frac{\pi+k-1}{2\pi}\right)^{s-1-l}(\mathbf{1}^\top \boldsymbol{a}^*)(\mathbf{1}^\top \boldsymbol{\epsilon}_l)\right| \right\}$$

$$\geq \mathbb{1}_{\mathcal{E}_t}\left\{ -c\sqrt{\frac{\eta k\log(1/\delta)}{N}\sum_{l=0}^{s-1}\left(1 - \eta\frac{\pi+k-1}{2\pi}\right)^{s-1-l}(\|\boldsymbol{a}^*\| \vee \max_i \|\boldsymbol{a}_l^i\|)^2} \right\}$$

$$\geq \mathbb{1}_{\mathcal{E}_t} \cdot -5c\|\boldsymbol{a}^*\|^2\sqrt{\frac{\eta \log(1/\delta)}{N}}, \tag{B.5}$$

holds with probability at least $1-\delta$. Plugging (B.3), (B.4) and (B.5) into (B.2), together with $t \leq \eta^{-2}$, we can get

$$\|\boldsymbol{a}_{t+1}\|^2 \mathbb{1}_{\mathcal{E}_t} \leq \|\boldsymbol{a}_0\|^2 + 5\|\boldsymbol{a}^*\|^2 + 5c\|\boldsymbol{a}^*\|^2\sqrt{\frac{\eta k\log(1/\eta\delta)}{N}} + 10c\|\boldsymbol{a}^*\|^2\sqrt{\frac{\eta\log(1/\eta\delta)}{N}}$$

$$+ 4\eta\frac{\pi+k-1}{\pi}\left(25\|\boldsymbol{a}^*\|^2 + \|\boldsymbol{a}^*\|^2\right) + \frac{50c\eta k\log(t/\delta)}{N}\|\boldsymbol{a}^*\|^2$$

$$\leq 9\|\boldsymbol{a}^*\|^2, \tag{B.6}$$

where the last inequality holds due to our assumption (B.1) and $\|\boldsymbol{a}_0\| \leq \|\boldsymbol{a}^*\|$. By setting $T = t+1$ and $K_a = 5\|\boldsymbol{a}^*\|$ in conclusion (1) of Lemma 15, we have

$$\mathbb{1}_{\mathcal{E}_t}\|\boldsymbol{a}_{t+1}^i - \boldsymbol{a}_{t+1}\| \leq \mathbb{1}_{\mathcal{E}_t}\frac{1}{N}\sum_{j=1}^N \|\boldsymbol{a}_{t+1}^i - \boldsymbol{a}_{t+1}^j\|$$

$$\leq \mathbb{1}_{\mathcal{E}_t} \cdot 10\|\boldsymbol{a}^*\|\left(2\eta I + c\eta\sqrt{kI\log(k/\delta)}\right)$$

$$\leq \mathbb{1}_{\mathcal{E}_t} \cdot 4\|\boldsymbol{a}^*\|, \tag{B.7}$$

where we used the assumption (B.1). In conjunction with (B.6), we can verify that

$$\mathbb{P}\left(\mathcal{E}_t \cap \{\|\boldsymbol{a}_{t+1}\| > 5\|\boldsymbol{a}^*\|\}\right) \leq \mathbb{P}\left(\text{One of (B.3), (B.4), (B.5), and (B.7) does not hold}\right)$$

$$\leq 4\delta.$$

Therefore, we have

$$\mathbb{P}\left(\mathcal{E}_{t+1}\right) = \mathbb{P}\left(\mathcal{E}_t \cap \{\|\boldsymbol{a}_{t+1}\| \leq 5\|\boldsymbol{a}^*\|\}\right)$$

$$\geq \mathbb{P}\left(\mathcal{E}_t\right) - \mathbb{P}\left(\mathcal{E}_t \cap \{\|\boldsymbol{a}_{t+1}\| > 5\|\boldsymbol{a}^*\|\}\right)$$

$$\geq \mathbb{P}\left(\mathcal{E}_t\right) - 4\delta.$$

Hence by induction, we can verify that $\mathbb{P}\left(\mathcal{E}_t\right) \geq 1 - 4t\delta$ for any $t \leq (\eta K_a^2)^{-2}$. By adjusting the level of $\delta$ and $\mathbb{P}(\mathcal{E}_0) = 1$ due to $\|\boldsymbol{a}_0\| \leq \|\boldsymbol{a}^*\|$, we can finish the proof. $\qquad\square$

## B.2 The Scale of the First Layer's weight

**Lemma 13** (The scale of the first layer's weight). *Under the conditions of Lemma 12. Let $K_a = 5\|\boldsymbol{a}^*\|$, we assume the learning rate satisfies*

$$c\eta\left(1 + \sqrt{\frac{I}{d}}\right)\sqrt{dI\log(Nd/\delta)}K_a^2 < 1,$$

*for a large absolute constant c. Then for any $0 \leq t \leq (800c\eta^2(\sqrt{Id} + I)\sqrt{\log(Nd/\delta)}K_a^4)^{-1}$, we are guaranteed that $1/2 \leq \|\boldsymbol{v}_t\| \leq 3$ with probability at least $1 - \delta$.*

*Proof.* In this proof, we suppose the event Lemma 12 holds and hide the indicator function everywhere. To facilitate the technical presentation, denote $\tilde{\boldsymbol{v}}_t^i = \boldsymbol{v}_t^i - \eta \mathbf{P}_t^i \nabla_{\boldsymbol{w}} L(\boldsymbol{w}_t^i, \boldsymbol{a}_t^i)$. Define the following event

$$\mathcal{U}_t = \left\{ \forall s \leq t \leq (800 c \eta^2 (\sqrt{Id} + I) \sqrt{\log(Nd/\delta)} K_a^4)^{-1} : 1/2 \leq \|\boldsymbol{v}_s\| \leq 3 \right\}.$$

*Step 1: show $\|\boldsymbol{v}_t\| \geq 1/2$ under the event $\mathcal{U}_{t-1}$.* From $\boldsymbol{v}_t = \frac{1}{N} \sum_{i=1}^N \boldsymbol{v}_t^i$ and $\mathbf{P}_{t-1} \boldsymbol{v}_{t-1} = \mathbf{0}$, we have

$$\|\boldsymbol{v}_t\|^2 = \left\| \boldsymbol{v}_{t-1} - \eta \frac{1}{N} \sum_{i=1}^N \mathbf{P}_{t-1}^i \nabla_{\boldsymbol{w}} L(\boldsymbol{w}_{t-1}^i, \boldsymbol{a}_{t-1}^i) \right\|^2 - \frac{2\eta}{N} \sum_{i=1}^N \langle \tilde{\boldsymbol{v}}_{t-1}^i, \boldsymbol{\xi}_{t-1} \rangle + \eta^2 \|\boldsymbol{\xi}_{t-1}\|^2$$

$$= \left\| \boldsymbol{v}_{t-1} - \eta \mathbf{P}_{t-1} \frac{1}{N} \sum_{i=1}^N \nabla_{\boldsymbol{w}} L(\boldsymbol{w}_{t-1}^i, \boldsymbol{a}_{t-1}^i) \right\|^2 - \frac{2\eta}{N} \sum_{i=1}^N \langle \tilde{\boldsymbol{v}}_{t-1}^i, \boldsymbol{\xi}_{t-1} \rangle + \eta^2 \|\boldsymbol{\xi}_{t-1}\|^2$$

$$+ 2\eta \left\langle \boldsymbol{v}_{t-1} - \eta \mathbf{P}_{t-1} \frac{1}{N} \sum_{i=1}^N \nabla_{\boldsymbol{w}} L(\boldsymbol{w}_{t-1}^i, \boldsymbol{a}_{t-1}^i), \frac{1}{N} \sum_{i=1}^N (\mathbf{P}_{t-1} - \mathbf{P}_{t-1}^i) \nabla_{\boldsymbol{w}} L(\boldsymbol{w}_{t-1}^i, \boldsymbol{a}_{t-1}^i) \right\rangle$$

$$+ \eta^2 \left\| \frac{1}{N} \sum_{i=1}^N (\mathbf{P}_{t-1} - \mathbf{P}_{t-1}^i) \nabla_{\boldsymbol{w}} L(\boldsymbol{w}_{t-1}^i, \boldsymbol{a}_{t-1}^i) \right\|^2$$

$$\geq \|\boldsymbol{v}_{t-1}\|^2 - \frac{2\eta}{N} \sum_{i=1}^N \langle \tilde{\boldsymbol{v}}_{t-1}^i, \boldsymbol{\xi}_{t-1} \rangle - 4\eta \|\boldsymbol{v}_{t-1}\|^2 \frac{1}{N} \sum_{i=1}^N \|\mathbf{P}_{t-1} - \mathbf{P}_{t-1}^i\| \|\nabla_{\boldsymbol{w}} L(\boldsymbol{w}_{t-1}^i, \boldsymbol{a}_{t-1}^i)\|$$

$$- 2\eta^2 \frac{1}{N} \sum_{i=1}^N \|\mathbf{P}_{t-1}\| \|\nabla_{\boldsymbol{w}} L(\boldsymbol{w}_{t-1}^i, \boldsymbol{a}_{t-1}^i)\| \frac{1}{N} \sum_{i=1}^N \|\mathbf{P}_{t-1} - \mathbf{P}_{t-1}^i\| \|\nabla_{\boldsymbol{w}} L(\boldsymbol{w}_{t-1}^i, \boldsymbol{a}_{t-1}^i)\|,$$

(B.8)

where the inequality follows from dropping positive terms and the inductive assumption $\|\boldsymbol{v}_{t-1}\| \geq 1/2$. With probability at least $1 - \delta$, it holds that

$$\mathbb{1}_{\mathcal{U}_{t-1}} \|\mathbf{P}_{t-1}^i - \mathbf{P}_{t-1}\| = \mathbb{1}_{\mathcal{U}_{t-1}} \left( \left\| \frac{1}{\|\boldsymbol{v}_{t-1}^i\|} (\mathbf{I} - \boldsymbol{w}_{t-1}^i (\boldsymbol{w}_{t-1}^i)^\top) - \frac{1}{\|\boldsymbol{v}_{t-1}\|} (\mathbf{I} - \boldsymbol{w}_{t-1} \boldsymbol{w}_{t-1}^\top) \right\| \right)$$

$$\leq \mathbb{1}_{\mathcal{U}_{t-1}} \left\{ \left| \frac{1}{\|\boldsymbol{v}_{t-1}^i\|} - \frac{1}{\|\boldsymbol{v}_{t-1}\|} \right| + \frac{1}{\|\boldsymbol{v}_{t-1}\|} \|\boldsymbol{w}_{t-1}^i (\boldsymbol{w}_{t-1}^i)^\top - \boldsymbol{w}_{t-1} \boldsymbol{w}_{t-1}^\top\| \right\}$$

$$\overset{(i)}{\leq} \mathbb{1}_{\mathcal{U}_{t-1}} \left\{ \left| \frac{1}{\|\boldsymbol{v}_{t-1}^i\|} - \frac{1}{\|\boldsymbol{v}_{t-1}\|} \right| + \frac{2\|\boldsymbol{w}_{t-1}^i - \boldsymbol{w}_{t-1}\|}{\|\boldsymbol{v}_{t-1}\|} \right\}$$

$$\overset{(ii)}{\leq} \mathbb{1}_{\mathcal{U}_{t-1}} \left\{ \left| \frac{1}{\|\boldsymbol{v}_{t-1}^i\|} - \frac{1}{\|\boldsymbol{v}_{t-1}\|} \right| + 4 \left\| \frac{\boldsymbol{v}_{t-1}^i}{\|\boldsymbol{v}_{t-1}^i\|} - \frac{\boldsymbol{v}_{t-1}}{\|\boldsymbol{v}_{t-1}\|} \right\| \right\}$$

$$\leq \mathbb{1}_{\mathcal{U}_{t-1}} \left\{ \left| \frac{1}{\|\boldsymbol{v}_{t-1}^i\|} - \frac{1}{\|\boldsymbol{v}_{t-1}\|} \right| + \frac{4}{\|\boldsymbol{v}_{t-1}^i\|} \|\boldsymbol{v}_{t-1}^i - \boldsymbol{v}_{t-1}\| + 4\|\boldsymbol{v}_{t-1}\| \left| \frac{1}{\|\boldsymbol{v}_{t-1}^i\|} - \frac{1}{\|\boldsymbol{v}_{t-1}\|} \right| \right\}$$

$$\overset{(iii)}{\leq} \mathbb{1}_{\mathcal{U}_{t-1}} \left\{ \frac{2\|\boldsymbol{v}_{t-1}^i - \boldsymbol{v}_{t-1}\|}{\|\boldsymbol{v}_{t-1}^i\|} + \frac{8\|\boldsymbol{v}_{t-1}^i - \boldsymbol{v}_{t-1}\|}{\|\boldsymbol{v}_{t-1}^i\|} \right\}$$

$$\overset{(iv)}{\leq} \mathbb{1}_{\mathcal{U}_{t-1}} \left\{ \frac{10 c \eta (\sqrt{Id} + I) \sqrt{\log(Nd/\delta)} K_a^2}{1/2 - 6 c \eta (\sqrt{Id} + I) \sqrt{\log(Nd/\delta)} K_a^2} \right\}$$

$$\overset{(v)}{\leq} \mathbb{1}_{\mathcal{U}_{t-1}} \cdot 40 c \eta (\sqrt{Id} + I) \sqrt{\log(Nd/\delta)} K_a^2,$$

(B.9)

where $(i)$ holds due to $\|\boldsymbol{w}_{t-1}^i\| = \|\boldsymbol{w}_{t-1}\| = 1$; $(ii)$ and $(iii)$ hold due to the inductive assumption $\|\boldsymbol{v}_{t-1}\| \geq 1/2$; $(iv)$ follows from the conclusion (1) in Lemma 14; and $(v)$ is true because $6 c \eta (\sqrt{Id} + I) \sqrt{\log(Nd/\delta)} \leq 1/4$. Substituting (B.9) into (B.8), with probability at least $1 - 2\delta$, we have

$$\mathbb{1}_{\mathcal{U}_{t-1}} \|\boldsymbol{v}_t\|^2 \geq \mathbb{1}_{\mathcal{U}_{t-1}} \left\{ \|\boldsymbol{v}_{t-1}\|^2 - \frac{2\eta}{N} \sum_{i=1}^N \langle \tilde{\boldsymbol{v}}_{t-1}^i, \boldsymbol{\xi}_{t-1} \rangle \right.$$

$$
- 160c\eta^2(\sqrt{Id}+I)\sqrt{\log(Nd/\delta)}K_a^2 \cdot \|\boldsymbol{v}_{t-1}\|^2 \cdot \frac{1}{N}\sum_{i=1}^{N}\|\nabla_{\boldsymbol{w}}L(\boldsymbol{w}_{t-1}^i,\boldsymbol{a}_{t-1}^i)\|
$$

$$
- 80\eta^3(\sqrt{Id}+I)\sqrt{\log(Nd/\delta)}K_a^2 \cdot \|\mathbf{P}_{t-1}\| \cdot \left(\frac{1}{N}\sum_{i=1}^{N}\|\nabla_{\boldsymbol{w}}L(\boldsymbol{w}_{t-1}^i,\boldsymbol{a}_{t-1}^i)\|\right)^2 \Bigg\}
$$

$$
\overset{(i)}{\geq} \mathbb{1}_{\mathcal{U}_{t-1}}\left\{ \left(1 - 800c\eta^2(\sqrt{Id}+I)\sqrt{\log(Nd/\delta)}K_a^4\right)\|\boldsymbol{v}_{t-1}\|^2 - \frac{2\eta}{N}\sum_{i=1}^{N}\langle\tilde{\boldsymbol{v}}_{t-1}^i,\boldsymbol{\xi}_{t-1}\rangle \right\}
$$

$$
- 1000\eta^3(\sqrt{Id}+I)\sqrt{\log(Nd/\delta)}K_a^6
$$

$$
\overset{(ii)}{\geq} \mathbb{1}_{\mathcal{U}_{t-1}}\Bigg\{ \left(1 - 800c\eta^2(\sqrt{Id}+I)\sqrt{\log(Nd/\delta)}K_a^4\right)^t\|\boldsymbol{v}_0\|
$$

$$
- 2\eta\left|\sum_{s=0}^{t-1}\left(1 - 800c\eta^2(\sqrt{Id}+I)\sqrt{\log(Nd/\delta)}K_a^4\right)^{t-s}\left\langle\frac{1}{N}\sum_{i=1}^{N}\tilde{\boldsymbol{v}}_s^i,\boldsymbol{\xi}_s\right\rangle\right|
$$

$$
- 1000\eta^3(\sqrt{Id}+I)\sqrt{\log(Nd/\delta)}K_a^6\sum_{s=0}^{t-1}\left(1 - 800c\eta^2(\sqrt{Id}+I)\sqrt{\log(Nd/\delta)}K_a^4\right)^{t-s} \Bigg\}
$$

$$
\overset{(iii)}{\geq} \mathbb{1}_{\mathcal{U}_{t-1}}\left\{0.36 - \frac{5}{4}\eta K_a^2 - 2\eta\left|\sum_{s=0}^{t-1}\left(1 - 800c\eta^2(\sqrt{Id}+I)\sqrt{\log(Nd/\delta)}K_a^4\right)^{t-s}\left\langle\frac{1}{N}\sum_{i=1}^{N}\tilde{\boldsymbol{v}}_s^i,\boldsymbol{\xi}_s\right\rangle\right|\right\},
$$
(B.10)

where $(i)$ holds because $\|\nabla_{\boldsymbol{w}}L(\boldsymbol{w}_t^i,\boldsymbol{v}_t^i)\| \leq |(\boldsymbol{a}_t^i)^\top\boldsymbol{a}^*| \leq 5K_a^2$ by using Lemma 12 and $\|\mathbf{P}_{t-1}\| \leq \frac{1}{\|\boldsymbol{v}_{t-1}\|} \leq 2$ holds under the event $\mathcal{U}_{t-1}$; $(ii)$ holds due to $\|\boldsymbol{v}_0\| = 1$; and $(iii)$ follows from $t \leq (800c\eta^2(d\vee I)K_a^4\sqrt{\log(Nd/\delta)})^{-1}$ such that

$$
\left(1 - 800c\eta^2(\sqrt{Id}+I)\sqrt{\log(Nd/\delta)}K_a^4\right)^t
$$

$$
\geq \left(1 - 800c\eta^2(\sqrt{Id}+I)\sqrt{\log(Nd/\delta)}K_a^4\right)^{\frac{1}{800c\eta^2(\sqrt{Id}+I)\sqrt{\log(Nd/\delta)}K_a^4}}
$$

$$
\geq (1 - 0.01)^{100} \geq 0.36.
$$

In fact, the bound above holds because $800c\eta^2(\sqrt{Id}+I)\sqrt{\log(Nd/\delta)}K_a^4 \leq 0.01$ and $(1-x)^{\frac{1}{x}}$ is decreasing in $(0,1)$. According to the definition of $\tilde{\boldsymbol{v}}_s^i$ and Lemma 12, we know

$$
\mathbb{1}_{\mathcal{U}_{t-1}}\left\|\frac{1}{N}\sum_{i=1}^{N}\tilde{\boldsymbol{v}}_s^i\right\| = \mathbb{1}_{\mathcal{U}_{t-1}}\left\|\boldsymbol{v}_s - \eta\frac{1}{N}\sum_{i=1}^{N}\mathbf{P}_s^i\nabla_{\boldsymbol{w}}L(\boldsymbol{w}_s^i,\boldsymbol{a}_s^i)\right\| \leq 3 + 5\eta K_a^2.
$$
(B.11)

Recall that $\boldsymbol{\xi}_s = \frac{1}{N}\sum_{i=1}^{N}\mathbf{P}_s^i\boldsymbol{\xi}_s^i$. Invoking Lemmas 11 and 10, we have

$$
\mathbb{1}_{\mathcal{U}_{t-1}}\eta\left|\sum_{s=0}^{t-1}\left(1 - 800\eta^2(\sqrt{Id}+I)\sqrt{\log(Nd/\delta)}K_a^4\right)^{t-1-s}\left\langle\frac{1}{N}\sum_{i=1}^{N}\tilde{\boldsymbol{v}}_s^i,\boldsymbol{\xi}_s\right\rangle\right|
$$

$$
\leq \mathbb{1}_{\mathcal{U}_{t-1}}\Bigg\{\frac{c\eta K_a^2\left(3+5\eta K_a^2\right)}{\sqrt{N}}\sqrt{\log(2/\delta)\sum_{s=0}^{t-1}\left(1 - 800\eta^2(\sqrt{Id}+I)\sqrt{\log(Nd/\delta)}K_a^4\right)^{2(t-1-s)}}
$$

$$
+ \frac{c\eta K_a^2\log(2/\delta)}{N}\Bigg\}
$$

$$
\leq \mathbb{1}_{\mathcal{U}_{t-1}}\left\{\frac{c\left(3+5\eta K_a^2\right)}{20\sqrt{I}\log(Nd/\delta)}\sqrt{\frac{\log(2/\delta)}{N}} + \frac{c\eta K_a^2\log(2/\delta)}{N}\right\}
$$

$$
\leq \mathbb{1}_{\mathcal{U}_{t-1}}\left\{\frac{c\left(3+5\eta K_a^2\right)}{20\sqrt{NI}} + \frac{c\eta K_a^2\log(2/\delta)}{N}\right\},
$$
(B.12)

holds with probability at least $1 - \delta$. Plugging (B.12) into (B.10), we get

$$\|\boldsymbol{v}_t\|^2 \mathbb{1}_{\mathcal{U}_{t-1}} \geq \mathbb{1}_{\mathcal{U}_{t-1}} \left\{ 0.36 - \frac{5}{4}\eta K_a^2 - 2\left( \frac{c\left(3 + 5\eta K_a^2\right)}{20\sqrt{I\log(Nd/\delta)}}\sqrt{\frac{\log(2/\delta)}{N}} + \frac{c\eta K_a^2 \log(2/\delta)}{N} \right) \right\}$$

$$\geq \mathbb{1}_{\mathcal{U}_{t-1}} \left\{ 0.36 - \frac{1}{16} - \frac{1}{16} \right\}$$

$$\geq \mathbb{1}_{\mathcal{U}_{t-1}} \cdot \frac{1}{4}, \tag{B.13}$$

where the first inequality holds since $\sqrt{Id\log(Nd/\delta)} \geq c$.

*Step 2: show $\|\boldsymbol{v}_t\| \leq 3$ under the event $\mathcal{U}_{t-1}$.* From the update rule of $\boldsymbol{v}_t^i$ and $\boldsymbol{v}_t = \frac{1}{N}\sum_{i=1}^N \boldsymbol{v}_t^i$, with probability at least $1 - \delta$, we have

$$\mathbb{1}_{\mathcal{U}_{t-1}}\|\boldsymbol{v}_t\|^2 = \mathbb{1}_{\mathcal{U}_{t-1}} \left\{ \left\| \boldsymbol{v}_{t-1} - \eta \mathbf{P}_{t-1}\frac{1}{N}\sum_{i=1}^N \nabla_{\boldsymbol{w}}L(\boldsymbol{w}_{t-1}^i, \boldsymbol{a}_{t-1}^i) \right\|^2 - \frac{2\eta}{N}\sum_{i=1}^N \langle \tilde{\boldsymbol{v}}_{t-1}^i, \boldsymbol{\xi}_{t-1} \rangle + \eta^2\|\boldsymbol{\xi}_{t-1}\|^2 \right.$$

$$+ 2\eta \left\langle \boldsymbol{v}_{t-1} - \eta\mathbf{P}_{t-1}\frac{1}{N}\sum_{i=1}^N \nabla_{\boldsymbol{w}}L(\boldsymbol{w}_{t-1}^i, \boldsymbol{a}_{t-1}^i), \frac{1}{N}\sum_{i=1}^N(\mathbf{P}_{t-1} - \mathbf{P}_{t-1}^i)\nabla_{\boldsymbol{w}}L(\boldsymbol{w}_{t-1}^i, \boldsymbol{a}_{t-1}^i) \right\rangle$$

$$\left. + \eta^2 \left\| \frac{1}{N}\sum_{i=1}^N(\mathbf{P}_{t-1} - \mathbf{P}_{t-1}^i)\nabla_{\boldsymbol{w}}L(\boldsymbol{w}_{t-1}^i, \boldsymbol{a}_{t-1}^i) \right\|^2 \right\}$$

$$\overset{(i)}{\leq} \mathbb{1}_{\mathcal{U}_{t-1}} \left\{ \|\boldsymbol{v}_{t-1}\|^2 + \eta^2 \left\| \frac{1}{N}\sum_{i=1}^N \mathbf{P}_{t-1}\nabla_{\boldsymbol{w}}L(\boldsymbol{w}_{t-1}^i, \boldsymbol{a}_{t-1}^i) \right\|^2 \right.$$

$$+ \frac{2\eta}{3}\|\boldsymbol{v}_{t-1}\|^2 \frac{1}{N}\sum_{i=1}^N \|\mathbf{P}_{t-1} - \mathbf{P}_{t-1}^i\|\|\nabla_{\boldsymbol{w}}L(\boldsymbol{w}_{t-1}^i, \boldsymbol{a}_{t-1}^i)\|$$

$$+ 2\eta^2 \left\| \frac{1}{N}\sum_{i=1}^N \mathbf{P}_{t-1}\nabla_{\boldsymbol{w}}L(\boldsymbol{w}_{t-1}^i, \boldsymbol{a}_{t-1}^i) \right\| \left\| \frac{1}{N}\sum_{i=1}^N(\mathbf{P}_{t-1} - \mathbf{P}_{t-1}^i)\nabla_{\boldsymbol{w}}L(\boldsymbol{w}_{t-1}^i, \boldsymbol{a}_{t-1}^i) \right\|$$

$$\left. + \eta^2 \left\| \frac{1}{N}\sum_{i=1}^N(\mathbf{P}_{t-1} - \mathbf{P}_{t-1}^i)\nabla_{\boldsymbol{w}}L(\boldsymbol{w}_{t-1}^i, \boldsymbol{a}_{t-1}^i) \right\|^2 - \frac{2\eta}{N}\sum_{i=1}^N \langle \tilde{\boldsymbol{v}}_{t-1}^i, \boldsymbol{\xi}_{t-1} \rangle + \eta^2\|\boldsymbol{\xi}_{t-1}\|^2 \right\}$$

$$\overset{(ii)}{\leq} \mathbb{1}_{\mathcal{U}_{t-1}} \left\{ \left( 1 + 5\eta K_a^2 \max_i \|\mathbf{P}_{t-1}^i - \mathbf{P}_{t-1}\| \right)\|\boldsymbol{v}_{t-1}\|^2 + 25\eta^2 K_a^4 + 25\eta^2 K_a^4 \max_i \|\mathbf{P}_{t-1}^i - \mathbf{P}_{t-1}\| \right.$$

$$\left. + \frac{25\eta^2 K_a^4}{4}\max_i \|\mathbf{P}_{t-1}^i - \mathbf{P}_{t-1}\|^2 - \frac{2\eta}{N}\sum_{i=1}^N \langle \tilde{\boldsymbol{v}}_{t-1}^i, \boldsymbol{\xi}_{t-1} \rangle + \eta^2\|\boldsymbol{\xi}_{t-1}\|^2 \right\}$$

$$\overset{(iii)}{\leq} \mathbb{1}_{\mathcal{U}_{t-1}} \left\{ \left( 1 + 200c\eta^2(\sqrt{Id} + I)\sqrt{\log(Nd/\delta)}K_a^4 \right)\|\boldsymbol{v}_{t-1}\|^2 + 25\eta^2 K_a^4 \right.$$

$$\left. + 1000(1 + 10\eta K_a^2)\eta^3 K_a^6(\sqrt{Id} + I)\sqrt{\log(Nd/\delta)} - \frac{2\eta}{N}\sum_{i=1}^N \langle \tilde{\boldsymbol{v}}_{t-1}^i, \boldsymbol{\xi}_{t-1} \rangle + \eta^2\|\boldsymbol{\xi}_{t-1}\|^2 \right\}$$

$$\leq \mathbb{1}_{\mathcal{U}_{t-1}} \left\{ \left( 1 + 200c\eta^2(\sqrt{Id} + I)\sqrt{\log(Nd/\delta)}K_a^4 \right)^t \|\boldsymbol{v}_0\|^2 \right.$$

$$+ 2000\eta^3(\sqrt{Id} + I)\sqrt{\log(Nd/\delta)}K_a^6 \sum_{s=0}^{t-1} \left( 1 + 200c\eta^2(\sqrt{Id} + I)\sqrt{\log(Nd/\delta)}K_a^4 \right)^{t-s}$$

$$\left. + 2\eta \left| \sum_{s=0}^{t-1} \left( 1 + 200c\eta^2(\sqrt{Id} + I)\sqrt{\log(Nd/\delta)}K_a^4 \right)^{t-1-s} \left\langle \frac{1}{N}\sum_{i=1}^N \tilde{\boldsymbol{v}}_s^i, \boldsymbol{\xi}_s \right\rangle \right| \right.$$

$$+ \eta^2 \sum_{s=0}^{t-1} \left( 1 + 200c\eta^2(\sqrt{Id} + I)\sqrt{\log(Nd/\delta)}K_a^4 \right)^{t-1-s} \|\boldsymbol{\xi}_s\|^2 \Big\}. \tag{B.14}$$

where $(i)$ comes from $\mathbf{P}_{t-1}\boldsymbol{v}_{t-1} = \mathbf{0}$ and $\|\boldsymbol{v}_{t-1}\| \leq 3$ under $\mathcal{U}_{t-1}$; $(ii)$ comes from $\|\mathbf{P}_{t-1}\| \leq 1/\|\boldsymbol{v}_{t-1}\| \leq 2$ under $\mathcal{U}_{t-1}$; and $(iii)$ holds due to (B.9). With probability at least $1 - \delta$, we have the following concentration

$$\mathbb{1}_{\mathcal{U}_{t-1}}\eta^2 \sum_{s=0}^{t-1} \left( 1 + 200c\eta^2(\sqrt{Id} + I)\sqrt{\log(Nd/\delta)}K_a^4 \right)^{t-1-s} \|\boldsymbol{\xi}_s\|^2$$

$$\leq \mathbb{1}_{\mathcal{U}_{t-1}} \frac{c\eta^2 d\log(1/\eta\delta)K_a^4}{N} \sum_{s=0}^{t-1} \left( 1 + 200c\eta^2(\sqrt{Id} + I)\sqrt{\log(Nd/\delta)}K_a^4 \right)^{t-1-s}$$

$$\leq \mathbb{1}_{\mathcal{U}_{t-1}} \frac{c\eta^2 d\log(1/\eta\delta)K_a^4}{N} \frac{\left( 1 + 400c\eta^2(I \vee d)\sqrt{\log(Nd/\delta)}K_a^4 \right)^t}{400c\eta^2(I \vee d)\sqrt{\log(Nd/\delta)}K_a^4}$$

$$\leq \mathbb{1}_{\mathcal{U}_{t-1}} \frac{ce\log(1/\eta\delta)}{N}, \tag{B.15}$$

where we used the assumption $t \leq (400c\eta^2(d \vee I)\sqrt{\log(Nd/\delta)}K_a^4)^{-1}$. Similar to (B.12), we can also guarantee

$$\mathbb{1}_{\mathcal{U}_{t-1}}\eta \left| \sum_{s=0}^{t-1} \left( 1 + 200c\eta^2(\sqrt{Id} + I)\sqrt{\log(Nd/\delta)}K_a^4 \right)^{t-1-s} \left\langle \frac{1}{N}\sum_{i=1}^{N} \tilde{\boldsymbol{v}}_s^i, \boldsymbol{\xi}_s \right\rangle \right|$$

$$\leq \mathbb{1}_{\mathcal{U}_{t-1}} \Bigg\{ \frac{c\eta K_a^2 \left( 3 + 5\eta K_a^2 \right)}{\sqrt{N}} \sqrt{\log(2/\delta) \sum_{s=0}^{t-1} \left( 1 + 200c\eta^2(\sqrt{Id} + I)\sqrt{\log(Nd/\delta)}K_a^4 \right)^{2(t-1-s)}}$$

$$+ \frac{c\eta K_a^2 \log(2/\delta)}{N} \Bigg\}$$

$$\leq \mathbb{1}_{\mathcal{U}_{t-1}} \left\{ \frac{ce \left( 3 + 5\eta K_a^2 \right)}{\sqrt{(I \vee d)N}} + \frac{c\eta K_a^2 \log(2/\delta)}{N} \right\}, \tag{B.16}$$

Plugging (B.15) and (B.16) into (B.14) gives that

$$\mathbb{1}_{\mathcal{U}_{t-1}}\|\boldsymbol{v}_t\|^2 \leq \mathbb{1}_{\mathcal{U}_{t-1}} \left\{ e + 10e\eta K_a^2 + \frac{2ce\log(1/\eta\delta)}{N} + 2\left( \frac{ce \left( 3 + 5\eta K_a^2 \right)}{\sqrt{(I \vee d)N}} + \frac{c\eta K_a^2 \log(2/\delta)}{N} \right) \right\}$$

$$\leq \mathbb{1}_{\mathcal{U}_{t-1}} \cdot 9,$$

holds with probability at least $1 - 3\delta$. Together with (B.13), we have showed that

$$\mathbb{P}\left( \mathcal{U}_{t-1} \cap \{1/2 \leq \|\boldsymbol{v}_t\| \leq 3\} \right) \geq \mathbb{P}\left( \mathcal{U}_{t-1} \right) - \mathbb{P}\left( \mathcal{U}_{t-1} \cap \{\|\boldsymbol{v}_t\| \leq 1/2\} \right) - \mathbb{P}\left( \mathcal{U}_{t-1} \cap \{\|\boldsymbol{v}_t\| > 3\} \right)$$

$$\geq \mathbb{P}\left( \mathcal{U}_{t-1} \right) - \mathbb{P}\left( \text{One of (B.9), (B.10), (B.13), and Lemma 12 does not hold} \right)$$

$$- \mathbb{P}\left( \text{One of (B.14), (B.15), (B.16), and Lemma 12 does not hold} \right)$$

$$\geq \mathbb{P}\left( \mathcal{U}_{t-1} \right) - 8\delta.$$

Using induction and $\mathbb{P}\left( \mathcal{U}_0 \right) = 1$ due to $\|\boldsymbol{v}_0\| = 1$, we can prove the desired conclusion by adjusting the level of $\delta$. $\qquad\square$

**Remark.** In the following proofs, we use $c$ to denote the absolute constant, which does not depend on any variables of the model or training process. For conciseness, we do not distinguish the specific value of $c$ in some contexts.

## C  Bound the Discrepancy during Training Process

### C.1  The Discrepancy in the First Layer

**Lemma 14** (The discrepancy in the first layer)**.** *Denote $O_{r,T}^{i,j} = \max_{t_r \leq t \leq T} \|\boldsymbol{v}_t^i - \boldsymbol{v}_t^j\|$ for $t_r \leq T \leq t_{r+1} - 1$. Suppose the learning rate satisfies the conditions in Lemmas 12 and 13.*

*If* $\max_{i \in [N], t_r \le t \le T-1} \|\boldsymbol{a}_t^i\| \le K_a$ *holds, with probability at least* $1 - \delta$*, we have*

$$\max_{i,j \in [N]} O_{r,T}^{i,j} \le c\eta(\sqrt{Id} + I)\sqrt{\log(Nd/\delta)}K_a^2.$$

*In addition, we also have*

$$\max_{t_r \le t \le T} |(\boldsymbol{a}_t^i - \boldsymbol{a}_t^j)^\top \boldsymbol{a}^*| \le c\eta(\sqrt{kI} + I)\sqrt{\log(1/\eta\delta)}K_a^2.$$

*Proof.* Denote the discrepancy term by

$$\boldsymbol{h}_t^{i,j} = \mathbf{P}_t^j \nabla_{\boldsymbol{w}} L(\boldsymbol{w}_t^j, \boldsymbol{a}_t^j) - \mathbf{P}_t^i \nabla_{\boldsymbol{w}} L(\boldsymbol{w}_t^i, \boldsymbol{a}_t^i)$$

$$= \left[ \frac{\pi - \phi_t^j}{2\pi} \frac{(\boldsymbol{a}_t^j)^\top \boldsymbol{a}^*}{\|\boldsymbol{v}_t^j\|} (\mathbf{I} - \boldsymbol{w}_t^j(\boldsymbol{w}_t^j)^\top) - \frac{\pi - \phi_t^i}{2\pi} \frac{(\boldsymbol{a}_t^i)^\top \boldsymbol{a}^*}{\|\boldsymbol{v}_t^i\|} (\mathbf{I} - \boldsymbol{w}_t^i(\boldsymbol{w}_t^i)^\top) \right] \boldsymbol{w}^*. \quad \text{(C.1)}$$

Since $\|\boldsymbol{w}^*\| = 1$ and $\|\boldsymbol{a}_t^i\| \le 5K_a$ in Lemma 12, we can bound $\boldsymbol{h}_t^{i,j}$ as

$$\|\boldsymbol{h}_t^{i,j}\| \le \frac{|\phi_t^j - \phi_t^i|}{2\pi} \frac{(\boldsymbol{a}_t^i)^\top \boldsymbol{a}^*}{\|\boldsymbol{v}_t^i\|} + \frac{\pi - \phi_t^j}{2\pi} \frac{|(\boldsymbol{a}_t^i - \boldsymbol{a}_t^j)^\top \boldsymbol{a}^*|}{\|\boldsymbol{v}_t^i\|} + \frac{\pi - \phi_t^j}{2\pi}(\boldsymbol{a}_t^j)^\top \boldsymbol{a}^* \left| \frac{1}{\|\boldsymbol{v}_t^i\|} - \frac{1}{\|\boldsymbol{v}_t^j\|} \right|$$

$$+ \frac{\pi - \phi_t^i}{\pi} \frac{(\boldsymbol{a}_t^j)^\top \boldsymbol{a}^*}{\|\boldsymbol{v}_t^j\|} \left| \frac{1}{\|\boldsymbol{v}_t^i\|} - \frac{1}{\|\boldsymbol{v}_t^j\|} \right|$$

$$\le 3K_a^2 + |(\boldsymbol{a}_t^i - \boldsymbol{a}_t^j)^\top \boldsymbol{a}^*| + 6K_a^2 \|\boldsymbol{v}_t^i - \boldsymbol{v}_t^j\| + 5K_a^2 \left( \frac{\|\boldsymbol{v}_t^i - \boldsymbol{v}_t^j\|}{\|\boldsymbol{v}_t^i\|} + \frac{\|\boldsymbol{v}_t^i - \boldsymbol{v}_t^j\|}{2\|\boldsymbol{v}_t^i\|\|\boldsymbol{v}_t^j\|} \right)$$

$$\le 3K_a^2 + |(\boldsymbol{a}_t^i - \boldsymbol{a}_t^j)^\top \boldsymbol{a}^*| + 10K_a^2 \|\boldsymbol{v}_t^i - \boldsymbol{v}_t^j\|, \quad \text{(C.2)}$$

where we also used the fact $\max_{i \in [N]} \|\mathbf{I} - \boldsymbol{w}_t^i(\boldsymbol{w}_t^i)^\top\| \le 1$ and

$$\|\boldsymbol{w}_t^i(\boldsymbol{w}_t^i)^\top \boldsymbol{w}^* - \boldsymbol{w}_t^j(\boldsymbol{w}_t^j)^\top \boldsymbol{w}^*\| \le \|\boldsymbol{w}_t^i(\boldsymbol{w}_t^i - \boldsymbol{w}_t^j)^\top \boldsymbol{w}^*\| + \|(\boldsymbol{w}_t^i - \boldsymbol{w}_t^j)(\boldsymbol{w}_t^j)^\top \boldsymbol{w}^*\|$$

$$\le 2 \left\| \boldsymbol{w}_t^i - \boldsymbol{w}_t^j \right\|$$

$$= 2 \left| \frac{1}{\|\boldsymbol{v}_t^i\|} - \frac{1}{\|\boldsymbol{v}_t^j\|} \right|.$$

From the definition of $B_s^i$ and $B_s^j$ in Lemma 2, we know for any $i, j \in [N]$,

$$\left| B_s^i - B_s^j \right| \le \left| g(\phi_s^j) - g(\phi_s^i) \right| + \frac{\eta}{2\pi} \frac{(\mathbf{1}^\top \boldsymbol{a}^*)^2}{\|\boldsymbol{a}^*\|^2} \sum_{l=t_r}^{s-1} \left( 1 - \eta \frac{\pi + k - 1}{2\pi} \right)^{s-1-l} \left| g(\phi_l^j) - g(\phi_l^i) \right|$$

$$\le \pi + \frac{\eta I}{2} \frac{(\mathbf{1}^\top \boldsymbol{a}^*)^2}{\|\boldsymbol{a}^*\|^2},$$

where the last inequality holds due to $g(\phi) \in [0, \pi]$ for $\phi \in [0, \pi]$. Using concentration (A.18) and $t \le k\eta^{-1}$, with probability at least $1 - \delta$, we can guarantee

$$|S(\boldsymbol{\epsilon}_{t_r:s-1})| = \frac{\eta^2 |\mathbf{1}^\top \boldsymbol{a}^*|}{2\pi} \left| \sum_{l=t_r}^{s-1} \left( 1 - \frac{2\eta(\pi - 1)}{\pi} \right)^{s-1-l} \sum_{m=t_r}^{l-1} \left( 1 - \eta \frac{\pi + k - 1}{2\pi} \right)^{l-1-m} (\mathbf{1}^\top \boldsymbol{\epsilon}_m) \right|$$

$$+ \eta \left| \sum_{j=t_r}^{s-1} \left( 1 - \frac{2\eta(\pi - 1)}{\pi} \right)^{s-1-j} \boldsymbol{\epsilon}_j^\top \boldsymbol{a}^* \right|$$

$$\le c\eta I \sqrt{\frac{\log(2t/\delta)}{N}} K_a^2 \cdot \left( \eta \frac{|\mathbf{1}^\top \boldsymbol{a}^*|}{\|\boldsymbol{a}^*\|} \sum_{l=t_r}^{s-1} \left( 1 - \frac{2\eta(\pi - 1)}{\pi} \right)^{s-1-l} + 1 \right)$$

$$\le 2c\eta I \sqrt{\frac{k \log(2t/\delta)}{N}} K_a^2. \quad \text{(C.3)}$$

Using the dynamic of $\boldsymbol{a}_t^\top \boldsymbol{a}^*$ in Lemma 2, with probability at least $1 - \delta$, we also have

$$
\begin{aligned}
\left| (\boldsymbol{a}_{t-1}^j - \boldsymbol{a}_{t-1}^i)^\top \boldsymbol{a}^* \right| &= \left| \frac{\eta \|\boldsymbol{a}^*\|^2}{2\pi} \sum_{s=t_r}^{t-2} \left( 1 - \eta \frac{\pi-1}{2\pi} \right)^{t-2-s} (B_s^j - B_s^i) + S(\boldsymbol{\epsilon}_{t_r:t-2}^j) - S(\boldsymbol{\epsilon}_{t_r:t-2}^i) \right| \\
&\leq \frac{\eta I \|\boldsymbol{a}^*\|^2}{2\pi} \left( \pi + \frac{\eta I}{2} \frac{(\mathbf{1}^\top \boldsymbol{a}^*)^2}{\|\boldsymbol{a}^*\|^2} \right) + \left| S(\boldsymbol{\epsilon}_{t_r:t-2}^j) - S(\boldsymbol{\epsilon}_{t_r:t-2}^i) \right| \\
&\leq \frac{\eta I}{2\pi} \left( \pi \|\boldsymbol{a}^*\|^2 + \frac{\eta I}{2} (\mathbf{1}^\top \boldsymbol{a}^*)^2 \right) + 2c\eta \sqrt{kI \log(t/\delta)} K_a^2 \\
&\leq 2\eta (\sqrt{kI} + I) \sqrt{\log(1/\eta\delta)} K_a^2, \quad\quad\quad (C.4)
\end{aligned}
$$

where the second last inequality holds due to the concentration (C.3). Plugging (C.4) into (C.2) gives

$$
\begin{aligned}
\|\boldsymbol{v}_t^i - \boldsymbol{v}_t^j\| &\leq \eta \left\| \sum_{s=t_r}^{t-1} \boldsymbol{h}_s^{i,j} \right\| + \eta \left\| \sum_{s=t_r}^{t-1} \mathbf{P}_s^i \boldsymbol{\xi}_s^i - \mathbf{P}_s^j \boldsymbol{\xi}_s^j \right\| \\
&\leq \eta \sum_{s=t_r}^{t-1} \left( 3K_a^2 + |(\boldsymbol{a}_s^i - \boldsymbol{a}_s^j)^\top \boldsymbol{a}^*| + 10K_a^2 \|\boldsymbol{v}_s^i - \boldsymbol{v}_s^j\| \right) + \eta \left\| \sum_{s=t_r}^{t-1} \mathbf{P}_s^i \boldsymbol{\xi}_s^i \right\| + \eta \left\| \sum_{s=t_r}^{t-1} \mathbf{P}_s^j \boldsymbol{\xi}_s^j \right\| \\
&\leq \eta I \left( 3K_a^2 + 2\eta(I + \sqrt{kI})K_a^2 + 10K_a^2 O_{r,T}^{i,j} \right) + 4c\eta \sqrt{dI \log(N/\delta)} K_a^2, \quad\quad (C.5)
\end{aligned}
$$

where we used the following concentration

$$
\max_i \left\| \sum_{s=t_r}^{t-1} \mathbf{P}_s^i \boldsymbol{\xi}_s^i \right\| \leq \sqrt{d} \max_{i \in [N], \ell \in [d]} \left| (\mathbf{P}_s^i \boldsymbol{\xi}_s^i)_\ell \right| \leq c \sqrt{dI \log(dN/\delta)} K_a^2.
$$

Taking maximum on the left hand side of (C.5) over $t_r \leq t \leq T$, we can get

$$
\begin{aligned}
O_{r,T}^{i,j} &\leq (1 - 10\eta I K_a^2) \left[ \eta I \left( 3K_a^2 + 2\eta(I + \sqrt{kI})K_a^2 \right) + 4c\eta \sqrt{dI \log(N/\delta)} K_a^2 \right] \\
&\leq 2 \left( 4\sqrt{\frac{I}{d \log(Nd/\delta)}} + 4c \right) \eta \sqrt{dI \log(Nd/\delta)} K_a^2 \\
&\leq 2 \left( 5c + 4\sqrt{\frac{I}{d}} \right) \eta \sqrt{dI \log(Nd/\delta)} K_a^2,
\end{aligned}
$$

where we also used the assumption $\eta(I + \sqrt{kI}) < 1$. $\qquad\qquad\square$

## C.2 The Discrepancy in the Second Layer

**Lemma 15** (The discrepancy in the second layer). *Let $\Xi_{r,T}^{i,j} = \max_{t_r \leq t \leq T} \|\boldsymbol{a}_t^i - \boldsymbol{a}_t^j\|$ for $t_r \leq T \leq t_{r+1} - 1$. Suppose the learning rate satisfies the conditions in Lemmas 12 and 13. If $\max_{i \in [N], t_r \leq t \leq T-1} \|\boldsymbol{a}_t^i\| \leq K_a$ holds for $K_a > \|\boldsymbol{a}^*\|$ and $c\eta kI < 1$, with probability at least $1 - \delta$, we have*

$$
\max_{i,j} \Xi_{r,T}^{i,j} \leq 2K_a \left( 2\eta I + c\eta \sqrt{kI \log(k/\delta)} \right).
$$

*Proof.* Recall the closed form of $\nabla_{\boldsymbol{a}} L(\boldsymbol{w}, \boldsymbol{a})$, then for $i, j \in [N]$, we have

$$
\nabla_{\boldsymbol{a}} L(\boldsymbol{w}_t^i, \boldsymbol{a}_t^i) - \nabla_{\boldsymbol{a}} L(\boldsymbol{w}_t^j, \boldsymbol{a}_t^j) = \frac{\eta}{2\pi} \left( \mathbf{1}\mathbf{1}^\top + (\pi-1)\mathbf{I} \right) (\boldsymbol{a}_t^i - \boldsymbol{a}_t^j) - \frac{\eta}{2\pi} \left( g(\phi_t^i) - g(\phi_t^j) \right) \boldsymbol{a}^*.
$$

Let $t_r$ be the previous synchronization time before $t$. Using the fact $\boldsymbol{a}_{t_r}^i = \boldsymbol{a}_{t_r}^j = \boldsymbol{a}_{t_r}$, we have

$$
\|\boldsymbol{a}_t^i - \boldsymbol{a}_t^j\| \leq \eta \left\| \sum_{s=t_r}^{t-1} \nabla_{\boldsymbol{a}} L(\boldsymbol{w}_s^i, \boldsymbol{a}_s^i) - \nabla_{\boldsymbol{a}} L(\boldsymbol{w}_s^j, \boldsymbol{a}_s^j) \right\| + \eta \left\| \sum_{s=t_r}^{t-1} \boldsymbol{\epsilon}_s^i - \boldsymbol{\epsilon}_s^j \right\|
$$

$$\leq \frac{\eta(\pi+k-1)}{2\pi}\sum_{s=t_r}^{t-1}\|\boldsymbol{a}_s^i-\boldsymbol{a}_s^j\| + \frac{\eta\|\boldsymbol{a}^*\|}{2\pi}\sum_{s=t_r}^{t-1}\left|g(\phi_{t-1}^i)-g(\phi_{t-1}^j)\right| + \eta\left\|\sum_{s=t_r}^{t-1}\boldsymbol{\epsilon}_s^i-\boldsymbol{\epsilon}_s^j\right\|$$

$$\leq \frac{\eta(\pi+k-1)}{2\pi}\sum_{s=t_r}^{t-1}\|\boldsymbol{a}_s^i-\boldsymbol{a}_s^j\| + \frac{\eta\|\boldsymbol{a}^*\|}{2\pi}\pi I + c\sqrt{kI\log(k/\delta)}K_a, \tag{C.6}$$

where the last inequality holds due to Lemma 11(1). Taking maximum on the both sides over $t_r \leq t \leq T$, we get

$$\Xi_{r,T}^{i,j} \leq \left(1 - \frac{\eta(\pi+k-1)I}{2\pi}\right)^{-1}\left[\frac{\pi\eta I}{2}\|\boldsymbol{a}^*\| + c\eta\sqrt{kI\log(k/\delta)}K_a\right]$$

$$\leq 2K_a\left(2\eta I + c\eta\sqrt{kI\log(k/\delta)}\right).$$

$\square$

## D  Self-correction of Signals

Recalling the dynamic in Lemma 2,

$$\boldsymbol{a}_t^\top\boldsymbol{a}^* = \left(1-\eta\frac{\pi-1}{2\pi}\right)^t\boldsymbol{a}_0^\top\boldsymbol{a}^* + \frac{1-\left(\frac{2\pi-\eta(k+\pi-1)}{2\pi-\eta(\pi-1)}\right)^t}{k}\left(1-\eta\frac{\pi-1}{2\pi}\right)^t A_0$$

$$+ \frac{\eta\|\boldsymbol{a}^*\|^2}{2\pi}\sum_{s=0}^{t-1}\left(1-\eta\frac{\pi-1}{2\pi}\right)^{t-1-s}\frac{1}{N}\sum_{i=1}^N B_s^i + S(\boldsymbol{\epsilon}_{0:t-1}), \tag{D.1}$$

where

$$B_s^i = g(\phi_s^i) - 1 + \frac{\eta}{2\pi}\frac{(\mathbf{1}^\top\boldsymbol{a}^*)^2}{\|\boldsymbol{a}^*\|^2}\sum_{l=0}^{s-1}\left(1-\eta\frac{\pi+k-1}{2\pi}\right)^{s-1-l}(\pi-g(\phi_l^i)). \tag{D.2}$$

Using concentration (A.18), with probability at least $1-\delta$, we can guarantee

$$\max_t|S(\boldsymbol{\epsilon}_{0:t-1})| = \frac{\eta^2|\mathbf{1}^\top\boldsymbol{a}^*|}{2\pi}\max_t\left\{\left|\sum_{l=0}^{t-1}\left(1-\frac{2\eta(\pi-1)}{\pi}\right)^{t-l-1}\sum_{m=0}^{l-1}\left(1-\eta\frac{\pi+k-1}{2\pi}\right)^{l-1-m}(\mathbf{1}^\top\boldsymbol{\epsilon}_m)\right|\right.$$

$$\left.+\eta\left|\sum_{j=0}^t\left(1-\frac{2\eta(\pi-1)}{\pi}\right)^{t-1-j}\boldsymbol{\epsilon}_j^\top\boldsymbol{a}^*\right|\right\}$$

$$\leq c\sqrt{\frac{\eta\log(2/\delta)}{N}}K_a^2\cdot\max_t\left[\eta\frac{|\mathbf{1}^\top\boldsymbol{a}^*|}{\|\boldsymbol{a}^*\|}\sum_{l=0}^{t-1}\left(1-\frac{2\eta(\pi-1)}{\pi}\right)^{t-l-1}+1\right]$$

$$\leq 2c\sqrt{\frac{\eta k\log(2/\delta)}{N}}K_a^2. \tag{D.3}$$

For any $0 \leq s \leq \widetilde{O}(\eta^{-2})$ and any $i \in [N]$, with probability at least $1-\delta$, it holds that

$$|\cos\phi_s^i - \cos\phi_s| = \left|\frac{(\boldsymbol{v}_s^i)^\top\boldsymbol{w}^*}{\|\boldsymbol{v}_s^i\|} - \frac{\boldsymbol{v}_s^\top\boldsymbol{w}^*}{\|\boldsymbol{v}_s\|}\right|$$

$$\leq \frac{|(\boldsymbol{v}_s^i-\boldsymbol{v}_s)^\top\boldsymbol{w}^*|}{\|\boldsymbol{v}_s^i\|} + |\boldsymbol{v}_s^\top\boldsymbol{w}^*|\left|\frac{1}{\|\boldsymbol{v}_s^i\|} - \frac{1}{\|\boldsymbol{v}_s\|}\right|$$

$$\overset{(i)}{\leq} \frac{\|\boldsymbol{v}_s^i-\boldsymbol{v}_s\|}{\|\boldsymbol{v}_s\| - \|\boldsymbol{v}_s^i-\boldsymbol{v}_s\|} + \frac{\|\boldsymbol{v}_s^i-\boldsymbol{v}_s\|}{\|\boldsymbol{v}_s\| - \|\boldsymbol{v}_s^i-\boldsymbol{v}_s\|}$$

$$\overset{(ii)}{\leq} \frac{2c\eta K_a^2(\sqrt{Id}+I)\sqrt{\log(Nd/\delta)}}{1/2 - c\eta K_a^2(\sqrt{Id}+I)\sqrt{\log(Nd/\delta)}}$$

$$\leq 12c\eta K_a^2(\sqrt{Id}+I)\sqrt{\log(Nd/\delta)}, \tag{D.4}$$

where $(i)$ follows from $\|\boldsymbol{w}^*\|=1$ and the triangle inequality; and $(ii)$ holds due to Lemma 14. Define a good event

$$\mathcal{E}_{\text{good}} = \cap_{\ell=0}^4 \mathcal{E}_\ell, \tag{D.5}$$

where

$$\mathcal{E}_0 = \left\{ \max_{t\geq 0} |S(\boldsymbol{\epsilon}_{0:t-1})| \leq 2c\sqrt{\frac{\eta k \log(2/\delta)}{N}} K_a^2 \right\},$$

$$\mathcal{E}_1 = \left\{ \max_{i\in[N], s\leq \widetilde{O}(\eta^{-2})} \left| \boldsymbol{a}_m^\top \boldsymbol{a}^* - (\boldsymbol{a}_m^i)^\top \boldsymbol{a}^* \right| \leq c\eta^2(I+\sqrt{kI})\sqrt{\log(N/\eta\delta)} K_a^2 \right\},$$

$$\mathcal{E}_2 = \left\{ \max_{s\leq \widetilde{O}(\eta^{-2}), 0\leq \tau\leq \widetilde{O}(\eta^{-2})} \left| \sum_{m=\tau}^{s-1} (\boldsymbol{\xi}_m^\top \boldsymbol{w}^* + \boldsymbol{h}_m^\top \boldsymbol{w}^*) \right| \leq cK_a\sqrt{\frac{\eta^{-1}\log(1/\eta\delta)}{N}} \right\},$$

$$\mathcal{E}_3 = \left\{ \max_{i\in[N], s\leq \widetilde{O}(\eta^{-2})} |\cos\phi_s^i - \cos\phi_s| \leq 12c\eta K_a^2(\sqrt{Id}+I)\sqrt{\log(Nd/\delta)} \right\},$$

$$\mathcal{E}_4 = \left\{ \max_{0\leq\tau\leq\widetilde{O}(\eta^{-2})} \|\boldsymbol{a}_\tau\| \leq K_a, \quad \min_{0\leq\tau\leq\tau_{1,1}} A_\tau \geq -c\eta\sqrt{\frac{\eta\log(1/\eta\delta)}{N}} \right\}.$$

According to Lemmas 14, 12 and relations (D.3), (D.4), (A.20), we know $\mathbb{P}(\mathcal{E}_\ell^c)\leq\delta$ for $0\leq\ell\leq 4$.

## D.1 Self-correction of the Second Layer

**Theorem 1 restated.** For any initial point $(\boldsymbol{v}_0, \boldsymbol{a}_0)$ with $\phi_0\in[0,\pi)$, we denote $\tau_a = \inf\{t\geq 0: \boldsymbol{a}_t^\top \boldsymbol{a}^* \geq \gamma_a\}$ the first time, where $\gamma_a = \frac{16(\pi-1)|\mathbf{1}^\top \boldsymbol{a}^*|^2}{k(\pi+k-1)}$. Under Assumption 1, if

$$ck^2\sqrt{\log(Ndk/\delta)}\max\left\{ \eta(\sqrt{Ik}+I), \eta(\sqrt{Id}+I), \sqrt{\frac{\eta k}{N}} \right\} \frac{\|\boldsymbol{a}^*\|^2}{|\mathbf{1}^\top \boldsymbol{a}^*|^2} < 1, \tag{D.6}$$

for a sufficiently large constant $c$, then $\tau_a \leq \widetilde{O}(\eta^{-2})$ with probability at least $1-\delta$.

*Proof.* We first define a sequence of events as

$$\mathcal{A}_t := \left\{ \forall s\leq t: \boldsymbol{a}_s^\top \boldsymbol{a}^* \leq \gamma_a \right\}, \quad \text{where } \gamma_a = \frac{16(\pi-1)(\mathbf{1}^\top \boldsymbol{a}^*)^2}{k(\pi+k-1)}.$$

Notice that once we show for some $\tau > 0$,

$$\mathbb{P}\left( \mathcal{A}_{\tau-1} \cap \{\boldsymbol{a}_\tau^\top \boldsymbol{a}^* < \gamma_a\} \right) \leq \delta,$$

then we can conclude that $\mathbb{P}(\mathcal{A}_\tau)\leq\delta$. According to the definition of $\tau_a$, we can guarantee $\tau_a\leq\tau$ with probability at least $1-\delta$.

For simplicity, we introduce the following stopping time w.r.t. the angle $\phi$ during the training process,

$$\tau_-(\phi) = \inf_{t\geq 0}\{t: \phi_t\geq\phi\} \quad \text{for} \quad \phi\in[0,\pi].$$

Define $\tilde{\phi}^l$ and $\tilde{\phi}^u$ such that

$$\cos\tilde{\phi}^l = \frac{144\pi}{\pi+k-1}\frac{(\mathbf{1}^\top \boldsymbol{a}^*)^2}{k}\log(4+k^2), \quad \cos\tilde{\phi}^u = -\cos\tilde{\phi}^l.$$

Choose $\tilde{\phi}^o = \arccos(1/5)$ such that $\cos\tilde{\phi}^l \leq 1/5 = \cos\tilde{\phi}^o$ due to Assumption 1 such that

$$(\mathbf{1}^\top \boldsymbol{a}^*)^2 \leq \frac{k(\pi+k-1)}{720\pi\log(4+k^2)}. \tag{D.7}$$

The correction condition (6) is equivalent to

$$\left(1 - \frac{32(\pi - 1)}{k}\right)\frac{\pi - 1}{\pi + k - 1}\frac{(\mathbf{1}^\top \boldsymbol{a}^*)^2}{\|\boldsymbol{a}^*\|^2} > 1. \tag{D.8}$$

We denote

$$\mathcal{Z}_1 = \left\{\tau_-(\tilde{\phi}_0) > \tau_{1,1} - 1\right\},$$
$$\mathcal{Z}_2 = \left\{\tau_-(\pi/2) > \tau_-(\tilde{\phi}^o) + \tau_{1,2} - 1\right\},$$
$$\mathcal{Z}_3 = \left\{\tau_-(\tilde{\phi}^u) > \tau_-(\pi/2) + \tau_{1,3} - 1\right\},$$

where $\tau_{1,1}$, $\tau_{1,2}$ and $\tau_{1,3}$ are deterministic and will be defined later.

**Initial region:** $(0, \tilde{\phi}^o)$. For any $s \leq \min\left\{\tau_-(\tilde{\phi}^o), \widetilde{O}(\eta^{-2})\right\}$, under the event $\mathcal{E}_{\text{good}}$, it holds that

$$\cos \phi_s^i \geq \cos \phi_s - |\cos \phi_s^i - \cos \phi_s|$$
$$\geq \cos \tilde{\phi}^o - 12c\eta K_a^2(\sqrt{Id} + I)\sqrt{\log(Nd/\delta)}$$
$$\geq \frac{\cos \tilde{\phi}^o}{2} = \frac{1}{10} > \arccos(0.46\pi).$$

Since $g(\cdot) \in [0, \pi]$ is decreasing in $[0, \pi]$, plugging it into (D.2), we can get the following lower bound

$$\frac{1}{N}\sum_{i=1}^{N} B_s^i \geq \frac{1}{N}\sum_{i=1}^{N}[g(\phi_s^i) - 1] \geq g(0.46\pi) - 1,$$

Under the event $\mathcal{E}_{\text{good}}$, using the lower bound of $A_0$ in (B.5), for $t \leq \min\left\{\tau_-(\tilde{\phi}^o), \widetilde{O}(\eta^{-2})\right\}$, we can guarantee

$$\boldsymbol{a}_t^\top \boldsymbol{a}^* \geq \eta\|\boldsymbol{a}^*\|^2\frac{g(0.46\pi) - 1}{2\pi}\sum_{s=0}^{t-1}\left(1 - \frac{\eta(\pi - 1)}{2\pi}\right)^{t-1-s}\left[1 - \left(1 - \eta\frac{\pi + k - 1}{2\pi}\right)^s\right]$$
$$- \|\boldsymbol{a}^*\|^2\left(1 - \frac{\eta(\pi - 1)}{2\pi}\right)^t - 2c\sqrt{\frac{\eta\log(2/\delta)}{N}}K_a^2$$
$$= \frac{g(0.46\pi) - 1}{\pi - 1}\|\boldsymbol{a}^*\|^2\left[1 - \left(1 - \frac{\eta(\pi - 1)}{2\pi}\right)^t\right]$$
$$- \frac{1 - \left(\frac{2\pi - \eta(k + \pi - 1)}{2\pi - \eta(\pi - 1)}\right)^t}{k}\left(1 - \eta\frac{\pi - 1}{2\pi}\right)^t[g(0.46\pi) - 1]\|\boldsymbol{a}^*\|^2$$
$$- \|\boldsymbol{a}^*\|^2\left(1 - \frac{\eta(\pi - 1)}{2\pi}\right)^t - 2c\sqrt{\frac{\eta k\log(2/\delta)}{N}}K_a^2.$$

Here we take

$$\tau_{1,1} = \frac{1}{\eta}\frac{\pi - 1}{2\pi}\log\left(4 + \frac{4(\pi - 1)}{g(0.46\pi) - 1}\right) \leq \frac{1}{\eta}\frac{\pi - 1}{2\pi}\log\left(4 + 20(\pi - 1)\right), \tag{D.9}$$

where we used the fact $g(0.46\pi) - 1 \geq 0.2$. Since $\tau_{1,1} - 1 \leq \widetilde{O}(\eta^{-2})$, under the event $\left\{\tau_{1,1} - 1 < \tau_-(\tilde{\phi}^o)\right\} \cap \mathcal{E}_{\text{good}}$, it holds that

$$\boldsymbol{a}_{\tau_{1,1}}^\top \boldsymbol{a}^* \geq \frac{(g(0.46\pi) - 1)\|\boldsymbol{a}^*\|^2}{2(\pi - 1)} - 2c\sqrt{\frac{\eta k\log(2t/\delta)}{N}}K_a^2$$
$$\overset{(i)}{\geq} \frac{0.2(\mathbf{1}^\top \boldsymbol{a}^*)^2}{2k(\pi - 1)} - 2c\sqrt{\frac{\eta k\log(2t/\delta)}{N}}K_a^2$$

$$\overset{(ii)}{\geq} \frac{32(\pi-1)(\mathbf{1}^\top \boldsymbol{a}^*)^2}{k(\pi+k-1)} - 2c\sqrt{\frac{\eta k \log(2t/\delta)}{N}} K_a^2$$

$$\overset{(iii)}{\geq} \frac{16(\pi-1)(\mathbf{1}^\top \boldsymbol{a}^*)^2}{k(\pi+k-1)} = \gamma_a,$$

where $(i)$ holds due to $\|\boldsymbol{a}^*\| \geq |\mathbf{1}^\top \boldsymbol{a}^*|/\sqrt{k}$; $(ii)$ follows from $k \geq 320(\pi-1)^2$; and $(iii)$ holds due to the assumption (D.6). We can conclude that

$$\mathbb{P}\left(\left\{\tau_{1,1}-1 < \tau_-(\check{\phi}^o)\right\} \cap \{\tau_a > \tau_{1,1}\}\right) \leq \mathbb{P}\left(\left\{\tau_{1,1}-1 < \tau_-(\check{\phi}^o)\right\} \cap \left\{\boldsymbol{a}_{\tau_{1,1}}^\top \boldsymbol{a}^* \leq \gamma_a\right\}\right)$$

$$\leq \mathbb{P}\left(\left\{\tau_{1,1}-1 < \tau_-(\check{\phi}^o)\right\} \cap (\mathcal{E}_0^c \cup \mathcal{E}_3^c)\right)$$

$$\leq \mathbb{P}\left(\mathcal{E}_0^c \cup \mathcal{E}_3^c\right) \leq 2\delta, \tag{D.10}$$

where the second last inequality holds since

$$\left\{\tau_{1,1}-1 < \tau_-(\check{\phi}^o) \leq \widetilde{O}(\eta^{-2})\right\} \cap \mathcal{E}_{\text{good}} \subseteq \{\boldsymbol{a}_{\tau_{1,1}}^\top \boldsymbol{a}^* \geq \gamma_a\},$$

such that

$$\left\{\tau_{1,1}-1 < \tau_-(\check{\phi}^o)\right\} \cap \left\{\boldsymbol{a}_{\tau_{1,1}}^\top \boldsymbol{a}^* \leq \gamma_a\right\} \subseteq \left\{\tau_{1,1}-1 < \tau_-(\check{\phi}^o)\right\} \cap \mathcal{E}_{\text{good}}^c.$$

If $\tau_{1,1} \geq \tau_-(\check{\phi}^o)$, we regard $(\boldsymbol{v}_{\tau_-(\check{\phi}^o)}, \boldsymbol{a}_{\tau_-(\check{\phi}^o)})$ as the new initial point and step into the analysis of the next region.

**Initial region:** $[\check{\phi}^o, \pi/2)$. Denote $t_0 = \tau_-(\check{\phi}^o)$. From now on, we assume the event $\mathcal{Z}_1^c$ happens and hide the indicator. Recalling the definition of $\boldsymbol{h}_t$, we have

$$\|\boldsymbol{h}_t\| \leq \frac{1}{N}\sum_{i=1}^N \|\mathbf{P}_t - \mathbf{P}_t^i\|\|\nabla_{\boldsymbol{w}} L(\boldsymbol{w}_t^i, \boldsymbol{a}_t^i)\|$$

$$\overset{(i)}{\leq} 24c\eta(\sqrt{Id}+I)\sqrt{\log(Nd/\delta)}K_a^2 \cdot \frac{1}{N}\sum_{i=1}^N \|\nabla_{\boldsymbol{w}} L(\boldsymbol{w}_t^i, \boldsymbol{a}_t^i)\|$$

$$\overset{(ii)}{\leq} 24c\eta(\sqrt{Id}+I)\sqrt{\log(Nd/\delta)}K_a^4, \tag{D.11}$$

where $(i)$ holds due to (B.9) and $(ii)$ is true because $\|\nabla_{\boldsymbol{w}} L(\boldsymbol{w}_t^i, \boldsymbol{a}_t^i)\| \leq |(\boldsymbol{a}_t^i)^\top \boldsymbol{a}^*| \leq K_a^2$. In addition, we also have

$$\boldsymbol{v}_{t+1}^\top \boldsymbol{w}^* = \boldsymbol{v}_t^\top \boldsymbol{w}^* - \eta \frac{1}{N}\sum_{i=1}^N \mathbf{P}_t^i \nabla_{\boldsymbol{w}} L(\boldsymbol{w}_t^i, \boldsymbol{a}_t^i; \mathbf{Z}_t^i)$$

$$= \check{\boldsymbol{v}}_{t+1}^\top \boldsymbol{w}^* - \eta \boldsymbol{h}_t^\top \boldsymbol{w}^*$$

$$= \boldsymbol{v}_t^\top \boldsymbol{w}^* - \eta \left(\mathbf{P}_t \frac{1}{N}\sum_{i=1}^N \nabla_{\boldsymbol{w}} L(\boldsymbol{w}_t, \boldsymbol{a}_t) + \boldsymbol{\xi}_t\right)^\top \boldsymbol{w}^* - \eta \boldsymbol{h}_t^\top \boldsymbol{w}^*$$

$$= \boldsymbol{v}_t^\top \boldsymbol{w}^* + \frac{\eta}{2\pi} \frac{1}{N}\sum_{i=1}^N \frac{\pi-\phi_t^i}{\|\boldsymbol{v}_t\|}(\boldsymbol{a}_t^i)^\top \boldsymbol{a}^* \sin^2 \phi_t - \eta \boldsymbol{h}_t^\top \boldsymbol{w}^* - \eta \boldsymbol{\xi}_t^\top \boldsymbol{w}^*. \tag{D.12}$$

Here we take

$$\tau_{1,2} = \frac{1}{\eta} \frac{2\pi}{\pi-1} \log\left(4 + \frac{20(\pi+k-1)\|\boldsymbol{a}^*\|^2}{(\mathbf{1}^\top \boldsymbol{a}^*)^2 (\pi - g(\pi/5))}\right) \leq \frac{1}{\eta} \frac{2\pi}{\pi-1} \log(4 + k^2). \tag{D.13}$$

For any $t_0 \leq s \leq t-1 \leq t_0 + \tau_{1,2}$, under the event $\mathcal{A}_{t-1} \cap \mathcal{E}_{\text{good}}$, invoking (D.12) guarantees that,

$$\cos\phi_s = \frac{1}{\|\boldsymbol{v}_s\|}\left\{\boldsymbol{v}_{t_0}^\top \boldsymbol{w}^* + \frac{\eta}{2\pi} \frac{1}{N}\sum_{i=1}^N \sum_{m=t_0}^{s-1} \frac{\pi-\phi_m^i}{\|\boldsymbol{v}_m\|}(\boldsymbol{a}_m^i)^\top \boldsymbol{a}^* \sin^2 \phi_m - \eta \sum_{m=t_0}^{s-1}(\boldsymbol{h}_m^\top \boldsymbol{w}^* + \boldsymbol{\xi}_m^\top \boldsymbol{w}^*)\right\}$$

$$\overset{(i)}{\leq} 2\cos\phi_{t_0} + 2s\eta\gamma_a + 2\eta\sum_{m=t_0}^{s-1}\max_i\left|a_m^\top a^* - (a_m^i)^\top a^*\right| + 2\eta\left|\sum_{m=t_0}^{s-1}(\xi_m^\top w^* + h_m^\top w^*)\right|$$

$$\overset{(ii)}{\leq} 2\cos\phi_{t_0} + 2\tau_{1,2}\eta\gamma_a + 2c\tau_{1,2}\eta^2(I+\sqrt{kI})\sqrt{\log(1/\eta\delta)}K_a^2 + 2\eta\max_{0\leq\tau\leq\tau_{1,1}}\left|\sum_{m=\tau}^{s-1}(\xi_m^\top w^* + h_m^\top w^*)\right|$$

$$\overset{(iii)}{\leq} 2\cos\tilde{\phi}^o + \frac{64\pi}{\pi+k-1}\frac{(\mathbf{1}^\top a^*)^2}{k}\log(4+k^2) + \frac{4\pi}{\pi-1}\log(4+k^2)\cdot c\eta(I+\sqrt{kI})\sqrt{\log(1/\eta\delta)}K_a^2$$

$$+ \frac{4\pi}{\pi-1}\log(4+k^2)\cdot 24c\eta K_a^2(\sqrt{Id}+I)\sqrt{\log(Nd/\delta)}$$

$$+ 2cK_a\sqrt{\frac{4\pi}{\pi-1}\log(4+k^2)\cdot\frac{\eta\log(Nd/\eta\delta)}{N}}$$

$$\overset{(iv)}{\leq} 2\cos\tilde{\phi}^o + \frac{128(\pi-1)}{\pi+k-1}\frac{(\mathbf{1}^\top a^*)^2}{k}\log(4+k^2) \leq 3\cos\tilde{\phi}^o, \tag{D.14}$$

where $(i)$ holds due to $\|v_s\| \geq 1/2$, $\|v_{t_0}\| \geq 1/2$ and the definition of $\mathcal{A}_{t-1}$; $(ii)$ comes from $t_0 \leq \tau_{1,1}$ and $\mathcal{E}_1$; $(iii)$ follows from $\phi_{t_0} \geq \tilde{\phi}^l$, and $\mathcal{E}_2$; and $(iv)$ holds due to the assumption (D.6). Further, under the event $\mathcal{A}_{t-1} \cap \mathcal{E}_{\text{good}}$, we have for any $i \in [N]$ and $t_0 \leq s \leq t-1 \leq t_0 + \tau_{1,2} - 1$

$$\cos\phi_s^i \leq 3\cos\tilde{\phi}^o + 12c\eta K_a^2(\sqrt{Id}+I)\sqrt{\log(Nd/\delta)}$$

$$\leq 4\cos\tilde{\phi}^o = \frac{4}{5} \leq \cos(\pi/5).$$

Under the event $\mathcal{A}_{t-1} \cap \mathcal{E}_{\text{good}} \cap \{\tau_-(\pi/2) - t_0 \geq \tau_{1,2}\}$, it implies that for any $t_0 \leq s \leq t-1 \leq t_0 + \tau_{1,2} - 1$,

$$\frac{1}{N}\sum_{i=1}^N B_s^i \geq \frac{1}{N}\sum_{i=1}^N\frac{\eta}{2\pi}\frac{(\mathbf{1}^\top a^*)^2}{\|a^*\|^2}\sum_{l=0}^{s-1}\left(1-\eta\frac{\pi+k-1}{2\pi}\right)^{s-1-l}(\pi - g(\phi_l^i))$$

$$\geq \frac{\pi - g(\pi/5)}{\pi+k-1}\frac{(\mathbf{1}^\top a^*)^2}{\|a^*\|^2}\left[1-\left(1-\eta\frac{\pi+k-1}{2\pi}\right)^s\right].$$

Together with (D.1) and (D.3), we can guarantee that

$$a_t^\top a^* = \left(1-\eta\frac{\pi-1}{2\pi}\right)^{t-t_0}a_{t_0}^\top a^* + \frac{1-\left(\frac{2\pi-\eta(k+\pi-1)}{2\pi-\eta(\pi-1)}\right)^{t-t_0}}{k}\left(1-\eta\frac{\pi-1}{2\pi}\right)^t A_{t_0}$$

$$+ \frac{\eta\|a^*\|^2}{2\pi}\sum_{s=t_0}^{t-1}\left(1-\eta\frac{\pi-1}{2\pi}\right)^{t-1-s}\frac{1}{N}\sum_{i=1}^N B_s^i + S(\epsilon_{t_0:t-1})$$

$$\geq \frac{\pi - g(\pi/5)}{\pi-1}\frac{(\mathbf{1}^\top a^*)^2}{\pi+k-1}\left[1-\left(1-\eta\frac{\pi-1}{2\pi}\right)^{t-t_0}\right]$$

$$- \frac{1-\left(\frac{2\pi-\eta(k+\pi-1)}{2\pi-\eta(\pi-1)}\right)^{t-t_0}}{k}\left(1-\eta\frac{\pi-1}{2\pi}\right)^{t-t_0}[\pi - g(\pi/5)]\frac{(\mathbf{1}^\top a^*)^2}{\pi+k-1}$$

$$- 5\|a^*\|^2\left(1-\frac{\eta(\pi-1)}{2\pi}\right)^{t-t_0} - 2c\sqrt{\frac{\eta k\log(1/\delta)}{N}}K_a^2.$$

Choosing $t = t_0 + \tau_{1,2}$, it holds that

$$a_{t_0+\tau_{1,2}}^\top a^* \geq \frac{3}{4}\frac{\pi - g(\pi/5)}{\pi-1}\frac{(\mathbf{1}^\top a^*)^2}{\pi+k-1} - 3c\sqrt{\frac{\eta k\log(1/\eta\delta)}{N}}K_a^2$$

$$\geq \frac{\pi - g(\pi/5)}{4(\pi-1)}\frac{(\mathbf{1}^\top a^*)^2}{\pi+k-1}$$

$$\geq \frac{0.5}{4(\pi-1)}\frac{(\mathbf{1}^\top a^*)^2}{\pi+k-1}$$

$$\geq \frac{16(\pi - 1)(\mathbf{1}^\top \boldsymbol{a}^*)^2}{k(\pi + k - 1)} = \gamma_a,$$

where we used the assumptions (D.6) and $k \geq 320(\pi - 1)^2$.

Recalling that $\mathcal{Z}_2 = \left\{\tau_-(\pi/2) \geq \tau_-(\tilde{\phi}^o) + \tau_{1,2}\right\}$, hence we have verified that

$$\mathbb{P}\left(\mathcal{Z}_1^c \cap \mathcal{Z}_2 \cap \{\tau_a > \tau_{1,1} + \tau_{1,2}\}\right) \leq \mathbb{P}\left(\mathcal{Z}_1^c \cap \mathcal{Z}_2 \cap \left\{\tau_a > \tau_-(\tilde{\phi}^o) + \tau_{1,2}\right\}\right)$$

$$= \mathbb{P}\left(\mathcal{Z}_1^c \cap \mathcal{Z}_2 \cap \mathcal{A}_{\tau_-(\tilde{\phi}^o)+\tau_{1,2}-1} \cap \left\{\boldsymbol{a}_{t_0+\tau_{1,2}}^\top \boldsymbol{a}^* \leq \gamma_a\right\}\right)$$

$$\leq \mathbb{P}\left(\mathcal{A}_{\tau_-(\tilde{\phi}^o)+\tau_{1,2}-1} \cap \mathcal{E}_{\text{good}}^c\right) \leq 5\delta. \tag{D.15}$$

If $\tau_{1,2} \geq \tau_-(\pi/2)$, we regard $(\boldsymbol{v}_{\tau_-(\pi/2)}, \boldsymbol{a}_{\tau_-(\pi/2)})$ as the new initial point and step into the analysis of the next region.

**Initial region:** $[\pi/2, \tilde{\phi}^u)$. From now on, we assign $t_0 = \tau_-(\pi/2)$ and consider the case $\left\{\tau_-(\tilde{\phi}^o) \leq \tau_{1,1}\right\} \cap \left\{t_0 < \tau_-(\tilde{\phi}^o) + \tau_{1,2}\right\}$. Take

$$\tau_{1,3} = \frac{1}{\eta}\frac{2\pi}{\pi - 1}\log\left(4 + \frac{24(\pi - 1)}{\zeta_a}\right) \leq \frac{1}{\eta}\frac{2\pi}{\pi - 1}\log(4 + k^2), \tag{D.16}$$

where the inequality holds due to our assumptions $k \geq 320(\pi - 1)^2$ and (D.8). Similar to (D.14), for any $t_0 \leq s \leq t - 1 \leq \widetilde{O}(\eta^{-2})$, under the event $\mathcal{A}_{t-1} \cap \mathcal{E}_{\text{good}}$, we have

$$\cos\phi_s = \frac{1}{\|\boldsymbol{v}_s\|}\left\{\boldsymbol{v}_{t_0}^\top \boldsymbol{w}^* + \frac{\eta}{2\pi}\frac{1}{N}\sum_{i=1}^{N}\sum_{m=t_0}^{s-1}\frac{\pi - \phi_m^i}{\|\boldsymbol{v}_m\|}(\boldsymbol{a}_m^i)^\top \boldsymbol{a}^* \sin^2\phi_m - \eta\sum_{m=t_0}^{s-1}(\boldsymbol{h}_m^\top \boldsymbol{w}^* + \boldsymbol{\xi}_m^\top \boldsymbol{w}^*)\right\}$$

$$\stackrel{(i)}{\leq} 2\tau_{1,3}\eta\gamma_a + 2c\tau_{1,3}\eta^2(I + \sqrt{kI})\sqrt{\log(1/\eta\delta)}K_a^2 + 2\eta\max_{\tau \leq \tau_{1,1}+\tau_{1,2}}\left|\sum_{m=\tau}^{s-1}(\boldsymbol{\xi}_m^\top \boldsymbol{w}^* + \boldsymbol{h}_m^\top \boldsymbol{w}^*)\right|$$

$$\stackrel{(ii)}{\leq} \frac{64\pi}{\pi + k - 1}\frac{(\mathbf{1}^\top \boldsymbol{a}^*)^2}{k}\log(4 + k^2) + \frac{4\pi}{\pi - 1}\log(4 + k^2) \cdot c\eta(I + \sqrt{kI})\sqrt{\log(1/\eta\delta)}K_a^2$$

$$+ \frac{6\pi}{\pi - 1}\log(4 + k^2) \cdot \left(24c\eta K_a^2(\sqrt{Id} + I)\sqrt{\log(Nd/\delta)} + cK_a\sqrt{\frac{\eta\log(Nd/\eta\delta)}{N}}\right)$$

$$\leq \frac{128\pi}{\pi + k - 1}\frac{(\mathbf{1}^\top \boldsymbol{a}^*)^2}{k}\log(4 + k^2), \tag{D.17}$$

where $(i)$ holds due to $\phi_{t_0} \geq \pi/2$ and the definition of $\mathcal{A}_{t-1}$; and $(ii)$ holds due to $s-1-\tau \leq \widetilde{O}(\eta^{-2})$. Under the event $\mathcal{A}_{t-1} \cap \mathcal{E}_{\text{good}}$, we can guarantee that

$$\max_i \cos\phi_s^i \leq \cos\phi_s + \max_i \left|\cos\phi_s - \cos\phi_s^i\right|$$

$$\leq \frac{128\pi}{\pi + k - 1}\frac{(\mathbf{1}^\top \boldsymbol{a}^*)^2}{k}\log(4 + k^2) + 12c\eta K_a^2(\sqrt{Id} + I)\sqrt{\log(Nd/\delta)}$$

$$\leq \frac{144\pi}{\pi + k - 1}\frac{(\mathbf{1}^\top \boldsymbol{a}^*)^2}{k}\log(4 + k^2)$$

$$= \cos\tilde{\phi}^l,$$

where we used the assumption (D.6). It implies that $\min_i \phi_s^i \geq \tilde{\phi}^l$ under the event $\mathcal{A}_{t-1} \cap \mathcal{E}_{\text{good}}$. In addition, we notice that $\cos\tilde{\phi}^l = -\cos\tilde{\phi}^u$, which implies $\tilde{\phi}^l + \tilde{\phi}^u = \pi$ and $\sin\tilde{\phi}^l = \sin\tilde{\phi}^u$. Under the event $\mathcal{A}_{t-1} \cap \mathcal{E}_{\text{good}} \cap \{t - 1 \leq \min\{\tau_-(\tilde{\phi}^u), \widetilde{O}(\eta^{-2})\}\}$, for any $s \leq t - 1$, we have

$$\frac{1}{N}\sum_{i=1}^{N}\left\{g(\phi_s^i) + \frac{\eta}{2\pi}\frac{(\mathbf{1}^\top \boldsymbol{a}^*)^2}{\|\boldsymbol{a}^*\|^2}\sum_{l=t_0}^{s-1}\left(1 - \eta\frac{\pi + k - 1}{2\pi}\right)^{s-1-l}(1 - g(\phi_l^i))\right\}$$

$$\geq g(\tilde{\phi}^u) + \frac{\eta}{2\pi} \frac{(\mathbf{1}^\top \mathbf{a}^*)^2}{\|\mathbf{a}^*\|^2} \sum_{l=t_0}^{s-1} \left(1 - \eta\frac{\pi+k-1}{2\pi}\right)^{s-1-l} \left(1 - g(\tilde{\phi}^l)\right)$$

$$\geq g(\tilde{\phi}^u) + \frac{\eta k}{2\pi} \sum_{l=t_0}^{s-1} \left(1 - \eta\frac{\pi+k-1}{2\pi}\right)^{s-1-l} \left(1 - g(\tilde{\phi}^l)\right)$$

$$= g(\tilde{\phi}^u) + \frac{k(1 - g(\tilde{\phi}^l))}{\pi+k-1} \left[1 - \left(1 - \eta\frac{\pi+k-1}{2\pi}\right)^s\right]$$

$$\geq g(\tilde{\phi}^u) + 1 - g(\tilde{\phi}^l)$$

$$= (\pi - \tilde{\phi}^u)\cos\tilde{\phi}^u + \sin\tilde{\phi}^u - (\pi - \tilde{\phi}^l)\cos\tilde{\phi}^l - \sin\tilde{\phi}^l + 1$$

$$= (2\pi - \tilde{\phi}^u - \tilde{\phi}^l)\cos\tilde{\phi}^u + 1$$

$$= -\frac{144\pi^2}{\pi+k-1} \frac{(\mathbf{1}^\top \mathbf{a}^*)^2}{k} \log(4 + k^2) + 1 \geq 0, \tag{D.18}$$

where the last inequality holds due to the condition (D.7). Together with the definition of $B_s^i$, we have

$$\frac{\mathbb{1}_{\mathcal{A}_{t-1}}}{N} \sum_{i=1}^N B_s^i = \frac{\mathbb{1}_{\mathcal{A}_{t-1}}}{N} \sum_{i=1}^N \left\{ (g(\phi_s^i) - 1) + \frac{\eta}{2\pi}\frac{(\mathbf{1}^\top \mathbf{a}^*)^2}{\|\mathbf{a}^*\|^2} \sum_{l=0}^{s-1} \left(1 - \eta\frac{\pi+k-1}{2\pi}\right)^{s-1-l} (\pi - g(\phi_l^i)) \right\}$$

$$\geq \frac{\eta}{2\pi}\frac{(\mathbf{1}^\top \mathbf{a}^*)^2}{\|\mathbf{a}^*\|^2} \sum_{l=0}^{s-1} \left(1 - \eta\frac{\pi+k-1}{2\pi}\right)^{s-1-l} (\pi - 1)$$

$$= \underbrace{\frac{\pi-1}{\pi+k-1}\frac{(\mathbf{1}^\top \mathbf{a}^*)^2}{\|\mathbf{a}^*\|^2} - 1}_{\zeta_a} - \frac{\pi-1}{\pi+k-1}\frac{(\mathbf{1}^\top \mathbf{a}^*)^2}{\|\mathbf{a}^*\|^2}\left(1 - \eta\frac{\pi+k-1}{2\pi}\right)^s.$$

We know $\zeta_a > 0$ due to the condition (D.8). Under $\mathcal{A}_{t-1} \cap \mathcal{E}_{\text{good}} \cap \{t-1 \leq \min\{\tau_-(\tilde{\phi}^u), \widetilde{O}(\eta^{-2})\}\}$, we can guarantee

$$\mathbf{a}_t^\top \mathbf{a}^* \geq \frac{\zeta_a \|\mathbf{a}^*\|^2}{\pi-1}\left[1 - \left(1 - \frac{\eta(\pi-1)}{2\pi}\right)^t\right] - 5K_a^2\left(1 - \frac{\eta(\pi-1)}{2\pi}\right)^t$$

$$- 2cK_a^2\sqrt{\frac{\eta \log(1/\eta\delta)}{N}} - \frac{1 - \left(\frac{2\pi-\eta(k+\pi-1)}{2\pi-\eta(\pi-1)}\right)^t}{k}\left(1 - \eta\frac{\pi-1}{2\pi}\right)^t\frac{(\pi-1)(\mathbf{1}^\top \mathbf{a}^*)^2}{\pi+k-1}$$

$$\geq \frac{\zeta_a\|\mathbf{a}^*\|^2}{\pi-1} - \left(1 - \frac{\eta(\pi-1)}{2\pi}\right)^t\left(6\|\mathbf{a}^*\|^2 + \frac{\zeta_a\|\mathbf{a}^*\|^2}{\pi-1}\right) - 2cK_a^2\sqrt{\frac{\eta k \log(1/\eta\delta)}{N}},$$

where we used $|\mathbf{a}_{t_0}^\top \mathbf{a}^*| \leq \|\mathbf{a}^*\| \cdot K_a = 5\|\mathbf{a}^*\|^2$. Now let $t = t_0 + \tau_{1,3}$, under $\mathcal{A}_{t-1} \cap \mathcal{E}_{\text{good}} \cap \{t_0 + \tau_{1,3} - 1 \leq \tau_-(\tilde{\phi}^u)\}$, we have

$$\mathbf{a}_{t_0+\tau_{1,3}}^\top \mathbf{a}^* \geq \frac{3}{4}\frac{\zeta_a\|\mathbf{a}^*\|^2}{\pi-1} - 2cK_a^2\sqrt{\frac{\eta k \log(1/\eta\delta)}{N}}$$

$$\geq \frac{24(\pi-1)(\mathbf{1}^\top \mathbf{a}^*)^2}{k(\pi+k-1)} - 2cK_a^2\sqrt{\frac{\eta k \log(1/\eta\delta)}{N}}$$

$$\geq \frac{16(\pi-1)(\mathbf{1}^\top \mathbf{a}^*)^2}{k(\pi+k-1)} = \gamma_a, \tag{D.19}$$

where we used the assumption (D.8) and (D.6). Since $\mathcal{Z}_3 = t_0 + \tau_{1,3} - 1 \leq \tau_-(\tilde{\phi}^u)$, we have

$$\mathbb{P}\left(\mathcal{Z}_1^c \cap \mathcal{Z}_2^c \cap \mathcal{Z}_3 \cap \{\tau_a > \tau_{1,1} + \tau_{1,2} + \tau_{1,3}\}\right) \leq 5\delta. \tag{D.20}$$

If $\tau_-(\tilde{\phi}^u) < \tau_-(\pi/2) + \tau_{1,3}$ (that is $\mathcal{Z}_3^c$), we regard $(\mathbf{v}_{\tau_-(\tilde{\phi}^u)}, \mathbf{a}_{\tau_-(\tilde{\phi}^u)})$ as the new initial point and step into the analysis of the next region.

**Initial region:** $[\tilde{\phi}^u, \pi)$. Now we assign $t_0 = \tau_-(\tilde{\phi}^u)$ and choose $\tau_{1,4} = \tau_{1,3}$. Recall that $\cos\tilde{\phi}^u = -\frac{144\pi}{\pi+k-1}\frac{(\mathbf{1}^\top \mathbf{a}^*)^2}{k}\log(4+k^2)$. Similar to what we showed in (D.17), for any $t_0 \leq s \leq t-1 \leq t_0 + \tau_{1,4} - 1$, under the event $\mathcal{A}_{t-1} \cap \mathcal{E}_{\text{good}}$, we have,

$$
\cos\phi_s = \frac{1}{\|\mathbf{v}_s\|}\left\{ \mathbf{v}_{t_0}^\top \mathbf{w}^* + \frac{\eta}{2\pi}\frac{1}{N}\sum_{i=1}^{N}\sum_{m=t_0}^{s-1}\frac{\pi - \phi_m^i}{\|\mathbf{v}_m\|}(\mathbf{a}_m^i)^\top \mathbf{a}^* \sin^2\phi_m - \eta\sum_{m=t_0}^{s-1}(\mathbf{h}_m^\top \mathbf{w}^* + \boldsymbol{\xi}_m^\top \mathbf{w}^*) \right\}
$$

$$
\overset{(i)}{\leq} \frac{1}{\|\mathbf{v}_s\|}\left\{ \frac{\cos\phi_{t_0}}{3} + \tau_{1,4}\eta\gamma_a + c\tau_{1,4}\eta^2(I + \sqrt{k}I)\sqrt{\log(k/\eta\delta)}K_a^2 + \eta\left|\sum_{m=t_0}^{s-1}(\boldsymbol{\xi}_m^\top \mathbf{w}^* + \mathbf{h}_m^\top \mathbf{w}^*)\right| \right\}
$$

$$
\leq \frac{1}{\|\mathbf{v}_s\|}\left\{ \frac{\cos\tilde{\phi}^u}{3} + \frac{36\pi}{\pi+k-1}\frac{(\mathbf{1}^\top \mathbf{a}^*)^2}{k}\log(4+k^2) \right\}
$$

$$
\overset{(ii)}{\leq} -\frac{4\pi}{\pi+k-1}\frac{(\mathbf{1}^\top \mathbf{a}^*)^2}{k}\log(4+k^2). \tag{D.21}
$$

where $(i)$ holds due to $\|\mathbf{v}_m\| \leq 3$, $\phi_0 > \pi/2$ and the definition of $\mathcal{A}_{t-1}$; and $(ii)$ holds due to $\|\mathbf{v}_s\| \leq 3$. Applying the bound $|\cos\phi_s - \cos\phi_s^i|$ in (D.4) and the assumption D.6, we have

$$
\max_i \cos\phi_s^i \leq \cos\phi_s + \max_i |\cos\phi_s - \cos\phi_s^i|
$$

$$
\leq -\frac{4\pi}{\pi+k-1}\frac{(\mathbf{1}^\top \mathbf{a}^*)^2}{k}\log(4+k^2) + 12c\eta K_a^2(\sqrt{I d} + I)\sqrt{\log(Nd/\delta)} \leq 0,
$$

which means $\min_i \phi_s^i \geq \frac{\pi}{2}$. It follows that for any $s \leq t-1$,

$$
\frac{1}{N}\sum_{i=1}^{N}B_s^i = \frac{1}{N}\sum_{i=1}^{N}\left\{ (g(\phi_s^i) - 1) + \frac{\eta}{2\pi}\frac{(\mathbf{1}^\top \mathbf{a}^*)^2}{\|\mathbf{a}^*\|^2}\sum_{l=t_0}^{s-1}\left(1 - \eta\frac{\pi+k-1}{2\pi}\right)^{s-1-l}(\pi - g(\phi_l^i)) \right\}
$$

$$
\geq \frac{1}{N}\sum_{i=1}^{N}\left\{ -1 + \frac{\eta}{2\pi}\frac{(\mathbf{1}^\top \mathbf{a}^*)^2}{\|\mathbf{a}^*\|^2}\sum_{l=t_0}^{s-1}\left(1 - \eta\frac{\pi+k-1}{2\pi}\right)^{s-1-l}(\pi - 1) \right\}
$$

$$
= \zeta_a - \frac{\pi-1}{\pi+k-1}\frac{(\mathbf{1}^\top \mathbf{a}^*)^2}{\|\mathbf{a}^*\|^2}\left(1 - \eta\frac{\pi+k-1}{2\pi}\right)^s. \tag{D.22}
$$

Plugging (D.22) and (D.3) into (D.1), under the event $\mathcal{A}_{t-1} \cap \mathcal{E}_{\text{good}}$, we can get

$$
\mathbf{a}_t^\top \mathbf{a}^* \geq \frac{\zeta_a\|\mathbf{a}^*\|^2}{\pi-1}\left[1 - \left(1 - \frac{\eta(\pi-1)}{2\pi}\right)^t\right] - |\mathbf{a}_0^\top \mathbf{a}^*|\left(1 - \frac{\eta(\pi-1)}{2\pi}\right)^t
$$

$$
- \frac{1 - \left(\frac{2\pi - \eta(k+\pi-1)}{2\pi - \eta(\pi-1)}\right)^t}{k}\left(1 - \eta\frac{\pi-1}{2\pi}\right)^t\frac{(\pi-1)(\mathbf{1}^\top \mathbf{a}^*)^2}{\pi+k-1}
$$

$$
- 2cK_a^2\sqrt{\frac{M\eta\log(1/\eta\delta)}{N}}. \tag{D.23}
$$

Plugging $t = t_0 + \tau_{1,4}$, according to (D.23) and (D.19), we can guarantee that $\phi_{\tau_a} \geq \pi/2$ and

$$
\mathbb{P}\left( \mathcal{Z}_1^c \cap \mathcal{Z}_2^c \cap \mathcal{Z}_3^c \cap \left\{ \tau_a > \sum_{q=1}^{4}\tau_{1,q} \right\} \right) \leq 5\delta. \tag{D.24}
$$

**Conclusion.** Combining (D.10), (D.15), (D.20) and (D.24), we have

$$
\mathbb{P}\left( \tau_a > \sum_{q=1}^{4}\tau_{1,q} \right) \leq \mathbb{P}\left(\{\tau_a > \tau_{1,1}\} \cap \mathcal{Z}_1\right) + \mathbb{P}\left( \left\{ \tau_a > \sum_{q=1}^{4}\tau_{1,q} \right\} \cap \mathcal{Z}_1^c \right)
$$

$$
\leq 4\delta + \mathbb{P}\left( \left\{ \tau_a > \sum_{q=1}^{4}\tau_{1,q} \right\} \cap \mathcal{Z}_1^c \cap \mathcal{Z}_2 \right) + \mathbb{P}\left( \left\{ \tau_a > \sum_{q=1}^{2}\tau_{1,q} \right\} \cap \mathcal{Z}_1^c \cap \mathcal{Z}_2^c \right)
$$

$$\leq 9\delta + \mathbb{P}\left(\left\{\tau_a > \sum_{q=1}^{3}\tau_{1,q}\right\} \cap \mathcal{Z}_1^c \cap \mathcal{Z}_2^c \cap \mathcal{Z}_3\right) + \mathbb{P}\left(\left\{\tau_a > \sum_{q=1}^{4}\tau_{1,q}\right\} \cap \mathcal{Z}_1^c \cap \mathcal{Z}_2^c \cap \mathcal{Z}_3^c\right)$$

$$\leq 14\delta + \mathbb{P}\left(\left\{\tau_a > \sum_{q=1}^{4}\tau_{1,q}\right\} \cap \mathcal{Z}_1^c \cap \mathcal{Z}_2^c \cap \mathcal{Z}_3^c\right)$$

$$\leq 19\delta.$$

Therefore, we have proved $\tau_a \leq \sum_{q=1}^{4}\tau_{1,q}$ with high probability for $\phi_0 \in [0, \tilde{\phi}^o)$. The conclusion for other initial regions can be obtained in similar arguments. From the definitions in (D.9), (D.13) and (D.16), we have

$$\sum_{q=1}^{4}\tau_{1,q} \lesssim \eta^{-1}\log k.$$

$\square$

### D.2 Proof of Lemma 3

**Lemma 3 restated.** Denote $\tilde{\phi}^u = \arccos\left(-\frac{144\pi}{\pi+k-1}\frac{(\mathbf{1}^\top\boldsymbol{a}^*)^2}{k}\log(4+k^2)\right)$ and

$$\varpi = \min\left\{\sin\tilde{\phi}^u\mathbb{1}_{\phi_0\leq\tilde{\phi}^u} + \sin\phi_0\mathbb{1}_{\phi_0>\tilde{\phi}^u}, \; \sin\left(\pi - \left((20\|\boldsymbol{a}^*\|^2)^{-1}\wedge 1\right)\right)\right\},$$

Under the same conditions of Theorem 1. If the learning rate satisfies

$$c\|\boldsymbol{a}^*\|^2\sqrt{\log(Ndk/\delta)}\max\left\{\eta(\sqrt{Id}+I), \sqrt{\frac{\eta}{N}}\right\} \leq \frac{\varpi^2}{16k}, \tag{D.25}$$

for a large absolute constant $c > 0$, we have $\sin\phi_{\tau_a} \geq \frac{\varpi}{15\sqrt{k}}$ with probability at least $1 - \delta$.

*Proof.* If $\phi_{\tau_a} \leq \tilde{\phi}^u$, the lower bound trivially holds. Next we prove the lower bound for $\sin\phi_{\tau_a}$ starting from the last initial region, that is $\phi_0 \in [\tilde{\phi}^u, \pi)$. From (D.16) and $|\mathbf{1}^\top\boldsymbol{a}^*| \geq \|\boldsymbol{a}^*\|$, we know that

$$\tau_{1,4} = \frac{1}{\eta}\log\left(4 + \frac{16(\pi-1)}{\zeta_a}\right) \leq \frac{1}{\eta}\log\left(4 + \frac{16(\pi-1)}{\frac{32(\pi-1)^2(\mathbf{1}^\top\boldsymbol{a}^*)^2}{k(\pi+k-1)\|\boldsymbol{a}^*\|^2}}\right) \leq \frac{1}{\eta}\log(4+k^2).$$

Let $C = \frac{\log(1+2k^2)}{4}$, then $e^{2C} \leq (1+2k^2)^{\frac{1}{2}} \leq 2k$. Denote $\varpi_0 = \sin\tilde{\phi}^u\mathbb{1}_{\phi_0<\tilde{\phi}^u} + \sin\phi_0\mathbb{1}_{\phi_0\geq\tilde{\phi}^u}$. From the definition of $\boldsymbol{h}_t$ and (B.9), we know

$$\|\boldsymbol{h}_t\| \leq \frac{1}{N}\sum_{i=1}^{N}\|\mathbf{P}_t^i - \mathbf{P}_t\|\|\nabla_{\boldsymbol{w}}L(\boldsymbol{w}_t^i, \boldsymbol{a}_t^i)\| \leq 50c\eta(\sqrt{Id}+I)\sqrt{\log(Nd/\delta)}K_a^4, \tag{D.26}$$

where we also used $\|\nabla_{\boldsymbol{w}}L(\boldsymbol{w}_t^i, \boldsymbol{a}_t^i)\| \leq 5K_a^2$. Recall that $\check{\boldsymbol{v}}_t - \boldsymbol{v}_t = \eta\boldsymbol{h}_{t-1}$, then it follows that

$$\begin{aligned}
\left|\|\check{\boldsymbol{v}}_t\|^2\sin^2\check{\phi}_t - \|\boldsymbol{v}_t\|^2\sin^2\phi_t\right| \\
&= \left|\|\check{\boldsymbol{v}}_t\|^2 - \|\boldsymbol{v}_t\|^2 - \left((\check{\boldsymbol{v}}_t^\top\boldsymbol{w}^*)^2 - (\boldsymbol{v}_t^\top\boldsymbol{w}^*)^2\right)\right| \\
&= \left|(\|\check{\boldsymbol{v}}_t\| + \|\boldsymbol{v}_t\|)(\|\check{\boldsymbol{v}}_t\| - \|\boldsymbol{v}_t\|)\right| + \left|(|\check{\boldsymbol{v}}_t^\top\boldsymbol{w}^*| - |\boldsymbol{v}_t^\top\boldsymbol{w}^*|)\left(|\check{\boldsymbol{v}}_t^\top\boldsymbol{w}^*| + |\boldsymbol{v}_t^\top\boldsymbol{w}^*|\right)\right| \\
&\leq 7\|\check{\boldsymbol{v}}_t - \boldsymbol{v}_t\| + 7\left|(\check{\boldsymbol{v}}_t - \boldsymbol{v}_t)^\top\boldsymbol{w}^*\right|^2 \\
&\leq 14\eta\|\boldsymbol{h}_{t-1}\|,
\end{aligned} \tag{D.27}$$

where we used $\|\boldsymbol{v}_t\| \leq 3$ in Lemma 13 and $\|\check{\boldsymbol{v}}_t\| \leq \|\boldsymbol{v}_t\| + \eta\|\boldsymbol{h}_t\| < 4$. For any $t$, we define a variable

$$\tau_t(K_a) = \sup_{0\leq s\leq t-1}\left\{\min_{i\in[N]}\phi_s^i \leq \pi - \left(\frac{1}{20K_a^2}\wedge 1\right)\right\}. \tag{D.28}$$

If $\tau_t(K_a)$ does not exist, that is $\min_{i\in[N]}\phi_s^i > \pi - \left((20K_a^2)^{-1}\wedge 1\right)$ holds for any $0 \le s \le t-1$, we let $\tau_t(K_a) = 0$. Then for any $\tau_t(K_a) \le s \le t-1$, we always have $\min_{i\in[N]}\phi_s^i > \pi - \left((20K_a^2)^{-1}\wedge 1\right)$ such that

$$
\begin{aligned}
1 - \eta\lambda_s\cos\phi_s = 1 - \eta\cos\phi_s \cdot \frac{1}{N}\sum_{i=1}^{N}\frac{\pi - \phi_s^i}{2\pi}\frac{(\boldsymbol{a}_s^i)^\top\boldsymbol{a}^*}{\|\boldsymbol{v}_s\|} \\
\ge 1 - \eta\frac{1}{N}\sum_{i=1}^{N}\frac{1}{2\pi\cdot(20K_a^2\vee 1)}\frac{|(\boldsymbol{a}_s^i)^\top\boldsymbol{a}^*|}{\|\boldsymbol{v}_s\|} \\
\overset{(ii)}{\ge} 1 - \eta\frac{5K_a^2}{2\pi\cdot(20K_a^2\vee 1)} \ge 1 - \frac{\eta}{8}.
\end{aligned}
\tag{D.29}
$$

Without loss of generality, we assume $\arg\min_{i\in[N]}\phi_{\tau_t(K_a)}^i = 1$. It follows that

$$
\begin{aligned}
\sin^2\phi_{\tau_t(K_a)} &\ge \sin^2\phi_{\tau_t(K_a)}^1 - \left|\sin^2\phi_{\tau_t(K_a)} - \sin^2\phi_{\tau_t(K_a)}^1\right| \\
&\ge \sin^2\phi_{\tau_t(K_a)}^1 - 2\left|\cos\phi_{\tau_t(K_a)} - \cos\phi_{\tau_t(K_a)}^1\right| \\
&\ge \sin^2\phi_{\tau_t(K_a)}^1 - \frac{2\|\boldsymbol{v}_{\tau_t(K_a)}^1 - \boldsymbol{v}_{\tau_t(K_a)}\|}{\|\boldsymbol{v}_{\tau_t(K_a)}\|} - 2\|\boldsymbol{v}_{\tau_t(K_a)}^1\|\left|\frac{1}{\|\boldsymbol{v}_{\tau_t(K_a)}\|} - \frac{1}{\|\boldsymbol{v}_{\tau_t(K_a)}^1\|}\right| \\
&\ge \sin^2(\pi - ((20K_a^2)^{-1}\wedge 1)) - 6c\eta(\sqrt{Id}+I)\sqrt{\log(Nd/\delta)}K_a^2 \\
&\ge \frac{\sin^2(\pi - ((20K_a^2)^{-1}\wedge 1))}{2}.
\end{aligned}
\tag{D.30}
$$

Notice that if $t \le 4C/\eta$, we have

$$
\left(1 - \frac{\eta}{8}\right)^{2t} \ge \left(1 - \frac{\eta}{8}\right)^{\frac{8}{\eta}\cdot C} \ge 2^{-2C} \ge e^{-2C} \ge \frac{1}{2k}.
\tag{D.31}
$$

It holds because $\eta/8 \le \frac{1}{2}$ and $f(x) = (1-x)^{1/x}$ is decreasing in $[0,1]$. Invoking Lemma 1, we know

$$
\begin{aligned}
\|\boldsymbol{v}_t\|^2\sin^2\phi_t &= (1 - \eta\lambda_{t-1}\cos\phi_{t-1})^2\|\boldsymbol{v}_{t-1}\|^2\sin^2\phi_{t-1} - 2\eta M_{1,t-1} + \eta^2 M_{2,t-1} + H_t \\
&\overset{(i)}{\ge} \left(1 - \frac{\eta}{8}\right)^{2(t-\tau_t(K_a))}\|\boldsymbol{v}_{\tau_t(K_a)}\|^2\sin^2\phi_{\tau_t(K_a)} - 2\eta\sum_{s=\tau_t(K_a)}^{t-1}\left(1-\frac{\eta}{8}\right)^{2(t-1-s)}(M_{1,s} + 7\|\boldsymbol{h}_s\|) \\
&\ge \left(1 - \frac{\eta}{8}\right)^{2t}\frac{1}{4}\min\left\{\sin^2\phi_{\tau_t(K_a)}, \varpi_0^2\right\} - 2\eta\sum_{s=\tau_t(K_a)}^{t-1}\left(1-\frac{\eta}{8}\right)^{2(t-1-s)}(M_{1,s} + 7\|\boldsymbol{h}_s\|) \\
&\overset{(ii)}{\ge} \left(1 - \frac{\eta}{8}\right)^{2t}\frac{1}{4}\min\left\{\frac{\sin^2\left[\pi - ((20K_a^2)^{-1}\wedge 1)\right]}{2}, \varpi_0^2\right\} \\
&\quad - 140c\eta(\sqrt{Id}+I)\sqrt{\log(Nd/\delta)}K_a^2 - 2\eta\max_{0\le\tau\le t-1}\left|\sum_{s=\tau}^{t-1}\left(1-\frac{\eta}{8}\right)^{2(t-1-s)}M_{1,s}\right| \\
&\overset{(iii)}{\ge} \frac{1}{2k}\cdot\frac{1}{4}\min\left\{\frac{\sin^2\left[\pi - ((20K_a^2)^{-1}\wedge 1)\right]}{2}, \varpi_0^2\right\} - 140c\eta(\sqrt{Id}+I)\sqrt{\log(Nd/\delta)}K_a^2 \\
&\quad - \frac{12cK_a^2\eta}{N}(3 + 5\eta K_a^2)\sqrt{\log(1/\eta\delta)\sum_{s=0}^{t-1}\sum_{i=1}^{N}\left(1-\frac{\eta}{8}\right)^{2(t-1-s)}} \\
&\overset{(iv)}{\ge} \frac{\min\left\{\sin^2\left[\pi - ((20K_a^2)^{-1}\wedge 1)\right], \varpi_0^2\right\}}{25k},
\end{aligned}
$$

where $(i)$ holds due to (D.27) and (D.29); $(ii)$ follows from (D.30) and (D.26); $(iii)$ comes from (D.31) and Lemma 11; and $(iv)$ holds due to the assumption D.25. Plugging the choice of $C =$

$\log(1 + 2k^2)/4$, together with $\|\boldsymbol{v}_t\| \leq 3$, we can prove the conclusion

$$\sin^2 \phi_{\tau_a} \geq \frac{\min\left\{\sin^2\left[\pi - \left((20K_a^2)^{-1} \wedge 1\right)\right], \varpi_0^2\right\}}{225k} \geq \frac{\varpi^2}{225k}.$$

$\square$

### D.3 Self-correction of the First Layer

**Theorem 2 restated.** For the initial point $(\boldsymbol{v}_0, \boldsymbol{a}_0)$ with $\boldsymbol{a}_0^\top \boldsymbol{a}^* \geq \gamma_a$ and $\sin\phi_0 > 0$, we denote $\tau_v = \inf\{t \geq 0 : \phi_v \leq \tilde{\phi}^l\}$, where $\cos\tilde{\phi}^l = \frac{144\pi}{\pi+k-1} \frac{(\mathbf{1}^\top \boldsymbol{a}^*)^2}{k} \log(4 + k^2)$. If the learning rate satisfies

$$c\|\boldsymbol{a}^*\|^2 \sqrt{\log(Ndk/\delta)} \max\left\{\eta(\sqrt{Id}+I), \sqrt{\frac{\eta}{N}}\right\} \leq \sin^2\phi_0, \tag{D.32}$$

for a large absolute constant $c > 0$, it holds that $\tau_v \leq O\left(\frac{1}{\eta} \frac{k^2}{\sin^3\phi_0}\right)$ and $\boldsymbol{a}_{\tau_v}^\top \boldsymbol{a}^* \geq \gamma_a/32$ with probability at least $1 - \delta$.

*Proof.* Find $\tilde{\phi}^f \in (\pi/2, \pi)$ such that $\cos\tilde{\phi}^f = -\frac{\pi}{\pi+k-1} \frac{(\mathbf{1}^\top \boldsymbol{a}^*)^2}{k} \log(4 + k^2)$. We introduce the following stopping time w.r.t. the angle $\phi$ during the training process,

$$\tau_+(\phi) = \inf_{t \geq 0} \{t : \phi_t \leq \phi\} \quad \text{for} \quad \phi \in [0, \pi].$$

**Initial angle: $\phi_0 \in (\tilde{\phi}^f, \pi)$.** For any $s \leq \tau_+(\tilde{\phi}^f)$, under the event $\mathcal{E}_{\text{good}}$, we have

$$\begin{aligned}
\cos\phi_s^i &\leq \cos\phi_s + \left|\cos\phi_s^i - \cos\phi_s\right| \\
&\leq \cos\tilde{\phi}^f + 12c\eta K_a^2(\sqrt{Id}+I)\sqrt{\log(Nd/\delta)} \\
&\leq -\frac{\pi}{\pi+k-1} \frac{(\mathbf{1}^\top \boldsymbol{a}^*)^2}{k} \log(4 + k^2) + \frac{\pi}{\pi+k-1} \frac{(\mathbf{1}^\top \boldsymbol{a}^*)^2}{k} \log(4 + k^2) = 0.
\end{aligned}$$

It means that $\phi_s^i \geq \pi/2$ holds for any $0 \leq s \leq \min\{\tau_+(\tilde{\phi}^f), \widetilde{O}(\eta^{-2})\}$. As a consequence, we have

$$\frac{1}{N} \sum_{i=1}^N B_s^i \geq \zeta_a - \frac{\pi-1}{\pi+k-1} \frac{(\mathbf{1}^\top \boldsymbol{a}^*)^2}{\|\boldsymbol{a}^*\|^2} \left(1 - \eta\frac{\pi+k-1}{2\pi}\right)^s.$$

Notice that if $t \leq \frac{1}{\eta} \frac{2\pi}{\pi-1}$, we have

$$\left(1 - \eta\frac{\pi-1}{2\pi}\right)^t \geq \left(1 - \eta\frac{\pi-1}{2\pi}\right)^{\frac{1}{\eta}\frac{2\pi}{\pi-1}} \geq \frac{1}{4},$$

which holds because $\eta\frac{\pi-1}{2\pi} \leq \frac{1}{2}$ and $(1-x)^{1/x}$ is decreasing in $[0,1]$. If $t \geq \frac{1}{\eta}\frac{2\pi}{\pi-1}$, we also have

$$1 - \left(1 - \eta\frac{\pi-1}{2\pi}\right)^t \geq 1 - \left(1 - \eta\frac{\pi-1}{2\pi}\right)^{\frac{1}{\eta}\frac{2\pi}{\pi-1}} \geq 1 - e^{-1} \geq \frac{1}{4},$$

because $(1-x)^{1/x} \leq e^{-1}$ for $x \in [0,1]$. Together with the dynamic (D.1) and $\boldsymbol{a}_0^\top \boldsymbol{a}^* \geq \gamma_a$, under the event $\mathcal{E}_{\text{good}}$, we have

$$\begin{aligned}
\boldsymbol{a}_s^\top \boldsymbol{a}^* &\geq \frac{\zeta_a\|\boldsymbol{a}^*\|^2}{2(\pi-1)}\left[1 - \left(1 - \frac{\eta(\pi-1)}{2\pi}\right)^s\right] - 2c\sqrt{\frac{\eta\log(2t/\delta)}{N}}K_a^2 \\
&\quad + \left(\gamma_a - \frac{(\pi-1)(\mathbf{1}^\top \boldsymbol{a}^*)^2}{k(\pi+k-1)}\right)\left(1 - \frac{\eta(\pi-1)}{2\pi}\right)^s \\
&\overset{(i)}{\geq} \frac{1}{4}\frac{32(\pi-1)}{k(\pi+k-1)} + \frac{1}{4}\left(\gamma_a - \frac{(\pi-1)(\mathbf{1}^\top \boldsymbol{a}^*)^2}{k(\pi+k-1)}\right) - 3c\sqrt{\frac{\eta k\log(1/\eta\delta)}{N}}K_a^2
\end{aligned}$$

$$\overset{(ii)}{=} \frac{\gamma_a}{2} + \frac{1}{4}\left(\gamma_a - \frac{\gamma_a}{16}\right) - 3c\sqrt{\frac{\eta k \log(1/\eta\delta)}{N}} K_a^2$$

$$\overset{(iii)}{\geq} \frac{\gamma_a}{4}, \tag{D.33}$$

where $(i)$ holds due to (D.8); $(ii)$ holds since $\gamma_a = \frac{16(\pi-1)}{k(\pi+k-1)}$ and $(iii)$ follows from the condition (D.6). Under the event $\mathcal{E}_{\text{good}}$, we know for any $s \leq \min\{\tau_+(\tilde{\phi}^f), \widetilde{O}(\eta^{-2})\}$

$$(\boldsymbol{a}_s^i)^\top \boldsymbol{a}^* \geq \boldsymbol{a}_s^\top \boldsymbol{a}^* - \left|\boldsymbol{a}_s^\top \boldsymbol{a}^* - (\boldsymbol{a}_s^i)^\top \boldsymbol{a}^*\right|$$

$$\geq \frac{\gamma_a}{4} - cK_a\eta(\sqrt{Ik} + I)\sqrt{\log(k/\delta)}$$

$$= \frac{4(\pi-1)}{k(\pi+k-1)} - cK_a\eta(\sqrt{Ik} + I)\sqrt{\log(k/\delta)}$$

$$\geq \frac{\gamma_a}{8}, \tag{D.34}$$

holds with probability at least $1 - \delta$. Recall that $\check{\boldsymbol{v}}_t - \boldsymbol{v}_t = \eta \boldsymbol{h}_{t-1}$, then it follows that

$$\left|\|\check{\boldsymbol{v}}_t\|^2 \sin^2 \check{\phi}_t - \|\boldsymbol{v}_t\|^2 \sin^2 \phi_t\right| = \left|\|\check{\boldsymbol{v}}_t\|^2 - \|\boldsymbol{v}_t\|^2 - ((\check{\boldsymbol{v}}_t^\top \boldsymbol{w}^*)^2 - (\boldsymbol{v}_t^\top \boldsymbol{w}^*)^2)\right|$$

$$= \left|2\eta \boldsymbol{v}_t^\top \boldsymbol{h}_{t-1} + \eta^2 \|\boldsymbol{h}_{t-1}\|^2 - 2\eta(\boldsymbol{v}_t^\top \boldsymbol{w}^*)(\boldsymbol{h}_{t-1}^\top \boldsymbol{w}^*) - \eta^2(\boldsymbol{h}_{t-1}^\top \boldsymbol{w}^*)^2\right|$$

$$\leq 2\eta \left|\boldsymbol{v}_t^\top \left(\mathbf{I} - \boldsymbol{w}^*(\boldsymbol{w}^*)^\top\right) \boldsymbol{h}_{t-1}\right| + \eta^2 \boldsymbol{h}_{t-1}^\top \left(\mathbf{I} - \boldsymbol{w}^*(\boldsymbol{w}^*)^\top\right) \boldsymbol{h}_{t-1}$$

$$\leq 2\eta \left\|\left(\mathbf{I} - \boldsymbol{w}^*(\boldsymbol{w}^*)^\top\right) \boldsymbol{v}_t\right\| \|\boldsymbol{h}_{t-1}\| + \eta^2 \boldsymbol{h}_{t-1}^\top \left(\mathbf{I} - \boldsymbol{w}^*(\boldsymbol{w}^*)^\top\right) \boldsymbol{h}_{t-1}$$

$$\overset{(i)}{=} 2\eta \left(\boldsymbol{v}_t^\top \left(\mathbf{I} - \boldsymbol{w}^*(\boldsymbol{w}^*)^\top\right) \boldsymbol{v}_t\right)^{1/2} \|\boldsymbol{h}_{t-1}\| + \eta^2 \boldsymbol{h}_{t-1}^\top \left(\mathbf{I} - \boldsymbol{w}^*(\boldsymbol{w}^*)^\top\right) \boldsymbol{h}_{t-1}$$

$$= 2\eta \left(\|\boldsymbol{v}_t\|^2 - (\boldsymbol{v}_t^\top \boldsymbol{w}^*)^2\right)^{1/2} \|\boldsymbol{h}_{t-1}\| + \eta^2 \boldsymbol{h}_{t-1}^\top \left(\mathbf{I} - \boldsymbol{w}^*(\boldsymbol{w}^*)^\top\right) \boldsymbol{h}_{t-1}$$

$$\overset{(ii)}{=} 2\eta \sin \phi_t \|\boldsymbol{v}_t\| \|\boldsymbol{h}_{t-1}\| + \eta^2 \|\boldsymbol{h}_{t-1}\|^2. \tag{D.35}$$

where $(i)$ holds due to $\mathbf{I} - \boldsymbol{w}^*(\boldsymbol{w}^*)^\top$ is idempotent; and $(ii)$ holds due to $\|\boldsymbol{w}^*\| = 1$. Invoking Lemma 1 and (D.26), for any $s \leq t - 1 \leq \min\{\tau_+(\tilde{\phi}^f), \widetilde{O}(\eta^{-2})\}$, under the event $\mathcal{E}_{\text{good}}$, we can get

$$\|\boldsymbol{v}_s\|^2 \sin^2 \phi_s = (1 - \eta\lambda_{s-1}\cos\phi_{s-1})^2 \|\boldsymbol{v}_{s-1}\|^2 \sin^2 \phi_{s-1} - 2\eta M_{1,s-1}$$

$$+ \eta^2 M_{2,s-1} + \|\boldsymbol{v}_s\|^2 \sin^2 \phi_s - \|\check{\boldsymbol{v}}_s\|^2 \sin^2 \check{\phi}_s$$

$$\overset{(i)}{\geq} \|\boldsymbol{v}_{s-1}\|^2 \sin^2 \phi_{s-1} - 2\eta M_{1,s-1} - 6\eta\|\boldsymbol{h}_{s-1}\| - \eta^2\|\boldsymbol{h}_{s-1}\|^2$$

$$\geq \|\boldsymbol{v}_0\|^2 \sin^2 \phi_0 - 2\eta \sum_{l=0}^{s-1} M_{1,l} - \eta \sum_{l=0}^{s-1}(6\|\boldsymbol{h}_l\| + \eta\|\boldsymbol{h}_l\|^2)$$

$$\geq \frac{\sin^2 \phi_0}{3} - 450c\eta(\sqrt{Id} + I)\sqrt{\log(Nd/\delta)} K_a^2 - 24cK_a\left(3 + 5\eta K_a^2\right)\sqrt{\frac{\eta \log(1/\delta)}{N}}$$

$$\geq \frac{\sin^2 \phi_0}{6},$$

where $(i)$ holds due to (D.34) and $\cos \phi_s < 0$ for $s \leq \tau_+(\tilde{\phi}^f)$; $(ii)$ follows from assumption (D.32). It implies that $\sin^2 \phi_s \geq \sin^2 \phi_0/54$ for any $s \leq t - 1 \leq \min\{\tau_+(\tilde{\phi}^f), \widetilde{O}(\eta^{-2})\}$. Then we can lower bound $\sin^2 \phi_s^i$ by

$$\sin^2 \phi_s^i \geq \sin^2 \phi_s - \left|\sin^2 \phi_s^i - \sin^2 \phi_s\right|$$

$$\geq \frac{\sin^2 \phi_0}{54} - 2\left|\cos \phi_s^i - \cos \phi_s\right|$$

$$\geq \frac{\sin^2 \phi_0}{54} - 4c\eta K_a^2(\sqrt{Id} + I)\sqrt{\log(Nd/\delta)}$$

$$\geq \frac{\sin^2 \phi_0}{64}.$$

Together with the dynamic (D.12) and lower bound (D.34), we can guarantee

$$\cos\phi_t = \frac{1}{\|\boldsymbol{v}_t\|}\left\{\boldsymbol{v}_0^\top\boldsymbol{w}^* + \eta\sum_{m=t_0}^{t-1}\frac{1}{N}\sum_{i=1}^N\frac{\pi - \phi_m^i}{2\pi}\frac{(\boldsymbol{a}_m^i)^\top\boldsymbol{a}^*}{\|\boldsymbol{v}_m\|}\sin^2\phi_m - \eta\sum_{m=t_0}^{t-1}(\boldsymbol{\xi}_m^\top\boldsymbol{w}^* + \boldsymbol{h}_m^\top\boldsymbol{w}^*)\right\}$$

$$\geq \frac{1}{\|\boldsymbol{v}_t\|}\left\{-3 + \eta t\cdot\frac{\gamma_a\sin^2\phi_0}{288}\cdot\frac{\arcsin(\sin\phi_0/6)}{2\pi}\right.$$

$$\left. - c\eta K_a^2(\sqrt{Id} + I)\sqrt{\log(Nd/\delta)} - cK_a\sqrt{\frac{\eta\log(Nd/\eta\delta)}{N}}\right\}. \qquad \text{(D.36)}$$

Now we take

$$\tau_{2,1} = \frac{1}{\eta}\cdot\frac{5429}{\gamma_a\sin^2\phi_0}\frac{1}{\arcsin(\sin\phi_0/6)} \leq \frac{40000}{\eta\gamma_a}\frac{1}{\sin^3\phi_0},$$

where the inequality holds since $\arcsin(\sin\phi_0/6) - \arcsin(0) \geq \sin\phi_0/6$ due to $\arcsin'(x) = \frac{1}{\sqrt{1-x^2}}$. Plugging it into (D.36), we have

$$\cos\phi_{\tau_{2,1}} \geq \frac{1}{\|\boldsymbol{v}_t\|}(-3 + 4 - 1) > \cos\tilde{\phi}^f,$$

which implies

$$\mathbb{P}\left(\tau_{2,1} \leq \tau_+(\tilde{\phi}^f)\right) = \mathbb{P}\left(\left\{\tau_{2,1} - 1 \leq \tau_+(\tilde{\phi}^f)\right\} \cap \left\{\cos\phi_{\tau_{2,1}} \leq \cos\tilde{\phi}^f\right\}\right)$$

$$\leq \mathbb{P}\left(\left\{\tau_{2,1} - 1 \leq \tau_+(\tilde{\phi}^f)\right\} \cap \mathcal{E}_{\text{good}}^c\right) \leq 5\delta.$$

**Initial angle:** $\phi_0 \in (\tilde{\phi}^l, \tilde{\phi}^f]$.  Now assign $t_0 = \tau_+(\tilde{\phi}^f)$. Denote

$$\tilde{\mathcal{A}}_s = \left\{\forall t_0 \leq m \leq s : \boldsymbol{a}_m^\top\boldsymbol{a}^* \geq \frac{\gamma_a}{16}\right\}, \quad \text{for } s \geq t_0.$$

Clearly, $\mathbb{P}(\tilde{\mathcal{A}}_{t_0}) \geq 1 - 7\delta$ due to (D.33). Similar to (D.34), we can verify that $\min_{i\in[N]}(\boldsymbol{a}_m^i)^\top\boldsymbol{a}^* \geq \gamma_a/32$ for any $m \leq s$ under $\tilde{\mathcal{A}}_s \cap \mathcal{E}_{\text{good}}$. In addition, under the event $\mathcal{E}_{\text{good}}$, for any $\ell \geq 0$ we have

$$\cos\phi_\ell = \frac{1}{\|\boldsymbol{v}_\ell\|}\left\{\boldsymbol{v}_0^\top\boldsymbol{w}^* + \eta\sum_{m=t_0}^{\ell-1}\frac{1}{N}\sum_{i=1}^N\frac{\pi - \phi_m^i}{2\pi}\frac{(\boldsymbol{a}_m^i)^\top\boldsymbol{a}^*}{\|\boldsymbol{v}_m\|}\sin^2\phi_m - \eta\sum_{m=t_0}^{\ell-1}(\boldsymbol{\xi}_m^\top\boldsymbol{w}^* + \boldsymbol{h}_m^\top\boldsymbol{w}^*)\right\}$$

$$\geq 6\cos\tilde{\phi}^f - 2c\eta K_a^2(\sqrt{Id} + I)\sqrt{\log(Nd/\delta)} - 2cK_a\sqrt{\frac{\eta\log(Nd/\eta\delta)}{N}}$$

$$\geq -\frac{6\pi}{\pi + k - 1}\frac{(\mathbf{1}^\top\boldsymbol{a}^*)^2}{k}\log(4 + k^2) - \frac{2\pi}{\pi + k - 1}\frac{(\mathbf{1}^\top\boldsymbol{a}^*)^2}{k}\log(4 + k^2)$$

$$\geq -\frac{144\pi}{\pi + k - 1}\frac{(\mathbf{1}^\top\boldsymbol{a}^*)^2}{k}\log(4 + k^2) = \cos\tilde{\phi}^u. \qquad \text{(D.37)}$$

It means that $\tilde{\phi}^l \leq \phi_\ell \leq \tilde{\phi}^f$ for any $\ell \leq s \leq \min\{\tau_+(\tilde{\phi}^l), \tilde{O}(\eta^{-2})\}$ under the event $\tilde{\mathcal{A}}_s \cap \mathcal{E}_{\text{good}}$. Hence we know (D.18) holds. Consequently, the lower bound (D.22) also holds for $\frac{1}{N}\sum_{i=1}^N B_s^i$. Therefore, similar to (D.33), we can further verify that

$$\boldsymbol{a}_{s+1}^\top\boldsymbol{a}^* \geq \frac{\zeta_a\|\boldsymbol{a}^*\|^2}{2(\pi - 1)}\left[1 - \left(1 - \frac{\eta(\pi - 1)}{2\pi}\right)^s\right] - 3c\sqrt{\frac{\eta k\log(1/\eta\delta)}{N}}K_a^2$$

$$+ \left(\frac{\gamma_a}{4} - \frac{(\pi - 1)(\mathbf{1}^\top\boldsymbol{a}^*)^2}{k(\pi + k - 1)}\right)\left(1 - \frac{\eta(\pi - 1)}{2\pi}\right)^s$$

$$\geq \frac{\gamma_a}{16},$$

where we used $\boldsymbol{a}_{t_0}^\top\boldsymbol{a}^* \geq \gamma_a/4$. This yields that for any $t_0 \leq s \leq t - 1$,

$$\tilde{\mathcal{A}}_s \cap \mathcal{E}_{\text{good}} \cap \left\{t - 1 \leq \tau_+(\tilde{\phi}^l)\right\} \subseteq \left\{\boldsymbol{a}_{s+1}^\top\boldsymbol{a}^* \geq \frac{\gamma_a}{16}\right\},$$

which implies that

$$\tilde{\mathcal{A}}_s \cap \mathcal{E}_{\text{good}} \cap \left\{ t - 1 \le \tau_+(\tilde{\phi}^l) \right\} \subseteq \tilde{\mathcal{A}}_{s+1}.$$

It follows that

$$
\begin{aligned}
\mathbb{P}\left( \tilde{\mathcal{A}}_t^c \cap \left\{ t - 1 \le \tau_+(\tilde{\phi}^l) \right\} \right) &= \mathbb{P}\left( \tilde{\mathcal{A}}_{t-1}^c \cap \left\{ t - 1 \le \tau_+(\tilde{\phi}^l) \right\} \right) \\
&\quad + \mathbb{P}\left( \tilde{\mathcal{A}}_{t-1} \cap \left\{ t - 1 \le \tau_+(\tilde{\phi}^l) \right\} \cap \left\{ \boldsymbol{a}_{s+1}^\top \boldsymbol{a}^* < \frac{\gamma_a}{16} \right\} \right) \\
&\le \mathbb{P}\left( \tilde{\mathcal{A}}_{t-1}^c \cap \left\{ t - 1 \le \tau_+(\tilde{\phi}^l) \right\} \right) + 5\delta \\
&\le \mathbb{P}\left( \tilde{\mathcal{A}}_{t_0}^c \cap \left\{ t - 1 \le \tau_+(\tilde{\phi}^l) \right\} \right) + 5\delta(t - t_0) \\
&\le \mathbb{P}\left( \tilde{\mathcal{A}}_{t_0}^c \right) - 5\delta(t - t_0) \\
&\le 7\delta + 5\delta(t - t_0). \tag{D.38}
\end{aligned}
$$

Similar to (D.34), we can guarantee that $\min_{i \in [N]} (\boldsymbol{a}_s^i)^\top \boldsymbol{a}^* \ge \gamma_a / 64$ holds for any $s \le t - 1 \le \min\{\tau_+(\tilde{\phi}^l), \widetilde{O}(\eta^{-2})\}$ under the event $\tilde{\mathcal{A}}_t \cap \mathcal{E}_{\text{good}}$. It follows that

$$
\begin{aligned}
\cos \phi_t &= \frac{1}{\|\boldsymbol{v}_t\|} \left\{ \boldsymbol{v}_0^\top \boldsymbol{w}^* + \eta \sum_{m=t_0}^{\ell-1} \frac{1}{N} \sum_{i=1}^N \frac{\pi - \phi_m^i}{2\pi} \frac{(\boldsymbol{a}_m^i)^\top \boldsymbol{a}^*}{\|\boldsymbol{v}_m\|} \sin^2 \phi_m - \eta \sum_{m=t_0}^{t-1} (\boldsymbol{\xi}_m^\top \boldsymbol{w}^* + \boldsymbol{h}_m^\top \boldsymbol{w}^*) \right\} \\
&\ge \frac{1}{\|\boldsymbol{v}_t\|} \left\{ \|\boldsymbol{v}_0\| \cos \phi_0 + \frac{\eta t}{3} \cdot \frac{\gamma_a \sin^2 \tilde{\phi}^u}{64} \cdot \frac{\pi - \tilde{\phi}^u}{2\pi} \right. \\
&\qquad \left. - 2c\eta K_a^2 (\sqrt{Id} + I) \sqrt{\log(Nd/\delta)} - 2cK_a \sqrt{\frac{\eta \log(Nd/\eta\delta)}{N}} \right\}. \tag{D.39}
\end{aligned}
$$

Taking

$$\tau_{2,2} = \frac{9 \cos \tilde{\phi}^l}{\eta} \frac{384\pi}{\gamma_a \sin^2 \tilde{\phi}^u} \frac{1}{\pi - \tilde{\phi}^u}.$$

Plugging $t = t_0 + \tau_{2,2}$ into (D.39), under $\left\{ t_0 + \tau_{2,2} - 1 \le \tau_+(\tilde{\phi}^l) \right\} \cap \tilde{\mathcal{A}}_{t_0 + \tau_{2,2}} \cap \mathcal{E}_{\text{good}}$, we can get

$$
\begin{aligned}
\cos \phi_{t_0 + \tau_{2,2}} &\ge 6 \cos \tilde{\phi}^f + 3 \cos \tilde{\phi}^l - 2c\eta K_a^2 (\sqrt{Id} + I) \sqrt{\log(Nd/\delta)} - 2cK_a \sqrt{\frac{\eta \log(Nd/\eta\delta)}{N}} \\
&\ge 2 \cos \tilde{\phi}^l - 2c\eta K_a^2 (\sqrt{Id} + I) \sqrt{\log(Nd/\delta)} - 2cK_a \sqrt{\frac{\eta \log(Nd/\eta\delta)}{N}} \ge \cos \tilde{\phi}^l,
\end{aligned}
$$

where we used the condition (D.6). Invoking (D.38), we can have

$$
\begin{aligned}
\mathbb{P}\left( \tau_+(\tilde{\phi}^l) \ge t_0 + \tau_{2,2} \right) &= \mathbb{P}\left( \left\{ \tau_+(\tilde{\phi}^l) \ge t_0 + \tau_{2,2} - 1 \right\} \cap \left\{ \cos \phi_{t_0 + \tau_{2,2}} \le \cos \tilde{\phi}^l \right\} \right) \\
&= \mathbb{P}\left( \left\{ \tau_+(\tilde{\phi}^l) \ge t_0 + \tau_{2,2} - 1 \right\} \cap \tilde{\mathcal{A}}_{t_0 + \tau_{2,2}} \cap \left\{ \cos \phi_{t_0 + \tau_{2,2}} \le \cos \tilde{\phi}^l \right\} \right) \\
&\quad + \mathbb{P}\left( \left\{ \tau_+(\tilde{\phi}^l) \ge t_0 + \tau_{2,2} - 1 \right\} \cap \tilde{\mathcal{A}}_{t_0 + \tau_{2,2}}^c \cap \left\{ \cos \phi_{t_0 + \tau_{2,2}} \le \cos \tilde{\phi}^l \right\} \right) \\
&\le 5\delta + \mathbb{P}\left( \left\{ \tau_+(\tilde{\phi}^l) \ge t_0 + \tau_{2,2} - 1 \right\} \cap \tilde{\mathcal{A}}_{t_0 + \tau_{2,2}}^c \right) \\
&\le 12\delta + 5\delta \cdot \tau_{2,2}.
\end{aligned}
$$

Recalling $t_0 = \tau_+(\tilde{\phi}^f)$ and (D.37), we have

$$
\begin{aligned}
\mathbb{P}\left( \tau_+(\tilde{\phi}^l) \ge \tau_{2,1} + \tau_{2,2} \right) &\le \mathbb{P}\left( \left\{ \tau_+(\tilde{\phi}^l) \ge \tau_+(\tilde{\phi}^f) + \tau_{2,2} \right\} \cap \left\{ \tau_+(\tilde{\phi}^f) \le \tau_{2,1} \right\} \right) \\
&\quad + \mathbb{P}\left( \left\{ \tau_+(\tilde{\phi}^l) \ge \tau_+(\tilde{\phi}^f) + \tau_{2,2} \right\} \cap \left\{ \tau_+(\tilde{\phi}^f) \ge \tau_{2,1} \right\} \right)
\end{aligned}
$$

$$\leq \mathbb{P}\left(\tau_+(\tilde{\phi}^l) \geq \tau_+(\tilde{\phi}^f) + \tau_{2,2}\right) + \mathbb{P}\left(\tau_+(\tilde{\phi}^f) \geq \tau_{2,1}\right)$$
$$\leq 17\delta + 5\delta \cdot \tau_{2,2}.$$

By adjusting the level of $\delta$, we can verify that $\tau_v \leq \tau_{2,1} + \tau_{2,2}$ such that $\phi_{\tau_v} \leq \tilde{\phi}^l$ and $\boldsymbol{a}_{\tau_v}^\top \boldsymbol{a}^* \geq \gamma_a/32$ with high probability. Now recalling that

$$\gamma_a = \frac{16(\pi-1)(\mathbf{1}^\top \boldsymbol{a}^*)^2}{k(\pi+k-1)}, \quad \sin^2 \tilde{\phi}^u = \sqrt{1 - \left(\frac{144\pi}{\pi+k-1}\frac{(\mathbf{1}^\top \boldsymbol{a}^*)^2}{k}\log(4+k^2)\right)^2} \geq \frac{1}{2},$$

we have

$$\tau_{2,1} + \tau_{2,2} = \frac{1}{\eta\gamma_a}\left(\frac{40000}{\sin^3 \phi_0} + \frac{1}{\sin^2 \tilde{\phi}^u}\frac{3465\pi \cos \tilde{\phi}^l}{\pi - \tilde{\phi}^u}\right)$$
$$\lesssim \frac{1}{\eta}\left(\frac{k^2}{\sin^3 \phi_0} + \log(k)\right)$$
$$\lesssim \frac{1}{\eta}\frac{k^2}{\sin^3 \phi_0},$$

where the last inequality holds due to (D.8). $\qquad\square$

# E  Convergence with Linear Speedup

## E.1  Convergence of the First Layer

**Lemma 4 restated.** Under the settings in Theorem 3. Suppose the initial point satisfies $\phi_0 \leq \tilde{\phi}^l$ and $\boldsymbol{a}_0^\top \boldsymbol{a}^* \geq \gamma_a/32$. With probability at least $1 - \delta$, we can guarantee that $\sin^2 \phi_t \lesssim \epsilon$ holds for any $\tilde{O}\left(k^2\eta^{-1}\right) \leq t \leq \tilde{O}(\eta^{-2})$.

*Proof.* Denote the event $\mathcal{C}_t = \{\boldsymbol{a}_s^\top \boldsymbol{a}^* \geq \gamma_a/128 \cdot (\|\boldsymbol{a}^*\|^2 \wedge 1) : \forall s \leq t\}$. For any $s \leq t-1$, with probability at least $1 - 3\delta$, we have

$$\mathbb{1}_{\mathcal{C}_{t-1}}\cos\phi_s = \frac{\mathbb{1}_{\mathcal{C}_{t-1}}}{\|\boldsymbol{v}_s\|}\left\{\boldsymbol{v}_0^\top \boldsymbol{w}^* + \eta\sum_{l=0}^{s-1}\frac{\pi-\phi_l}{2\pi}\frac{\boldsymbol{a}_l^\top \boldsymbol{a}^*}{\|\boldsymbol{v}_l\|}\sin^2\phi_l - \eta\sum_{l=0}^{s-1}(\boldsymbol{\xi}_l^\top \boldsymbol{w}^* + \boldsymbol{h}_l^\top \boldsymbol{w}^*)\right\}$$
$$\geq \frac{\mathbb{1}_{\mathcal{C}_{t-1}}}{\|\boldsymbol{v}_s\|}\left\{\frac{\cos\tilde{\phi}^l}{3} - 2c\eta K_a^2(\sqrt{Id}+I)\sqrt{\log(Nd/\delta)} - 2cK_a\sqrt{\frac{\eta\log(Nd/\eta\delta)}{N}}\right\}$$
$$\geq \mathbb{1}_{\mathcal{C}_{t-1}}\cdot\frac{2(\mathbf{1}^\top \boldsymbol{a}^*)^2}{k(\pi+k-1)}, \tag{E.1}$$

where we used the assumption $I \lesssim \frac{d}{\epsilon N}\min\left\{1, \frac{k^2 d^{1/2}}{N^{1/2}}, \frac{k^4 d}{N^2 \epsilon}\right\}$ and $\epsilon < dk^{-2}$. Together with (D.4), we can guarantee $\cos\phi_t^i \geq \frac{(\mathbf{1}^\top \boldsymbol{a}^*)^2}{k(\pi+k-1)}$ holds for any $i \in [N]$ with probability at least $1 - \delta$. It follows that for any $s \leq t-1$,

$$\mathbb{1}_{\mathcal{C}_{t-1}}\frac{1}{N}\sum_{i=1}^{N}B_s^i \geq \mathbb{1}_{\mathcal{C}_{t-1}}\left\{\frac{1}{N}\sum_{i=1}^{N}g(\phi_s^i) - 1\right\}$$
$$\geq \mathbb{1}_{\mathcal{C}_{t-1}}\left\{g\left(\arccos\left(\frac{(\mathbf{1}^\top \boldsymbol{a}^*)^2}{k(\pi+k-1)}\right)\right) - g(\pi/2)\right\}$$
$$\overset{(i)}{\geq} \mathbb{1}_{\mathcal{C}_{t-1}}\left\{g'(\pi/6)\left|\arccos\left(\frac{(\mathbf{1}^\top \boldsymbol{a}^*)^2}{k(\pi+k-1)}\right) - \arccos(0)\right|\right\}$$
$$\overset{(ii)}{\geq} \mathbb{1}_{\mathcal{C}_{t-1}}\cdot\frac{5\pi}{12}\frac{(\mathbf{1}^\top \boldsymbol{a}^*)^2}{k(\pi+k-1)}, \tag{E.2}$$

where $(i)$ holds since $g'(x) = -(\pi - x)\sin x$ is decreasing in $(0, \pi/2]$ and $\frac{(\mathbf{1}^\top \mathbf{a}^*)^2}{k(\pi+k-1)} \leq \frac{\sqrt{3}}{2}$ due to Assumption 1; and $(ii)$ holds since $\arccos'(x) = -\frac{1}{\sqrt{1-x^2}}$. Using the dynamic in Lemma 2, under the event $\mathcal{C}_{t-1}$, we can get

$$
\begin{aligned}
\mathbf{a}_t^\top \mathbf{a}^* &\geq \left(1 - \eta\frac{\pi-1}{2\pi}\right)^t \mathbf{a}_0^\top \mathbf{a}^* + \frac{\eta\|\mathbf{a}^*\|^2}{2\pi} \sum_{s=0}^{t-1} \left(1 - \eta\frac{\pi-1}{2\pi}\right)^{t-1-s} \frac{1}{N}\sum_{i=1}^{N} B_s^i - |S(\boldsymbol{\epsilon}_{0:t-1})| \\
&\geq \left(1 - \eta\frac{\pi-1}{2\pi}\right)^t \frac{\gamma_a}{32} + \left[1 - \left(1 - \eta\frac{\pi-1}{2\pi}\right)^t\right] \frac{5\|\mathbf{a}^*\|^2}{12} \frac{(\mathbf{1}^\top \mathbf{a}^*)^2}{k(\pi+k-1)} - 2c\sqrt{\frac{\eta k \log(1/\eta\delta)}{N}} K_a^2 \\
&\geq \frac{\|\mathbf{a}^*\|^2 \wedge 1}{4} \min\left\{\frac{\gamma_a}{32}, \frac{5(\mathbf{1}^\top \mathbf{a}^*)^2}{12k(\pi+k-1)}\right\} - 2c\sqrt{\frac{\eta k \log(1/\eta\delta)}{N}} K_a^2 \\
&\geq \frac{\gamma_a}{128} \cdot (\|\mathbf{a}^*\|^2 \wedge 1).
\end{aligned}
$$

By adjusting the level of $\delta$, we can guarantee

$$
\mathbb{P}\left(\mathcal{C}_{\widetilde{O}(\eta^{-2})}\right) = \mathbb{P}\left(\mathbf{a}_s^\top \mathbf{a}^* \geq \frac{\gamma_a}{128}\cdot(\|\mathbf{a}^*\|^2 \wedge 1): \forall s \leq t\right) \geq 1 - \delta.
$$

Next we assume $\mathcal{C}_{\widetilde{O}(\eta^{-2})}$ happens and hide the indicator.

Invoking (E.1), we have for any $s \leq \widetilde{O}(\eta^{-2})$

$$
\begin{aligned}
\cos\phi_s \lambda_s = \cos\phi_s \frac{1}{N}\sum_{i=1}^{N} \frac{\pi - \phi_t^i}{2\pi} \frac{(\mathbf{a}_t^i)^\top \mathbf{a}^*}{\|\mathbf{v}_t\|^2} &\geq \frac{2(\mathbf{1}^\top \mathbf{a}^*)^2}{k(\pi+k-1)} \cdot \frac{\pi - \pi/2}{2\pi} \frac{\gamma_a}{128} \cdot (\|\mathbf{a}^*\|^2 \wedge 1) \\
&\geq \frac{\gamma_a}{512} \cdot (\|\mathbf{a}^*\|^2 \wedge 1) =: \frac{\tilde{\gamma}_a}{12}, \tag{E.3}
\end{aligned}
$$

where we used $\phi_t^i < \pi/2$. Invoking Lemma 1, we have

$$
\begin{aligned}
\|\mathbf{v}_t\|^2 \sin^2\phi_t &\leq \left(1 - \frac{\eta\tilde{\gamma}_a}{12}\right)^2 \|\mathbf{v}_{t-1}\|^2 \sin^2\phi_{t-1} - 2\eta M_{1,t-1} + \eta^2 M_{2,t-1} + H_t \\
&\leq \left(1 - \frac{\eta\tilde{\gamma}_a}{12}\right)^{2t} \|\mathbf{v}_0\|^2 \sin^2\phi_0 + 2\eta \left|\sum_{s=0}^{t-1}\left(1 - \frac{\eta\tilde{\gamma}_a}{12}\right)^{2(t-1-s)} M_{1,s}\right| \\
&\quad + \sum_{s=0}^{t-1}\left(1 - \frac{\eta\tilde{\gamma}_a}{12}\right)^{2(t-1-s)} \left(\eta^2 M_{2,t-1} + \|\mathbf{v}_t\|^2 \sin^2\phi_t - \|\check{\mathbf{v}}_t\|^2 \sin^2\check{\phi}_t\right). \tag{E.4}
\end{aligned}
$$

Recall the choices

$$
\eta = \frac{1}{ck^2\|\mathbf{a}^*\|^2}\frac{N\epsilon}{d\log(dN/\epsilon\delta)}, \quad T_v = \frac{1}{\eta}\log\left(\frac{1}{\epsilon}\right)\cdot\frac{ck^2}{(\mathbf{1}^\top \mathbf{a}^*)^2}\frac{1}{\|\mathbf{a}^*\|^2 \wedge 1}. \tag{E.5}
$$

According to (D.35) and (D.26), with probability at least $1 - \delta$, we have

$$
\begin{aligned}
&\sum_{s=0}^{t-1}\left(1 - \frac{\eta\tilde{\gamma}_a}{12}\right)^{2(t-1-s)} \left|\|\mathbf{v}_t\|^2 \sin^2\phi_s - \|\check{\mathbf{v}}_s\|^2 \sin^2\check{\phi}_s\right| \\
&\leq \sum_{s=0}^{t-1}\left(1 - \frac{\eta\tilde{\gamma}_a}{12}\right)^{2(t-1-s)} \left(6\eta \sin\phi_t\|\mathbf{h}_s\| + \eta^2\|\mathbf{h}_s\|^2\right) \\
&\leq \frac{c\eta^2(\sqrt{Id}+I)\sqrt{\log(Nd/\eta\delta)}K_a^4 + c\eta^4(Id+I^2)\log(Nd/\eta\delta)K_a^8}{1 - \left(1 - \frac{\eta\tilde{\gamma}_a}{12}\right)^2} \\
&\leq \frac{24}{\tilde{\gamma}_a}\left\{c\eta(\sqrt{Id}+I)\log^{1/2}(Nd/\eta\delta)K_a^4 + c\eta^3(Id+I^2)\log(Nd/\eta\delta)K_a^8\right\} \\
&\leq N\epsilon\left(\sqrt{\frac{I}{d}} + \frac{I}{d}\right) + \frac{N^3\epsilon^3(I^2+Id)}{d^3k^4}, \tag{E.6}
\end{aligned}
$$

where we used $\eta\tilde{\gamma}_a/12 < 1/2$. Using Lemma 11 and recalling the definition of $M_{1,t}$, we have the following concentration

$$\eta\left|\sum_{s=0}^{t-1}\left(1-\frac{\eta\tilde{\gamma}_a}{12}\right)^{2(t-1-s)}M_{1,s}\right|$$

$$= \eta\left|\sum_{s=0}^{t-1}\left(1-\frac{\eta\tilde{\gamma}_a}{12}\right)^{2(t-1-s)}\left(\boldsymbol{v}_s - \eta\mathbf{P}_s\frac{1}{N}\sum_{i=1}^{N}\nabla_{\boldsymbol{w}}L(\boldsymbol{w}_s^i,\boldsymbol{a}_s^i)\right)^{\top}\left(\mathbf{I}-\boldsymbol{w}^*(\boldsymbol{w}^*)^{\top}\right)\boldsymbol{\xi}_s\right|$$

$$\leq \frac{12cK_a^2\eta}{\sqrt{N}}\sqrt{\log(1/\delta)\sum_{s=0}^{t-1}\left(1-\frac{\eta\tilde{\gamma}_a}{12}\right)^{2(t-1-s)}\left\|\frac{1}{N}\sum_{i=1}^{N}\boldsymbol{v}_s^i - \eta\mathbf{P}_s\nabla_{\boldsymbol{w}}L(\boldsymbol{w}_s^i,\boldsymbol{a}_s^i)\right\|^2}$$

$$\leq \frac{cK_a^2\eta}{\sqrt{N}}\sqrt{\log(1/\delta)\sum_{s=0}^{t-1}\left(1-\frac{\eta\tilde{\gamma}_a}{12}\right)^{2(t-1-s)}(9+\eta^2K_a^2)}$$

$$\leq \frac{2cK_a^2\eta}{\sqrt{N}}\sqrt{\frac{10\log(1/\delta)}{1-\left(1-\frac{\eta\tilde{\gamma}_a}{12}\right)^2}} \leq 12cK_a^2\sqrt{\frac{\eta\log(1/\delta)}{N\tilde{\gamma}_a}} \leq \sqrt{\frac{\epsilon}{d}} \tag{E.7}$$

holds with probability at least $1-\delta$. Moreover, it holds that

$$M_{2,t} = \boldsymbol{\xi}_t^{\top}\left(\mathbf{I}-\boldsymbol{w}^*(\boldsymbol{w}^*)^{\top}\right)\boldsymbol{\xi}_t \leq \|\boldsymbol{\xi}_t\|^2 \leq \frac{cK_a^4d\log(d/\eta\delta)}{N},$$

which yields with probability at least $1-\delta$ such that

$$\eta^2\sum_{s=0}^{t-1}\left(1-\frac{\eta\tilde{\gamma}_a}{12}\right)^{2(t-1-s)}M_{2,s} \leq \frac{24\eta}{\tilde{\gamma}_a}\cdot\frac{cK_a^4d\log(d/\eta\delta)}{N} \leq \epsilon. \tag{E.8}$$

Plugging (E.6), (E.7) and (E.8) into (E.4), we can guarantee that with probability at least $1-3\delta$, for any $T_v \leq t \leq \widetilde{O}(\eta^{-2})$

$$\sin^2\phi_t \leq \frac{1}{\|\boldsymbol{v}_t\|^2}\left\{3\epsilon^2 + \sqrt{\frac{\epsilon}{d}} + \left[N\left(\sqrt{\frac{I}{d}}+\frac{I}{d}\right)+\frac{N^3\epsilon^2(I^2+Id)}{d^3k^4}+1\right]\epsilon\right\}$$

$$\leq 4\max\left\{\sqrt{\frac{\epsilon}{d}}+\frac{(\sqrt{Id}+I)N\epsilon}{d},\epsilon\right\}$$

$$= 4\left(\sqrt{\frac{\epsilon}{d}}+\frac{(\sqrt{Id}+I)N\epsilon}{d}\right), \tag{E.9}$$

where we used $I \lesssim \frac{d}{\epsilon N}\min\left\{1,\frac{k^2d^{1/2}}{N^{1/2}},\frac{k^4d}{N^2\epsilon}\right\}$ and $\epsilon < d^{-1}$. Then we further have

$$\left\|\left(\mathbf{I}-\boldsymbol{w}^*(\boldsymbol{w}^*)^{\top}\right)\left(\boldsymbol{v}_t-\eta\mathbf{P}_t\frac{1}{N}\sum_{i=1}^{N}\nabla_{\boldsymbol{w}}L(\boldsymbol{w}_t^i,\boldsymbol{a}_t^i)\right)\right\|^2$$

$$\leq \boldsymbol{v}_t^{\top}\left(\mathbf{I}-\boldsymbol{w}^*(\boldsymbol{w}^*)^{\top}\right)\boldsymbol{v}_t + \eta^2\frac{1}{N}\sum_{i=1}^{N}\left\|\mathbf{P}_t\nabla_{\boldsymbol{w}}L(\boldsymbol{w}_t^i,\boldsymbol{a}_t^i)\right\|^2$$

$$\leq \|\boldsymbol{v}_t\|^2\sin^2\phi_t + 4\eta^2\frac{1}{N}\sum_{i=1}^{N}\|\nabla_{\boldsymbol{w}}L(\boldsymbol{w}_t^i,\boldsymbol{a}_t^i)\|^2$$

$$\leq 36\left(\sqrt{\frac{\epsilon}{d}}+\frac{(\sqrt{Id}+I)N\epsilon}{d}\right)+4\eta^2K_a^4$$

$$\leq 49\left(\sqrt{\frac{\epsilon}{d}}+\frac{(\sqrt{Id}+I)N\epsilon}{d}\right).$$

With this upper bound, we can improve the concentration (E.7) through

$$\eta \left| \sum_{s=0}^{t-1} \left(1 - \frac{\eta\tilde{\gamma}_a}{12}\right)^{2(t-1-s)} M_{1,s} \right|$$

$$\leq \frac{cK_a^2\eta}{\sqrt{N}} \sqrt{\log(1/\delta) \sum_{s=0}^{t-1} \left(1 - \frac{\eta\tilde{\gamma}_a}{12}\right)^{2(t-1-s)} \cdot 49 \left( \sqrt{\frac{\epsilon}{d}} + \frac{(\sqrt{Id}+I)N\epsilon}{d} \right)}$$

$$\leq 4\sqrt{\frac{\epsilon}{d}} \left( \sqrt{\frac{\epsilon}{d}} + \frac{(\sqrt{Id}+I)N\epsilon}{d} \right)^{\frac{1}{2}}. \tag{E.10}$$

Denote $H = c\eta K_a^4(\sqrt{Id}+I)\sqrt{\log(Nd/\delta)}$. Using (E.9), we can further refine the bound (E.6) by

$$\left| \|\boldsymbol{v}_t\|^2 \sin^2 \phi_t - \|\check{\boldsymbol{v}}_t\|^2 \sin^2 \check{\phi}_t \right| \leq 6\eta \sin \phi_t \|\boldsymbol{h}_{t-1}\| + \eta^2 \|\boldsymbol{h}_{t-1}\|^2$$

$$\leq 4 \left( \sqrt{\frac{\epsilon}{d}} + \frac{(\sqrt{Id}+I)N\epsilon}{d} \right)^{\frac{1}{2}} \cdot 6\eta \|\boldsymbol{h}_{t-1}\| + \eta^2 \|\boldsymbol{h}_{t-1}\|^2$$

$$\leq 4 \left( \sqrt{\frac{\epsilon}{d}} + \frac{(\sqrt{Id}+I)N\epsilon}{d} \right)^{\frac{1}{2}} \cdot 6\eta H + \eta^2 H^2.$$

It follows that

$$\sum_{s=0}^{t-1} \left(1 - \frac{\eta\tilde{\gamma}_a}{12}\right)^{2(t-1-s)} \left| \|\boldsymbol{v}_s\|^2 \sin^2 \phi_s - \|\check{\boldsymbol{v}}_s\|^2 \sin^2 \check{\phi}_s \right|$$

$$\leq 4 \frac{(\sqrt{Id}+I)N\epsilon}{d} \left( \sqrt{\frac{\epsilon}{d}} + \frac{(\sqrt{Id}+I)N\epsilon}{d} \right)^{\frac{1}{2}} + \epsilon. \tag{E.11}$$

Combining (E.10), (E.11) and (E.8), we can show that for any $2T_v \leq t \leq (\eta K_a^2)^{-2}$

$$\sin^2 \phi_t \leq 4 \max \left\{ \left( \sqrt{\frac{\epsilon}{d}} + \frac{(\sqrt{Id}+I)N\epsilon}{d} \right)^{1+\frac{1}{2}}, \epsilon \right\}. \tag{E.12}$$

Repeating (E.10)-(E.12) for $K = \log(1/\epsilon)$ times, we can guarantee that for any $(K+1)T_v \leq t \leq \widetilde{O}(\eta^{-2})$

$$\sin^2 \phi_t \leq 4 \max \left\{ \left( \sqrt{\frac{\epsilon}{d}} + \frac{(\sqrt{Id}+I)N\epsilon}{d} \right)^{1+\frac{K}{2}}, \epsilon \right\}$$

$$= 4\epsilon.$$

$$\square$$

## E.2  Convergence of the Second Layer

**Lemma 5 restated.** Under the choice for $\eta$ and conditions for $\epsilon$ in Theorem 3. Suppose $\sin^2 \phi_t \leq \epsilon$ holds for any $0 \leq t \leq \widetilde{O}(\eta^{-2})$. With probability at least $1 - \delta$, we can guarantee that $\|\boldsymbol{a}_t - \boldsymbol{a}^*\|^2 \lesssim \epsilon$ holds for any $\widetilde{O}(\eta^{-1}) \leq t \leq \widetilde{O}(\eta^{-2})$.

*Proof.* Let $\boldsymbol{h}_t^i = \boldsymbol{v}_t^i - \boldsymbol{v}_t$. Similar to (D.35), we can verify that for any $i \in [N]$

$$\left| \|\boldsymbol{v}_t^i\|^2 \sin^2 \phi_t^i - \|\boldsymbol{v}_t\|^2 \sin^2 \phi_t \right| \leq 2\eta \sin \phi_t \|\boldsymbol{v}_t\| \|\boldsymbol{h}_{t-1}^i\| + \eta^2 \|\boldsymbol{h}_{t-1}^i\|^2$$

$$\leq 2\eta\epsilon \cdot c\eta K_a^2(\sqrt{Id}+I)\sqrt{\log(Nd/\delta)} + c\eta^4(\sqrt{Id}+I)^2 \log(Nd/\delta)$$

$$\leq 4\eta\epsilon,$$

where we used Lemma 14 and $\sin^2 \phi_t \leq \epsilon$. It further implies that

$$
\begin{aligned}
\sin^2 \phi_t^i &\leq \frac{\|\boldsymbol{v}_t\|^2 \sin^2 \phi_t + \left| \|\boldsymbol{v}_t^i\|^2 \sin^2 \phi_t^i - \|\boldsymbol{v}_t\|^2 \sin^2 \phi_t \right|}{\|\boldsymbol{v}_t^i\|^2} \\
&\leq \frac{9\epsilon + 4\eta\epsilon}{\left( \|\boldsymbol{v}_t\| - \|\boldsymbol{v}_t - \boldsymbol{v}_t^i\| \right)^2} \\
&\leq 100\epsilon.
\end{aligned}
\tag{E.13}
$$

Further, for any $0 \leq t \leq \widetilde{O}(\eta^{-2})$ and $i \in [N]$ we have

$$
\pi - g(\phi_t^i) = g(0) - g(\phi_t^i) \overset{(i)}{\leq} (\pi - \phi_t^i) \sin \phi_t^i \cdot \phi_t^i \leq 5\pi\epsilon^{1/2} \cdot \phi_t^i \overset{(ii)}{\leq} 5\pi\epsilon,
\tag{E.14}
$$

where $(i)$ holds since $g'(x) = -(\pi - x)\sin x$; and $(ii)$ holds due to $\arcsin'(x) = \frac{1}{\sqrt{1-x^2}}$. Applying this bound, we can get

$$
\begin{aligned}
\eta \left\langle \frac{1}{N} \sum_{i=1}^N \nabla_{\boldsymbol{a}} L(\boldsymbol{w}_t^i, \boldsymbol{a}_t^i), \boldsymbol{a}_t - \boldsymbol{a}^* \right\rangle &= \frac{\eta}{2\pi} (\boldsymbol{a}_t - \boldsymbol{a}^*)^\top \left( \mathbf{1}\mathbf{1}^\top + (\pi - 1)\mathbf{I} \right) (\boldsymbol{a}_t - \boldsymbol{a}^*) \\
&\quad + \frac{\eta}{2\pi} \left( \pi - \frac{1}{N} \sum_{i=1}^N g(\phi_t^i) \right) (\boldsymbol{a}_t - \boldsymbol{a}^*)^\top \boldsymbol{a}^* \\
&\geq \frac{\eta(\pi-1)}{2\pi} \|\boldsymbol{a}_t - \boldsymbol{a}^*\|^2 - 3\eta\epsilon \|\boldsymbol{a}_t - \boldsymbol{a}^*\| \|\boldsymbol{a}^*\| \\
&\geq \frac{\eta(\pi-1)}{2\pi} \|\boldsymbol{a}_t - \boldsymbol{a}^*\|^2 - 6\eta\epsilon K_a^2,
\end{aligned}
\tag{E.15}
$$

where the last inequality holds due to Lemma 12 such that $\|\boldsymbol{a}_t - \boldsymbol{a}^*\| \leq 6\|\boldsymbol{a}^*\|$. In addition, it holds that

$$
\begin{aligned}
\left\| \frac{1}{N} \sum_{i=1}^N \nabla_{\boldsymbol{a}} L(\boldsymbol{w}_t^i, \boldsymbol{a}_t^i) \right\|^2 &= \frac{1}{4\pi^2} \left\| \left( \mathbf{1}\mathbf{1}^\top + (\pi-1)\mathbf{I} \right) (\boldsymbol{a}_t - \boldsymbol{a}^*) + \left( \frac{1}{N} \sum_{i=1}^N g(\phi_t^i) - \pi \right) \boldsymbol{a}^* \right\|^2 \\
&\leq \frac{(\pi + k - 1)^2}{2\pi^2} \|\boldsymbol{a}_t - \boldsymbol{a}^*\|^2 + 15\epsilon^2 \|\boldsymbol{a}^*\|^2,
\end{aligned}
\tag{E.16}
$$

where we used the relation (E.14). Denote $\tilde{\boldsymbol{a}}_t = \boldsymbol{a}_t - \boldsymbol{a}^* - \eta \frac{1}{N} \sum_{i=1}^N \nabla_{\boldsymbol{a}} L(\boldsymbol{w}_t^i, \boldsymbol{a}_t^i)$, then we have

$$
\begin{aligned}
\|\boldsymbol{a}_t - \boldsymbol{a}^*\|^2 &= \left\| \boldsymbol{a}_{t-1} - \boldsymbol{a}^* - \eta \frac{1}{N} \sum_{i=1}^N \nabla_{\boldsymbol{a}} L(\boldsymbol{w}_{t-1}^i, \boldsymbol{a}_{t-1}^i) - \eta \boldsymbol{\epsilon}_t \right\|^2 \\
&= \left\| \boldsymbol{a}_{t-1} - \boldsymbol{a}^* - \eta \frac{1}{N} \sum_{i=1}^N \nabla_{\boldsymbol{a}} L(\boldsymbol{w}_{t-1}^i, \boldsymbol{a}_{t-1}^i) \right\|^2 - 2\eta \langle \tilde{\boldsymbol{a}}_{t-1}, \boldsymbol{\epsilon}_{t-1} \rangle + \eta^2 \|\boldsymbol{\epsilon}_{t-1}\|^2 \\
&\leq \left( 1 - \frac{\eta(\pi-1)}{2\pi} + \frac{\eta^2(\pi+k-1)^2}{2\pi^2} \right) \|\boldsymbol{a}_{t-1} - \boldsymbol{a}^*\|^2 \\
&\quad - 2\eta \langle \tilde{\boldsymbol{a}}_{t-1}, \boldsymbol{\epsilon}_{t-1} \rangle + \eta^2 \|\boldsymbol{\epsilon}_{t-1}\|^2 + 6\eta^2\epsilon^2 K_a^2 + \frac{\eta^2\epsilon^4}{\pi} \|\boldsymbol{a}^*\|^2 \\
&\leq \left( 1 - \frac{\eta(\pi-1)}{4\pi} \right) \|\boldsymbol{a}_{t-1} - \boldsymbol{a}^*\|^2 - 2\eta \langle \tilde{\boldsymbol{a}}_{t-1}, \boldsymbol{\epsilon}_{t-1} \rangle + \eta^2 \|\boldsymbol{\epsilon}_{t-1}\|^2 + 6\eta\epsilon K_a^2 + \frac{\eta^2\epsilon^2}{\pi} \|\boldsymbol{a}^*\| \\
&\leq \left( 1 - \frac{\eta(\pi-1)}{4\pi} \right)^t \|\boldsymbol{a}_0 - \boldsymbol{a}^*\|^2 + 2\eta \left| \sum_{s=0}^{t-1} \left( 1 - \frac{\eta(\pi-1)}{4\pi} \right)^s \langle \tilde{\boldsymbol{a}}_s, \boldsymbol{\epsilon}_s \rangle \right| \\
&\quad + \eta^2 \sum_{s=0}^{t-1} \left( 1 - \frac{\eta(\pi-1)}{4\pi} \right)^{t-s} \|\boldsymbol{\epsilon}_s\|^2 \\
&\quad + \sum_{s=0}^{t-1} \left( 1 - \frac{\eta(\pi-1)}{4\pi} \right)^{t-1-s} \left( 6\eta\epsilon K_a^2 + 15\eta^2\epsilon^2 \|\boldsymbol{a}^*\|^2 \right).
\end{aligned}
\tag{E.17}
$$

Using Lemma 12 again, we can verify that

$$\|\tilde{\boldsymbol{a}}_t\| \le \|\boldsymbol{a}_t - \boldsymbol{a}^*\| + \eta \frac{1}{N} \sum_{i=1}^{N} \|\nabla_{\boldsymbol{a}} L(\boldsymbol{w}_t^i, \boldsymbol{a}_t^i)\|$$

$$\le 6K_a + \frac{\eta}{2\pi} \left( \| \left( \mathbf{1}\mathbf{1}^\top + (\pi - 1)\mathbf{I} \right) (\boldsymbol{a}_t - \boldsymbol{a}^*) \| + (\pi - g(\phi_t^i))\|\boldsymbol{a}^*\| \right)$$

$$\le 6K_a + \frac{\eta(\pi + k - 1)}{2\pi} \cdot 6\|\boldsymbol{a}^*\| + \frac{\eta}{2}\|\boldsymbol{a}^*\|$$

$$\le 8K_a. \tag{E.18}$$

Similar to (A.17) with $\alpha_s = \left(1 - \frac{\eta(\pi-1)}{4\pi}\right)^{t-s}$, we can guarantee

$$\eta \left| \sum_{s=0}^{t-1} \left(1 - \frac{\eta(\pi - 1)}{4\pi}\right)^{t-s} \tilde{\boldsymbol{a}}_s^\top \boldsymbol{\epsilon}_s \right| \le c\eta K_a^2 \sqrt{\frac{\log(2/\delta)}{N} \sum_{s=0}^{t} \left(1 - \frac{\eta(\pi - 1)}{4\pi}\right)^{t-s}} + \frac{c\eta K_a^2 \log(2/\delta)}{N}$$

$$\le 4cK_a^2 \sqrt{\frac{\eta \log(2/\delta)}{N}} \tag{E.19}$$

with probability at least $1 - \delta$. Now take $T_a = \frac{1}{\eta} \frac{8\pi}{\pi-1} \log\left(\frac{1}{\epsilon}\right)$. Plugging (E.19) into (E.17), together with the concentration inequality (A.19), we have for any $T_a \le t \le \widetilde{O}(\eta^{-2})$

$$\|\boldsymbol{a}_t - \boldsymbol{a}^*\|^2 \le \left(1 - \frac{\eta(\pi - 1)}{4\pi}\right)^t \|\boldsymbol{a}_0 - \boldsymbol{a}^*\|^2 + 4cK_a^2 \sqrt{\frac{\eta \log(2/\delta)}{N}}$$

$$+ \frac{2\pi c\eta k \log(t/\delta)}{N} K_a^2 + 12\pi\epsilon K_a^2 + 2\eta\epsilon^2 K_a^2$$

$$\le \epsilon\|\boldsymbol{a}_0 - \boldsymbol{a}^*\|^2 + \sqrt{\epsilon}K_a^2 + \epsilon^2 K_a^2 + 12\pi\epsilon K_a^2 + 2\eta\epsilon^2 K_a^2$$

$$\le 2\sqrt{\epsilon}K_a^2. \tag{E.20}$$

With this upper bound, we can refine (E.18) by

$$\|\tilde{\boldsymbol{a}}_t\|^2 \le 2\|\boldsymbol{a}_t - \boldsymbol{a}^*\|^2 + 2\eta^2 \frac{1}{N} \sum_{i=1}^{N} \|\nabla_{\boldsymbol{a}} L(\boldsymbol{w}_t^i, \boldsymbol{a}_t^i)\|^2 \le 8\sqrt{\epsilon}K_a^2.$$

Consequently, the bound (E.19) can be improved to

$$\eta \left| \sum_{s=0}^{t-1} \left(1 - \frac{\eta(\pi - 1)}{4\pi}\right)^{t-s} \tilde{\boldsymbol{a}}_s^\top \boldsymbol{\epsilon}_s \right| \le 4cK_a^2 \sqrt{\frac{\eta\epsilon^{1/2} \log(2/\delta)}{N}}.$$

Then we can guarantee that for any $2T_a \le t \le \widetilde{O}(\eta^{-2})$,

$$\|\boldsymbol{a}_t - \boldsymbol{a}^*\|^2 \le \left(1 - \frac{\eta(\pi - 1)}{4\pi}\right)^t \|\boldsymbol{a}_0 - \boldsymbol{a}^*\|^2 + 4cK_a^2 \sqrt{\frac{\eta\epsilon \log(2/\delta)}{N}}$$

$$+ \frac{2\pi c\eta k \log(t/\delta)}{N} K_a^2 + 12\pi\epsilon K_a^2 + 2\eta\epsilon^2 K_a^2$$

$$\le \epsilon\|\boldsymbol{a}_0 - \boldsymbol{a}^*\|^2 + \epsilon^{3/4}K_a^2 + \epsilon^2 K_a^2 + 12\pi\epsilon K_a^2 + 2\eta\epsilon^2 K_a^2$$

$$\lesssim \epsilon^{4/3}K_a^2. \tag{E.21}$$

Repeating the refinement from (E.20) to (E.21) for at most $\log(1/\epsilon)$ times, we can show finish the proof. $\qquad\square$

## E.3 Conclusion

**Theorem 3 restated.** Suppose the initial point $(\boldsymbol{v}_0, \boldsymbol{a}_0)$ satisfies $\boldsymbol{a}_0^\top \boldsymbol{a}^* \ge \gamma_a/32$ and $\phi_0 \le \tilde{\phi}^l$. For any $\epsilon > 0$, we choose $\eta = \frac{1}{ck^2\|\boldsymbol{a}^*\|^2} \frac{N\epsilon}{d \log(dN/\epsilon\delta)}$ for some absolute constant $c > 0$. If $I \lesssim \frac{d}{\epsilon N} \min\left\{1, \frac{k^2 d^{1/2}}{N^{1/2}}, \frac{k^4 d}{N^2 \epsilon}\right\}$ and $\epsilon < \min\{N^{-1}, d^{-1}, dk^{-2}\}$, then $\ell(\boldsymbol{v}_T, \boldsymbol{a}_T) = O(\epsilon K_a^2)$ holds with probability at least $1 - \delta$ where $T = \widetilde{O}\left(\frac{dk^4}{N\epsilon}\right)$.

*Proof.* According to the dynamic of $a_t$, we know

$$\mathbf{1}^\top(a_t - a^*) = \left(1 - \frac{\eta(\pi + k - 1)}{2\pi}\right)\mathbf{1}^\top(a_{t-1} - a^*) + \frac{\eta}{2\pi}\left(\frac{1}{N}\sum_{i=1}^{N}g(\phi_{t-1}^i) - \pi\right)\mathbf{1}^\top a^* + \eta\mathbf{1}^\top\epsilon_{t-1}$$

$$= \left(1 - \frac{\eta(\pi + k - 1)}{2\pi}\right)^t\mathbf{1}^\top(a_0 - a^*) + \eta\sum_{s=0}^{t-1}\left(1 - \frac{\eta(\pi + k - 1)}{2\pi}\right)^{t-1-s}\mathbf{1}^\top\epsilon_s$$

$$+ \frac{\eta}{2\pi}\sum_{s=0}^{t-1}\left(1 - \frac{\eta(\pi + k - 1)}{2\pi}\right)^{t-1-s}\left(\frac{1}{N}\sum_{i=1}^{N}g(\phi_s^i) - \pi\right)\mathbf{1}^\top a^*. \qquad \text{(E.22)}$$

It follows from (E.14) that

$$\left|\frac{\eta}{2\pi}\sum_{s=0}^{t-1}\left(1 - \frac{\eta(\pi + k - 1)}{2\pi}\right)^{t-1-s}\left(\frac{1}{N}\sum_{i=1}^{N}g(\phi_s^i) - \pi\right)\right| \leq \frac{5\pi\epsilon}{\pi + k - 1}. \qquad \text{(E.23)}$$

Using Lemma 11(1), it holds that

$$\eta\left|\sum_{s=0}^{t-1}\left(1 - \frac{\eta(\pi + k - 1)}{2\pi}\right)^{t-1-s}\mathbf{1}^\top\epsilon_s\right| \leq cK_a\sqrt{\frac{\eta\log(1/\delta)}{N}}. \qquad \text{(E.24)}$$

Plugging (E.23) and (E.24) into (E.22), and taking $T_1 = \frac{1}{\eta}\frac{2\pi}{\pi + k - 1}\log\left(\frac{1}{\epsilon}\right)$, we can have

$$\left|\mathbf{1}^\top(a_t - a^*)\right| \leq \epsilon\left|\mathbf{1}^\top(a_0 - a^*)\right| + \sqrt{\epsilon}K_a + \frac{5\pi\epsilon}{\pi + k - 1}\left|\mathbf{1}^\top a^*\right|$$

$$\leq \epsilon\sqrt{k}K_a + \sqrt{\epsilon}K_a + \frac{5\pi\epsilon}{\pi + k - 1}\left|\mathbf{1}^\top a^*\right|$$

$$\leq 2\sqrt{\epsilon}K_a, \qquad \text{(E.25)}$$

where we used $|\mathbf{1}^\top a_0| \leq |\mathbf{1}^\top a^*|$ and $\epsilon < 1/k$. Invoking Lemma 6 and the normalized network in (2), we know

$$\ell(v, a) = L(w, a) = \frac{1}{2}\left[\frac{\pi - 1}{2\pi}\|a^*\|^2 + \frac{\pi - 1}{2\pi}\|a\|^2 - \frac{g(\phi) - 1}{\pi}a^\top a^*\right.$$

$$\left. + \frac{1}{2\pi}(\mathbf{1}^\top a^*)^2 + \frac{1}{2\pi}(\mathbf{1}^\top a)^2 - \frac{1}{\pi}(\mathbf{1}^\top a)(\mathbf{1}^\top a)\right]$$

$$= \frac{1}{2}\left[\frac{\pi - 1}{2\pi}\|a - a^*\|^2 + \frac{\pi - g(\phi)}{\pi}a^\top a^* + \frac{1}{2\pi}|\mathbf{1}^\top(a - a^*)|^2\right]. \qquad \text{(E.26)}$$

Similar to (E.14), for any $t \geq T_v$, we also have

$$\pi - g(\phi_t) = g(0) - g(\phi_t) \leq (\pi - \phi_t)\sin\phi_t \cdot \phi_t \leq \pi\epsilon \cdot \phi_t \leq \pi\epsilon. \qquad \text{(E.27)}$$

Invoking Lemma 4 and 5 to (E.26), together with (E.27), we conclude that for $T = \widetilde{O}\left(\frac{dk^4}{N\epsilon}\right)$,

$$\ell(v_T, a_T) \leq \frac{1}{2}\left(\frac{15}{2}\epsilon K_a + \epsilon\|a_T\|\|a^*\| + 4\epsilon K_a^2\right) \lesssim \epsilon\|a^*\|^2,$$

with probability at least $1 - \delta$. $\qquad \square$

