# OpenReview forum: "Global Convergence Analysis of Local SGD for Two-layer Neural Network without Overparameterization"
_NeurIPS.cc/2023/Conference — NeurIPS 2023 poster_

### Official Review · Reviewer_GGhs · 2023-07-04

**Soundness:** 2 fair
**Presentation:** 3 good
**Contribution:** 2 fair
**Rating:** 5
**Confidence:** 3

**Summary:**

This paper studies the global convergence of local SGD for One-hidden-layer Convolutional Neural Networks. The authors present solid theoretical understanding. They show that without overparameterization and injecting noise,  local SGD have global convergence by new proof techniques and new understanding.

**Strengths:**

The paper is clear and well-written.
Understanding the optimization dynamics of gradient-based method is a significant theoretical issue.
This paper provides a solid theoretical analysis without highly over-parameterization and injecting noise, which results in the highly non-convexity.
Theoretically, the authors provide a novel understanding of the training dynamics by dividing them into two phases: self-correction and convergence.

**Weaknesses:**

The article's significance is limited due to the restriction of input data to Gaussian distributions. However, this limitation is not a major concern since the problem itself is non-convex, even in this simplified scenario. Additionally, I have significant concerns regarding Assumption 1 about the target function, which I will discuss in detail in the Questions section.

**Questions:**

My main question pertains to Assumption 1 about the teacher traget.
It appears that this assumption suggestes that stronger conditions on $\boldsymbol{a}^*$ are required for larger values of $k$.
I am curious to know whether this requirement is essential or merely technical.
In other words, if one use a sufficiently large $k$ that contradicts this assumption, do the training dynamics in the last region (in the proof of Theorem 1) fundamentally change?
The authors could attempt to demonstrate this aspect experimentally or theoretically.

**Limitations:**

The limitations of this article lie in the requirements for data and target Assumption.
However, these limitations should not be grounds for rejection.
Given our limited understanding of the training dynamics of nonconvex optimization, studying such simple setups can still provide valuable insights and contribute to the field.

---

> ### Author Rebuttal · Authors · 2023-08-07
>
> Thanks for taking your precious time to review our paper and please see our responses to your questions.
>
> **1. My main question pertains to Assumption 1 about the teacher target. It appears that this assumption suggests that stronger conditions on $a^{*}$ are required for larger values of k. I am curious to know whether this requirement is essential or merely technical. In other words, if one uses a sufficiently large k that contradicts this assumption, do the training dynamics in the last region (in the proof of Theorem 1) fundamentally change? The authors could attempt to demonstrate this aspect experimentally or theoretically.**
>
> We thank the reviewer for this question. In Assumption 1, the condition on the upper bound of $(1^{\prime}a^*)^2$ is merely technical, and the condition in Eq.(5) is almost essential to show the global convergence with arbitrary initialization. Let $\alpha = \frac{(1^{\prime}a^*)^2}{k |a^*|^2}$ where $\alpha \in (0,1]$ roughly stands for the degree of sparsity of the vector $a^{*}$. Then Eq.(5) will hold if
> $$\alpha > (1-32(\pi-1)/k)^{-1}\frac{\pi +k-1}{(\pi-1)k}.$$
> This right-hand side can be a constant in $(0,1)$ for proper $k$. For example, when $k \geq 320(\pi-1)^2$ (a technical choice as in Theorem 1), this condition will be satisfied as long as $\alpha > 0.48$.
>
> To show that (5) is essential, we have added an additional simulation with different $\alpha$ to show that this condition is nearly necessary for arbitrary initialization. In each trial, we randomly select an initial point from the initial region defined in our paper. Then we calculate the probability of convergence over 100 independent trials with $k=64$. From the following table (**Table 1 in the rebuttal PDF file**), we can see the probability of convergence becomes larger when $\alpha$ tends to 1. Specially, the probability becomes 1 when $\alpha > 1/4$.
>
> | alpha     | 1/64 | 1/32 | 1/16 | 1/8  | 1/4 | 1/2 |
> |-----------|------|------|------|------|-----|-----|
> | Local SGD | 0.54 | 0.65 | 0.74 | 0.87 | 1   | 1 |
> | SGD       | 0.54 | 0.62 | 0.73 | 0.85 | 1   | 1   |
>
> It is challenging to theoretically demonstrate the last region in the proof of Theorem 1 if Eq.(5) does not hold because it would require a lower-bound analysis for this specific model. Experimentally, we plot the trajectories of SGD and local SGD that start from the region with $\phi_0$ and converge to the spurious local minima. It means that the self-correction process could fail if Eq.(5) does not hold for the ground truth.
>
>
> **2. The limitations of this article lie in the requirements for data and target Assumption. However, these limitations should not be grounds for rejection. Given our limited understanding of the training dynamics of nonconvex optimization, studying such simple setups can still provide valuable insights and contribute to the field.**
>
> We thank the reviewer for these insightful and encouraging comments. To the best of our knowledge, this is the first global convergence result for local SGD for such a nonconvex function which does not require overparameterization and NTK analysis.

---

> > ### Comment · Reviewer_GGhs · 2023-08-11
> >
> > No further questions from me. I will keep my score as is.

---

### Official Review · Reviewer_bCYH · 2023-07-04

**Soundness:** 3 good
**Presentation:** 4 excellent
**Contribution:** 4 excellent
**Rating:** 7
**Confidence:** 4

**Summary:**

This paper provides a rigorous theoretical justification on how the local SGD (without the injection of noise) can find the global minima for CNN without relying on the NTK-type analysis (i.e., overparametrization) under federated learning framework.

**Strengths:**

A.	The contributions of the paper are clearly written.
B.	Nice simulation results which are consistent with the theoretical predictions.
C.	Like the part where the authors divide the landscape of the objective function into several regions. This type of analysis is not something that I can find in the current literature.


**Weaknesses:**

A.	Writing is little bit technical. It would be great if the authors can draw some cartoons of dynamics of each layer for the global convergence.

**Questions:**

A.	Why is the f (Z, w, a) CNN? It looks to me like a shallow fully connected neural network. What is the exact structure of $w \in \mathbb{R}^{d}$?
B.	In line 90, the paper says the algorithm starts from the random initialization from the same initial region as in [13, 63]. In line 203, it says arbitrary initialization. I was bit confused while reading the paper on this point. And it becomes clear while reading line 312~313.
C.	Just to clear my understanding, so your result is saying that weights in the first layer converges (in polynomial time), and then the weights in the second layer converges afterward?



**Limitations:**

This work has no negative societal impact.

---

> ### Author Rebuttal · Authors · 2023-08-07
>
> Thank you for appreciating our work and please see our responses to your questions.
>
> **1. Why is the f (Z, w, a) CNN? It looks to me like a shallow fully connected neural network. What is the exact structure of $w$?**
>
> We apologize for the confusion. Please note that we follow the assumption and terminology in Du et al. [13] and Zhou et al. [63] to refer to the network as a CNN. We acknowledge that it is indeed a fully-connected network, since there is no overlap between patches in CNN. In our paper, we elect to refer to the network as a CNN so that we can keep the terminology consistent with the most related works [13, 63]. We will add a sentence to explain this in the revised version of the paper. There is no special structural for $w$.
>
> **2. In line 90, the paper says the algorithm starts from the random initialization from the same initial region as in [13, 63]. In line 203, it says arbitrary initialization. I was bit confused while reading the paper on this point. And it becomes clear while reading line 312~313.**
>
> We thank the reviewer for pointing this out. We will modify the statements in lines 90 and 203 to make it more precise. We will mention explicitly that the local SGD converges from arbitrary initialization except for a measure zero set, where the angle of the first layer between initialization and the global minimum is $\pi$.
>
>
> **3. Just to clear my understanding, so your result is saying that weights in the first layer converges (in polynomial time), and then the weights in the second layer converges afterward?**
>
> We thank the reviewer for this insightful question. Lemmas 4 and 5 in Section 4.3 are only used to prove the convergence results, which are not saying the second layer converges afterward the first layer. They actually converge simultaneously without particular order. We will add this remark in our revision. Our simulation results in **Figure 1 of rebuttal PDF file** indeed show that two layers almost converge simultaneously after the self-correction process.
>
> **References**
>
> [13] Simon Du, Jason Lee, Yuandong Tian, Aarti Singh, and Barnabas Poczos. Gradient descent learns one-hidden-layer cnn: Don’t be afraid of spurious local minima. In International Conference on Machine Learning, pages 1339–1348. PMLR, 2018.
>
> [63] Mo Zhou, Tianyi Liu, Yan Li, Dachao Lin, Enlu Zhou, and Tuo Zhao. Toward understanding the importance of noise in training neural networks. In International Conference on Machine Learning, pages 7594–7602. PMLR, 2019.

---

> > ### Comment · Reviewer_bCYH · 2023-08-13
> >
> > No further questions from my side. I will keep my score as is

---

### Official Review · Reviewer_coFq · 2023-07-05

**Soundness:** 2 fair
**Presentation:** 1 poor
**Contribution:** 2 fair
**Rating:** 3
**Confidence:** 3

**Summary:**

The paper asserts that local SGD achieves convergence to the global minimum in the presence of Gaussian input. The experimental setup involves a two-layer student model, where the ground truth is generated by a two-layer teacher model. The authors highlight their main contribution as a proof that does not necessitate noise injection.

**Strengths:**

The paper proved global convergence of local SGD without noise injection.

**Weaknesses:**

1. The presentation of the results is unclear and the proof is difficult to follow. It would be beneficial for the authors to provide a more intuitive interpretation of the terms, lemmas, and equations. For instance:
- Could the authors clarify the purpose and significance of the intermediate step $\check{v}_{t+1}$?
- How does Theorem 1 demonstrate the self-correction of the second layer?

2. The realism of the assumptions made by the authors is questionable. For instance:
- What is the probability that a randomly selected $a^*$ satisfies equation (5)? It appears to be very low in my estimation.

3.  I'm uncertain why the authors chose to consider CNN instead of linear layers. It seems to add unnecessary confusion to the problem.

4. What sets "local" SGD apart from regular SGD? It would be helpful for the authors to clarify the distinctive aspects (in terms of proof) of "local" SGD in comparison to regular SGD.

**Questions:**

Please review the suggested revisions for the mentioned weaknesses:


Minor comments:

In line 167, could you please clarify the distinction between L(w, a; Z) and l(w, a; Z)?
In line 169, "wights" should be corrected to "weights".
In line 215, "learning" should be changed to "learning rate".

**Limitations:**


I believe it is important for the authors to provide justification for the assumptions made. Without such justification, the scope and applicability of the proposed results may be limited.

---

> ### Author Rebuttal · Authors · 2023-08-07
>
> Thank you for reviewing our paper.
>
> **1. Could the authors clarify the purpose and significance of the intermediate step $\check{v}\_{t+1}$.**
>
> The purpose of the intermediate step $\check{v}_\{t+1}$ is to establish an exact recursion to characterize how the global angle $\phi\_{t+1}$ changes in the training process.  In traditional analysis of local SGD, the descent inequality is obtained by taking the average on local weights and local gradients. However, this analysis is not applicable in our setting since we have to consider the angle $\phi_t$ instead of the inner product between $v_t$ and the ground truth. Due to the nonlinear relationship between the global angle $\phi_t$ and local angle $\phi_t^i$, we cannot get a global dynamic of $\phi_t$ by averaging local dynamics of $\phi_t^i$ as provided in [13]. To address this challenge, we introduce the virtual sequence $\check {v}\_{t+1}$ and use the corresponding angle $\check{\phi}\_{t+1}$ as a proxy between $\phi\_{t+1}$ and $\phi_t$.
>
> **2. How does Theorem 1 demonstrate the self-correction of the second layer?**
>
> The intermediate step $\check{v}\_{t+1}$ appears in the term $H_t$ in (3), so its significance in the proof hinges on the recursion (3). First, the sign of $\lambda_t \cos \phi_t$ in (3) determines the direction of the first layer’s update, and this observation enlightens us to analyze the self-correction of the first layer in different regions. Second, $H_t$ is the discrepancy term, $M\_{1,t}$ is a martingale difference sequence, and $M\_{2,t}$ is the variance term of noise. Hence the recursion (3) actually resembles the descent inequality of local SGD analysis for general functions. It helps us show the linear speedup and reduce communication rounds in the convergence stage.
>
> Du et al. [13] proved that GD can converge to the global minimum if the initialization is in the attraction basin, that is $\{(v_0, a_0): \phi_0 < \pi/2, a_0^{\prime}a^* > 0\}$. In our paper, we consider local SGD with almost arbitrary initialization including the bad area with wrong signals such that $\phi_0 > \pi/ 2$ or $a_0^{\prime}a^* \leq 0$. The self-correction process means SGD or local SGD can correct the wrong signals from the initialization and enter the attraction basin in polynomial time. In Theorem 1, $\tau_a$ is the time stamp when the signal of the second layer turns positive (and the initial signal could be negative). Therefore, $\tau_a \leq O(\eta^{-1} \log k)$ means that the signal of the second layer can be corrected in $O(\eta^{-1} \log k)$ steps for any initialization.
>
> **3. The realism of the assumptions made by the authors is questionable. For instance: What is the probability that a randomly selected $a^{*}$ satisfies equation (5)? It appears to be very low in my estimation.**
>
> Since $a^*$ is a fixed vector in our paper, we respectfully disagree that one can use random selection to evaluate the condition Eq.(5). But it can be satisfied when $a^*$ is dense. For example, we let $\alpha = \frac{(1^{\prime}a^*)^2}{k |a^*|^2}$ where $\alpha \in (0,1]$ roughly stands for the degree of sparsity of the vector $a^{*}$. Then Eq.(5) will hold if
> $$\alpha > (1-32(\pi-1)/k)^{-1}\frac{\pi +k-1}{(\pi-1)k}.$$
> This right-hand side can be a constant in $(0,1)$ for proper $k$. For example, when $k \geq 320(\pi-1)^2$ (a technical choice as in Theorem 1), this condition will be satisfied as long as $\alpha > 0.48$.
>
> We have added an additional simulation with different $\alpha$ to show that this condition is almost necessary to show the global convergence with arbitrary initialization. In each trial, we randomly select an initial point from the initial region defined in our paper. Then we calculate the probability of convergence over 100 independent trials with $k=64$. From **Table 1 in the rebuttal PDF file**, we can see the probability of convergence becomes larger when $\alpha$ tends to 1. Specifically, the probability becomes 1 when $\alpha > 1/8$. Even though the coefficient in our condition on $\alpha$ may require it to be greater than $1/4$ due to the technical relaxation, the results indicate that a constant $\alpha$ is necessary.
>
> **4. I'm uncertain why the authors chose to consider CNN instead of linear layers. It seems to add unnecessary confusion to the problem.**
>
> We apologize for the confusion. Please note that we follow the assumption and terminology in Du et al. [13] and Zhou et al. [63] to refer to the network as a CNN. We acknowledge that it is indeed a fully-connected network since there is no overlap between patches in CNN. In our paper, we elect to refer to the network as CNN to keep the terminology consistent with the most related works [13, 63]. We will add a sentence to explain this in the revised version of the paper.
>
> **5. What sets "local" SGD apart from regular SGD? It would be helpful for the authors to clarify the distinctive aspects (in terms of proof) of "local" SGD in comparison to regular SGD.**
>
> We thank the reviewer for this suggestion. For regular SGD (that is, single machine with $N=1$), the intermediate step $\check{v}\_{t+1}$ equals to $v\_{t+1}$, so the discrepancy $H_t$ is zero. Then the distinctive aspects of local SGD in the proof are how to quantify and control the scale of the discrepancy term $H_t$. The traditional analysis in distributed nonconvex optimization relies on the bounded smoothness condition to bound the discrepancy term and attain linear speedup. Our paper carries out a more careful analysis because the loss function of the neural network model has an unbounded smooth parameter.
>
> **6. In line 167, could you please clarify the distinction between $L(w, a; Z)$ and $\ell(w, a; Z)$?**
>
> We apologize for this. We first want to clarify that only $\ell(v,a;Z)$ is used in our paper, and there is no definition for $\ell(w,a;Z)$. The loss $\ell$ is defined in terms of $v$, and the loss $L$ is defined in terms of $w = v/\|v\|$. We will take care to address this in the paper.

---

> > ### Author Response · Authors · 2023-08-16
> > **Looking forward to rebuttal feedback**
> >
> > Dear Reviewer coFq,
> >
> > Thank you for reviewing our paper again! We have posted our responses to your questions and concerns. We are wondering if you have a new evaluation of our paper after reading our responses. If you have other questions, we are very happy to discuss them with you.
> >
> > Best,
> >
> > Authors

---

> > > ### Comment · Reviewer_coFq · 2023-08-16
> > >
> > > I appreciate the authors for their comprehensive responses. However, my concerns about Assumption 1 persist. While the teacher's weight $a^\star$ is a fixed vector, it's crucial to determine whether it is likely to satisfy Assumption 1 since it is provided by nature. I remain unconvinced that "how likely $a^\star$ meets all the stipulated inequalities"?
> > >
> > > - $k\geq 320(\pi-1)^2$
> > > - $(1^\intercal a^\star)^2 \leq \frac{k(\pi+k-1)}{720\pi \log (4+k^2)}$
> > > - $\left(1-\frac{32(\pi-1)}{k}\right)\frac{\pi-1}{\pi+k-1} \frac{(1^\intercal a^\star)^2}{||a^\star||^2} > 1$.
> > >
> > > If Assumption 1 is only valid for specific choices of $a^\star$ and $k$, then the findings would be narrowly applicable, limiting their relevance to a constrained set of teacher setups.
> > >
> > > As a case in point, do the $a^\star$ values in the supplementary PDF adhere to Assumption 1?
> > > Despite experimental evidence, which may be contingent on specific choices of $a^\star$, the theorem appears to impose overly stringent constraints. Given these observations, I have chosen to retain my original score.

---

> > > > ### Author Response · Authors · 2023-08-16
> > > > **Response to the comment**
> > > >
> > > > Thanks for your response. We still have several points to clarify.
> > > >
> > > > 1. Condition (5) is used to show the global convergence of local SGD with **arbitrary initialization**, which has not been exploited even for SGD or GD in prior works. The analysis of arbitrary initialization is very difficult due to the interplay of signals from two layers. Even for GD, Du et al. [13] only proved the convergence to the global minimum if the initial points are from the attraction basin ($\phi_0 < \pi/2, a_0^{\prime}a^* > 0$). In addition, Du et al. [13] also proved that GD from the initial point such that $\phi_0 > \pi/2, a_0^{\prime}a^* < 0$ will converge to the local minimum if $|1^{\prime}a^*|^2/\| a^*\|^2 < 1/poly(p)$. The simulation in Du et al. [13] investigated the convergence probability of GD with random initialization. The results are very similar to ours in the rebuttal PDF file, which at least show that global convergence with arbitrary initialization cannot be guaranteed if the ratio $\alpha = |1^{\prime}a^*|^2/(k \| a^*\|^2)$ doesn’t exceed some constant threshold. Our condition (5) is equivalent to
> > > > $$
> > > > \alpha \geq \(1 - \frac{32(\pi-1)}{k}\)^{-1} \frac{\pi+k-1}{(\pi-1)k}.
> > > > $$
> > > > The right-hand side is constant in (0,1) with proper $k$. Therefore, our condition is reasonable and consistent with the simulation results.
> > > >
> > > > 2. In Assumption 1, $k \geq 320(\pi - 1)^2$ and $|1^{\prime}a^*|^2 \leq \frac{k(\pi+k-1)}{720 \pi \log(4+k^2)}$ are technical conditions, and the latter can be easily satisfied as long as $\|a^*\|$ is not too large since $|1^{\prime}a^*|^2 \leq k\|a^*\|^2$. Next, we introduce a random scheme on the sparsity of $a^*$. Let $a^* = (1,...,1,0,...,0)/\sqrt{s}$, where $s$ is number of nonzero entries in $a^*$. We assume $s$ is uniformly distributed in $\{1,2,...,k\}$. Then the probability that condition (5) holds is
> > > > $$
> > > > 1 - \frac{1}{k} \lceil (1 - \frac{32(\pi-1)}{k}\)^{-1} \frac{\pi+k-1}{\pi-1}\rceil.
> > > > $$
> > > > The probability is greater than 0.5 if $k \geq 320(\pi - 1)^2$.

---

### Official Review · Reviewer_p3Vw · 2023-08-02

**Soundness:** 3 good
**Presentation:** 2 fair
**Contribution:** 3 good
**Rating:** 5
**Confidence:** 2

**Summary:**

This paper considers a federated learning setting for a  one-hidden-layer convolutional neural network without overparameterization. It is proven that vanilla local SGD (where in each iteration each node updates the model by the local gradient and then all nodes synchronize the model parameters) can converge to a global minimum.

**Strengths:**

1. The convergence analysis does not require over-parameterization (i.e., in the NTK region) and injected noise.

2. Based on a careful landscape study, the paper proposes a self-correction mechanism that ensures the algorithm enters a good landscape region in polynomial time. This explains why local SGD can converge in practice even though the landscape is not well-conditioned.

**Weaknesses:**

1. The online learning assumption, i.e., each node samples data i.i.d. from the same distribution in each iteration, is crucial to the analysis of this paper, but it deviates from the practical settings, where each node computes its gradient according to a given set of local data.

2. Why is the proposed model a CNN? It seems a fully-connected network.

3. The weight-normalization technique twists the vanilla SGD.

**Questions:**

Please see "weakness".

**Limitations:**

I don't think the authors adequately addressed the limitations.

---

> ### Author Rebuttal · Authors · 2023-08-07
>
> Thanks for taking your precious time to review our paper! Next, we provide detailed responses to your comments and questions.
>
> **1. The online learning assumption, i.e., each node samples data i.i.d. from the same distribution in each iteration, is crucial to the analysis of this paper, but it deviates from the practical settings, where each node computes its gradient according to a given set of local data.**
>
> Thank you for this important point. We agree that non-i.i.d. is a relevant problem in federated learning. However, one of the points we want to make in the paper is that the global convergence of SGD under one hidden layer neural network without overparameterization in the single-machine setting is still unknown until our paper. Specifically, due to the non-convexity, prior works only show the local convergence of GD [13] and the global convergence of perturbed GD by injecting noise [63]. Currently, there is no literature analyzing vanilla SGD for this neural network, let alone local SGD. Our paper proves the global convergence of local SGD under this setting, which is even more challenging than SGD since we have to handle the effects of local steps. We devote significant efforts to establishing the new recursive dynamics and developing new analysis techniques (namely, self-correction) to demonstrate the convergence to global minima with almost arbitrary initialization, despite the loss landscape being nonconvex. Note that the technique in local SGD for general nonconvex problems can only guarantee finding a stationary point instead of a global minimum [56,62].
>
> **2. Why is the proposed model a CNN? It seems a fully-connected network.**
>
> We apologize for the confusion. Please note that we follow the assumption and terminology in Du et al. [13] and Zhou et al. [63] to refer to the network as a CNN. We acknowledge that it is indeed a fully-connected network since there is no overlap between patches in CNN. In our paper, we elect to refer to the network as CNN to keep the terminology consistent with the most related works [13, 63]. We will add a sentence to explain this in the revised version of the paper.
>
> **3. The weight-normalization technique twists the vanilla SGD.**
>
> The vanilla SGD in our paper refers to optimizing the loss function of the model defined in Eq.(2) after applying weight-normalization, which is a function of $v$ and $a$. Note that the description of Algorithm 1 on page 4 is indeed vanilla SGD because it applies stochastic gradient updates over $(v,a)$ but not $(w,a)$.
>
> Applying the weight normalization is consistent with Du et al. [13] and Zhou et al. [63]. The GD algorithm in Du et al. [13] and the perturbed GD in Zhou et al. [63] are also conducted under the same model. The original network $f(Z, w, a)$ has a positive-homogeneity issue: for any $c>0$, it holds $f(Z, cw, a/c) = f(Z, w, a)$. This property allows the network to be rescaled without changing the function value. It also implies that only the direction of the first layer $w$ will essentially affect the loss function. The weight-normalization technique makes the learning algorithm scaling-invariant and can stabilize the training process [46].
>
> **References**
>
> [13] Simon Du, Jason Lee, Yuandong Tian, Aarti Singh, and Barnabas Poczos. Gradient descent learns one-hidden-layer cnn: Don’t be afraid of spurious local minima. In International Conference on Machine Learning, pages 1339–1348. PMLR, 2018.
>
> [46] Tim Salimans and Durk P Kingma. Weight normalization: A simple reparameterization to accelerate training of deep neural networks. Advances in neural information processing systems, 29, 2016.
>
> [56] Hao Yu, Rong Jin, and Sen Yang. On the linear speedup analysis of communication efficient momentum sgd for distributed non-convex optimization. arXiv preprint arXiv:1905.03817, 2019.
>
> [62] Fan Zhou and Guojing Cong. On the convergence properties of a k-step averaging stochastic gradient descent algorithm for nonconvex optimization. arXiv preprint arXiv:1708.01012, 2017.
>
> [63] Mo Zhou, Tianyi Liu, Yan Li, Dachao Lin, Enlu Zhou, and Tuo Zhao. Toward understanding the importance of noise in training neural networks. In International Conference on Machine Learning, pages 7594–7602. PMLR, 2019.

---

> > ### Comment · Reviewer_p3Vw · 2023-08-19
> >
> > Thanks to the author for the comprehensive response.
> >
> > I think this paper has made a considerable contribution even under the i.i.d. data assumption.
> >
> > For the CNN issue, I don't think it's appropriate to use a misleading term in order to "keep terminology consistent with the most related works". I also see two other reviewers that have raised this issue.
> >
> > My prior review "Weight normalization twists the vanilla SGD" is not entirely precise. What I wanted to convey is that the weight normalization re-parameterizes the model. Consequently, the analysis of SGD on the re-parameterized model diverges from that of the original model. The author argued that this re-parameterization was also adopted in existing works [13], [63]. However, [13] and [63] are pioneer works in this field, dating back at least 3 years. Rather than adhering to the unrealistic assumptions in these prior works, it would be more valuable to take efforts to relax these assumptions.
> >
> > (By the way, [46] was published 7 years ago. It was indeed an insightful work. However, the experiments were carried out on MNIST and Cifar-10. To my knowledge, practitioners nowadays do not use such type of weight normalization in deep networks.)
> >
> > I will keep the score as it is.

---

> > > ### Author Response · Authors · 2023-08-20
> > >
> > > Dear Reviewer p3Vw,
> > >
> > > Thanks for your response. We will fix the terminology of CNN in the revised version. However, we respectfully disagree with the statement that weight normalization is unrealistic. This technique was proposed to stabilize the training process of neural networks, which also inspired similar techniques in recent years (e.g., Year 2021~2023). For example, a well-known paper, Miyato et al. [1] generalized the weight normalization to spectral normalization and applied it to stabilize the training of the discriminator in Generative Adversarial Networks. Bjorck et al. [2] also used this technique in the training of deep reinforcement learning, which enables stable training with large modern architectures. Liu et al. [3] proposed a spectral-normalized neural Gaussian process that applies spectral normalization to hidden weights to enforce bi-Lipschitz smoothness in the hidden layer and improves the quality of hidden representations of neural networks. Please note that [2] and [3] were published very recently: [2] was published in NeurIPS 2021, and [3] was published in JMLR 2023. Therefore, we believe that weight normalization is still relevant in the training of deep neural networks.
> > >
> > > [1]  Miyato, Takeru, Toshiki Kataoka, Masanori Koyama, and Yuichi Yoshida. "Spectral normalization for generative adversarial networks."  ICLR 2018.
> > >
> > > [2] Bjorck, Nils, Carla P. Gomes, and Kilian Q. Weinberger. "Towards deeper deep reinforcement learning with spectral normalization." Advances in Neural Information Processing Systems 34 (2021): 8242-8255.
> > >
> > > [3] Liu, Jeremiah Zhe, Shreyas Padhy, Jie Ren, Zi Lin, Yeming Wen, Ghassen Jerfel, Zachary Nado, Jasper Snoek, Dustin Tran, and Balaji Lakshminarayanan. "A Simple Approach to Improve Single-Model Deep Uncertainty via Distance-Awareness." J. Mach. Learn. Res. 24 (2023): 42-1.
> > >
> > > Best,
> > >
> > > Authors

---

### Author Rebuttal · Authors · 2023-08-07

Dear Reviewers:

We appreciate your constructive reviews and would like to thank you for your time in helping us to improve our paper. We have responded to the reviews and questions individually and have added several new simulations in the PDF file. Here we provide a general response to address the two frequently asked questions by reviewers about the terminology and assumption in the paper:

**Regarding the terminology of CNNs.** We apologize for the confusion. We have followed the assumption and terminology in Du et al. [13] and Zhou et al. [63] to refer to the network as a CNN. We acknowledge that it is indeed a fully-connected network since there is no overlap between patches in the CNN. In our paper, we elect to refer to the network as a CNN so that we can keep the terminology consistent with the most related works [13, 63]. We will add a sentence to explain this in the revised version of the paper.

**Regarding the Assumption in Theorem 1.** To illustrate the necessity of condition (6) to guarantee the global convergence with almost arbitrary initialization, we calculate the probabilities of convergence to the global minimum under different values of $(1^{\prime}a^*)^2/\|a^*\|^2$ in the new simulations. **Table 1 in the rebuttal PDF file** shows that the condition where the probability of converging to the global minimum reaches 1 is very close to condition (6). Therefore, we believe (6) is nearly essential to the self-correction mechanism as illustrated in Theorem 1.

[13] Simon Du, Jason Lee, Yuandong Tian, Aarti Singh, and Barnabas Poczos. Gradient descent learns one-hidden-layer cnn: Don’t be afraid of spurious local minima. In International Conference on Machine Learning, pages 1339–1348. PMLR, 2018.

[63] Mo Zhou, Tianyi Liu, Yan Li, Dachao Lin, Enlu Zhou, and Tuo Zhao. Toward understanding the importance of noise in training neural networks. In International Conference on Machine Learning, pages 7594–7602. PMLR, 2019.

---

### Decision · Program_Chairs · 2023-09-21

**Decision:**

Accept (poster)

**Comment:**

The paper analyzes the convergence behavior of local SGD for training two-layer neural networks in a federated learning setup. It studies this problem in a teacher-student model in which the goal is to reproduce the output of the teacher network, and the input vectors are drawn from a Gaussian distribution. Local SGD interleaves gradient updates based on local data with consensus (averaging) steps. The paper analyzes the dynamics of both network layers, and identifies a ``self-correction’’ mechanism, by which, if the second layer weights have coherent signs (in the sense that <1,a> is large compared to ||a||), in polynomial time the iterates enter a basin in which they then exhibit linear convergence to the ground truth. For the federated version of the problem, the paper shows a linear speedup in the number of machines.

 Reviewers produced a split evaluation of the paper. The main concern involved the condition on the ground truth second layer weights a_*. Secondary issues include terminology (this is not a convolutional neural network) and the paper’s relatively dense presentation. On the positive side, the paper produces novel results on local stochastic gradient descent for federated learning, and, as a by-product, strengthens existing results on single machine SGD, by articulating conditions under which gradient descent is converges globally, without theoretical conveniences such as noise injection. The author response addresses issue with the a_* conditions in two ways: first, by arguing that this assumption (or some similar form of sign consistency) is almost necessary for the global convergence of GD, and second, by arguing that under certain random models, this condition holds with high probability. In the final version, the authors are strongly encouraged to (i) avoid the terminology of convolutional neural networks, (ii) provide additional discussion of the paper’s conditions on a_* (e.g., by including material from the response to reviewers).